# Structural Knowledge Informed Continual Multivariate Time Series Forecasting

## Abstract

Recent studies in multivariate time series (MTS) forecasting reveal that explicitly modeling the hidden dependencies among different time series can yield promising forecasting performance and reliable explanations. However, modeling variable dependencies remains underexplored when a sequence of MTS under different regimes (stages) is continuously accumulated. Due to the potential distribution and dependency disparities, the underlying model may encounter the catastrophic forgetting problem, *i.e.*, it is challenging to memorize and infer different types of variable dependencies across different regimes while maintaining forecasting performance. To this end, we propose a novel Structural Knowledge Informed Continual Learning (SKI-CL) framework to perform MTS forecasting under the continual learning setting, which leverages the structural knowledge to characterize the dynamic variable dependencies within each regime. Specifically, we first develop a deep forecasting model with a graph learner that enables fine-grained dynamic MTS modeling. Next, we impose a regularization scheme to ensure the consistency between the learned variable dependencies and the structure knowledge (*e.g.*, physical constraints, domain knowledge, feature similarity). Finally, we develop a representation-matching memory replay scheme to tackle the catastrophic forgetting problem, which maximizes the temporal coverage of MTS data to efficiently preserve the underlying temporal dynamics and dependency structures of each regime. Thoroughly empirical studies on synthetic and real-world benchmarks validate SKI-CL's efficacy and advantages over the state-of-the-art in tackling continual MTS forecasting tasks. In addition, SKI-CL can accurately infer learned dependencies in the test stage based on MTS data without knowing the identities of different regimes.

## 1 Introduction

Multivariate time series (MTS) forecasting aims to predict the future horizons of multiple time series based on their historical values and has shown its importance in various applications, *e.g.*, healthcare, traffic control, energy management, and finance Jin et al. (2018); Gonzalez-Vidal et al. (2019); Guo et al. (2019); Zhang et al. (2017). Accurate forecasting not only relies on capturing temporal dynamics of the historical multivariate time series data, *e.g.*, autoregressive integrated moving average (ARIMA) Box et al. (2015), deep convolutional neural networks van den Oord et al. (2016); Bai et al. (2018); Borovykh et al. (2017), recurrent neural networks Bai et al. (2018); Lai et al. (2018); Shih et al. (2019), and transformer models Zhou et al. (2021); Wu et al. (2021); Zhou et al. (2022); Liu et al. (2021), but also relies on modeling dependency structures among different time series Graves (2013); Lai et al. (2018); Shih et al. (2019). However, the nonstationary nature of most MTS data implies that the structural dependencies among different time series vary across time. To make it worse, in real-world applications, different regimes (stages) of MTS data are often continuously collected under different internal logic of the target system.

To develop an effective forecasting model that can handle continuously accumulated MTS data from different regimes and infer the dependency structures accordingly, there are two fundamental challenges. The first one is how to model dependency structures among different time series by incorporating the structural knowledge of the target system. Recently, there is increasing evidence suggesting that jointly optimizing the inferred structure toward the existing structural knowledge and

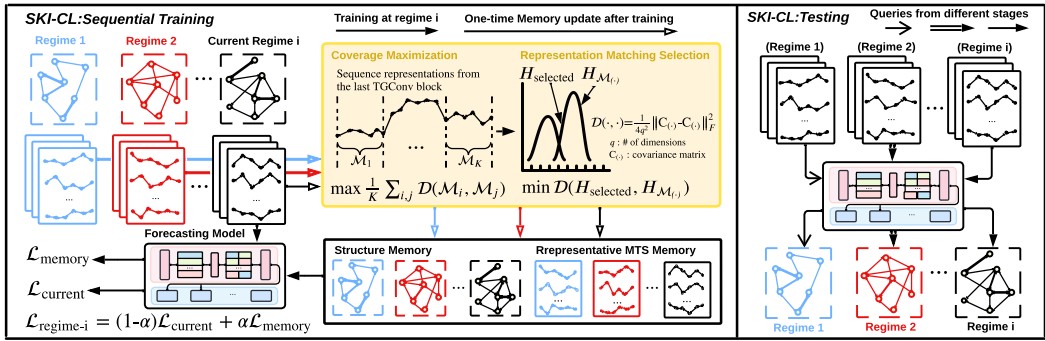

Figure 1: The proposed SKI-CL framework for continual MTS forecasting. The training objectives for each regime contains the current training data and the memory buffer. After training at each regime, the structural knowledge and samples selected by our representation-matching scheme are added to the current memory. At testing phase, SKI-CL is able to dynamically infer faithful dependency structures for different regimes without accessing the memory buffer.

forecasting objectives can render reliable explanations and competing performance Shang & Chen (2021). The structural knowledge can be available in different formats, *e.g.*, physical constraints such as traffic networks, power grids, and sensor networks Li et al. (2017); Luo et al. (2021); Khodayar & Wang (2018); Yan et al. (2018), or dependencies (correlations) that are inferred or derived from raw data by leveraging either domain knowledge or traditional statistical methods Chen et al. (2022); Duan et al. (2022); Lin et al. (2021); Cao et al. (2020); Shang & Chen (2021). The second challenge is how to keep track of the latest regime of MTS data while maintaining the performance on existing regimes. One potential solution is to retrain the forecasting model periodically over the newly collected regime to incorporate the latest structural knowledge. This method, however, will lead to the catastrophic forgetting problem, *i.e.*, the model performance will deteriorate over existing regimes as their associated structural knowledge cannot be maintained. Note that joint training may not be a feasible solution due to the memory constraints of storing all historical data (different regimes) and the computational complexity of handling an ever-increasing number of diverse scenarios.

In this paper, we present a novel Structural Knowledge Informed Continual Learning (SKI-CL) framework that incrementally learns and preserves meaningful dependency structures for MTS forecasting. As shown in Figure 1, the key idea is to exploit structural knowledge to characterize the variable dependencies within each different regime. Specifically, we first design a dynamic graph structure learning module that captures the temporal patterns to construct the parameterized interactions between all variable pairs. Instead of learning a single fixed graph structure Bai et al. (2020); Wu et al. (2020), our module dynamically infers dependency structures (in the form of graphs) based on different MTS input windows to cope with the dependencies variations within each regime, thus enabling non-stationary MTS modeling. We further enforce the consistency between the learned structures and the structural knowledge by designing an adaptive regularization scheme that accommodates different forms of structural knowledge that exist in MTS. Four common scenarios regarding the description of dependencies and the availability of structural knowledge is considered, *i.e.*, the discrete/continuous edge description, and fully/partially observed structural knowledge. Finally, we present a novel representation-matching memory replay scheme to tackle the catastrophic forgetting problem, which maximizes the temporal coverage of MTS data to efficiently preserve the underlying temporal dynamics and dependency structure of each regime (Figure 1(middle)). By jointly learning from the current regime and the constructed memory, the obtained model is able to maintain the forecasting performance and infer the learned structures from the existing regimes.

In summary, our work makes the following four contributions. (1) We present a novel Structural Knowledge Informed Continual Learning (SKI-CL) framework to perform MTS forecasting and infer dependency structures under the continual learning setting. (2) We develop a dynamic graph structure learning module to capture temporal dependencies and infer dependency structures, and employ an adaptive regularization scheme to incorporate structural knowledge. (3) We propose a novel representation-matching memory replay scheme to maximize the temporal coverage of MTS data and preserve the underlying temporal dynamics and dependency structures of each regime. (4) Thoroughly experiments on one synthetic dataset and three benchmark datasets demonstrate the superiority of SKI-CL over the state-of-the-art in continual MTS forecasting and dependency structure inference.

## 2 RELATED WORK

### 2.1 MODELING DEPENDENCIES IN MULTIVARIATE TIME SERIES FORECASTING

In recent years, modeling variable dependencies of MTS has received increasing attention for forecasting tasks. Early methods apply linear or convolution transformations to capture variable dependencies in an implicit recurrent process Lai et al. (2018); Graves (2013); Shih et al. (2019), which fall short of modeling the non-Euclidean interactions due to the underlying fully-connected or translation-invariant assumptions. The advent of Graph Neural Networks (GNNs) has inspired the formulation of variable dependencies as a given or learnable graph, with variables being nodes and pairwise relationships being edges. Existing literature models the dependency structures based on different topological perspectives and temporal granularity (*e.g.*, undirectedYu et al. (2018) and directed graph Li et al. (2017), static Bai et al. (2020); Wu et al. (2020) and dynamic graphs Ye et al. (2022); Cao et al. (2020), single or multiple layers Lin et al. (2021)). On the other hand, structural knowledge has been an important component in GNN-based forecasting methods. In many tasks such as the traffic Guo et al. (2019); Li et al. (2017) and skeleton-based action prediction Yan et al. (2018), structural knowledge is explicitly presented as spatial connections. In other cases without an explicit topological structure, the structural knowledge can be drawn from either domain knowledge (*e.g.*, transfer entropy Duan et al. (2022), Mel-frequency cepstral coefficients Lin et al. (2021)) or feature similarity (*e.g.*, the correlations of decomposed time series Ng et al. (2022), kNN graph Shang & Chen (2021)). Their promising results suggest the capability of structural knowledge to convey meaningful dependency information. Our proposed method enforces the consistency between the learned graph structures and the structural knowledge so as to characterize the underlying relation-temporal dependencies and improve the continual MTS forecasting performance.

### 2.2 CONTINUAL LEARNING IN MULTIVARIATE TIME SERIES FORECASTING

Deep learning models resort to continual learning to address the catastrophic forgetting issue when sequentially adapting to new tasks. Existing literature in continual learning lies in three folds: the experience-replayed methods Rolnick et al. (2019), the parameter-isolation methods Rusu et al., and regularization-based methods Kirkpatrick et al. (2017); Li & Hoiem (2017). Current continual learning works have been extensively studied on images Wang et al. (2022), texts Ke & Liu (2022), and graph data Zhang et al. (2022a;b). However, much less attention is drawn to time series data, and the focus has primarily been on classification and forecasting tasks without explicitly addressing complex variable dependencies Gupta et al. (2021); He & Sick (2021). How to maintain the meaningful dependency structures and forecasting performance over different regimes is underexplored. Our proposed SKI-CL tackles this issue by jointly optimizing the inferred structure toward the existing structural knowledge and forecasting objectives based on samples drawn from the current regime and a representative memory.

We also notice that recent efforts have been made to perform forecasting Pham et al. (2022); Zhang et al. (2023) and/or topology identification Money et al. (2021); Natali et al. (2022); Isufi et al. (2019); Zaman et al. (2020) from MTS data in an online learning setting, which focus on adapting the forecasting model and dependency structure from the historical MTS data to future unseen data. On the contrary, our study aims to maintain the forecasting performance and infer the learned structures from the existing regimes while continuously updating the model over the latest regime. Without forgetting the structure knowledge that has been acquired previously, the model can easily cope with similar regimes that may be encountered in the future. In addition to online MTS forecasting, we discuss the relationships between our study and other related research topics in Appendix B.

## 3 METHODOLOGY

In this section, we first formulate the continual MTS forecasting task with dependency structure learning. Then, we introduce the proposed framework including the structural knowledge-informed graph learning model and the representation-matching sample selection scheme for continual MTS forecasting, as shown in Figure 2 and Figure 1(middle), respectively.

### 3.1 PROBLEM STATEMENT

We first introduce the MTS forecasting task in a single regime (stage). Let $X \in \mathbb{R}^{N \times T}$ denote the MTS data containing $N$ variables and $T$ total time steps, where $X_{:,t} \in \mathbb{R}^{N \times 1}$ denotes $t$-th time

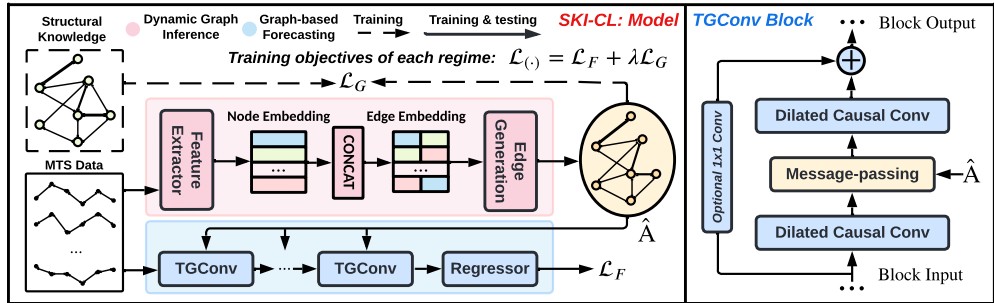

Figure 2: The proposed SKI-CL Model for dependencies modeling and MTS forecasting.

step across all variables and $X_{i,:} \in \mathbb{R}^{1 \times T}$ denotes $i$-th variable. Our target is to learn a model that includes a dynamic graph inference module $\mathcal{G}(\cdot)$ summarizing a historical $\tau$-step window of MTS as a graph to encode the dependency structure, as well as a forecasting module $\mathcal{F}(\cdot)$ predicting the next $\tau'$ time steps based on the input window and inferred graph. Mathematically, at a starting time step t, the corresponding forecast is defined as: $\hat{X}_{:,t+\tau:t+\tau+\tau'-1} = \mathcal{F}(X_{:,t:t+\tau-1}, \mathcal{G}(X_{:,t:t+\tau-1}))$.

We further extend the forecasting task to a continual learning setting. In continual learning, there exist $S$ distinct regimes of MTS data with different dependencies and temporal dynamics. The model can only access MTS data of the current regime. Denoting the data of $s$-th regime as $X^{(s)}$, the objective is to learn a model to minimize the forecasting error across all seen regimes: $\mathcal{F}^*, \mathcal{G}^* = \arg\min_{\mathcal{F},\mathcal{G}} \sum_{s=1}^{\mathcal{S}} L\left(\hat{X}_{:,t:t+\tau-1}^{(s)}, X_{:,t+\tau:t+\tau+\tau'-1}^{(s)}\right)$ with $L$ being the loss function. We assume there is a readily available or extracted structural knowledge $A \in \mathbb{R}^{N \times N}$ (either partial or completed) at each regime that serves as a reference to characterize the underlying dependencies.

## 3.2 STRUCTURAL KNOWLEDGE INFORMED GRAPH LEARNING FOR MTS FORECASTING

**Dynamic Graph Inference Module**. Different from the existing works that generate a static graph at the regime level Wu et al. (2020); Bai et al. (2020); Shang & Chen (2021), we aim to model the non-stationary variable dependencies of MTS as a dynamic graph at the granularity of an input window. Therefore, as shown in Figure 2, we construct a dynamic graph inference module that more precisely reveals the relation-temporal dynamics in a single regime and has the capacity to handle dependencies change when the regime shifts.

In this module, each edge is explicitly modeled for all node pairs, which is parameterized as follows. For a possible edge connecting node $i$ and $j$, we use the stacked convolution $p$ times along the temporal dimension as a feature extractor to yield node embedding $z_i$ and $z_j$ (*i.e.*, $z_i = (\mathbf{FC} \circ \mathbf{Conv}^{[p]})(X_{i,t:t+\tau-1})$) which are concatenated as the edge embedding. Next, $p'$ fully connected layers finalize the generation of a parameterized edge as $\hat{A}_{ij} = \mathbf{FC}^{[p']}(z_i\|z_j)$. The output graph $\hat{A} \in \mathbb{R}^{N \times N}$ summarizes the variable interactions from the temporal dynamics within a sequence, and can be further utilized to generate the forecasts.

**Incorporating Structural Knowledge for Dependencies Characterization**. To learn a faithful dynamic graph structure that characterizes the underlying dependencies of each regime, we incorporate structural knowledge as a reference in learning objectives. In real-world MTS modeling, the edges can be either continuous if we can quantify the strength in the context, or binary if we are more confident of the connection in a qualitative sense (*e.g.*, physical connections). Moreover, it also interleaves with the fact that if the structural knowledge can be fully observed, as in many cases we are only confident in the existence of certain relationships.

To fully leverage different forms of structural knowledge in dependency structure learning, we design an adaptive scheme that imposes different constraints on the parameterized graph in the objective function, which is denoted as $\mathcal{L}_G$ as shown in Figure 2 (left). If an edge is treated as a binary variable, we activate the parameterized edge with a sigmoid function to approximate the Bernoulli distribution $\hat{A}_{ij} \sim \text{Bern}(\theta_{ij})$, where $P(\hat{A}_{ij} = 1) = \theta_{ij}$ is the probability that an edge is forming between node $i$ and node $j$. Then, we encourage the probability of edge to be consistent with the prior, which essentially minimizes the binary cross entropy: $\mathcal{L}_G = \sum_{i,j} -A_{ij} \log \theta_{ij} - (1 - A_{ij}) \log (1 - \theta_{ij})$.

If an edge is treated as a continuous variable, we activate the parameterized edge with a ReLU function to remove the weak connections, and enforce the consistency between the numerical values of the parameterized edge and prior, representing a similar interaction strength. This is achieved by the MSE objective $\mathcal{L}_{\mathrm{G}} = \frac{1}{N^2} \sum_{i,j} \| A_{ij} - \hat{A}_{ij} \|^2$. So far, we have discussed the cases when structural knowledge is readily available/fully observed. For partially observed structural knowledge, we only enforce the consistency between the known entries in the structural knowledge and corresponding parameterized edges, as the dynamic graph inference module is still able to capture and infer the underlying dynamic dependencies via optimizing based on the existing structural knowledge and the forecasting objective.

**Graph-based Forecasting Module**. To further exploit the structural and temporal dependencies and render forecasts, we design a graph-based forecasting module consisting of multiple Temporal Graph Convolution (TGConv) blocks, as shown in Figure 2. In each block, we leverage a dilated causal convolution Bai et al. (2018); van den Oord et al. (2016) to effectively captures forward dynamics of time series. The dilated causal convolution operation on a 1D sequence input $\mathbf{h}$ is expressed as $r_t = \sum_{k=0}^{\mathcal{K}-1} f(k) \cdot h_{t-d \cdot k}$, where $r_t$ denotes the $t$-th step of obtained representation $\mathbf{r}$, $d$ represents dilation factor, $f(k)$ is the convolution kernel with size $k$. Since dilated causal convolution exclusively processes univariate time series, we facilitate the modeling of dependencies in MTS by exchanging and aggregating information through the learned structure, denoted as $\hat{A}$. This is achieved by a simple yet effective message-passing neural operation Morris et al. (2019), given the collection of univariate representations $(\mathbf{r}_1, \cdots, \mathbf{r}_N)$:

$$\mathrm{MessagePassing}_{\hat{A}}(\mathbf{r}_i) = \mathbf{W}_1 \mathbf{r}_i + \mathbf{W}_2 \sum_{j \in \mathcal{N}(i)} e_{j,i} \cdot \mathbf{r}_j \tag{1}$$

where $\mathcal{N}(i)$ represents the neighbors of variable $i$, $e_{j,i} \in \hat{A}$ denotes an edge weight, $\mathbf{W}_{(\cdot)}$ denotes the learnable weights, and the bias term is omitted for simplicity.

We stack both operations to construct a TGConv block, with an optional $1 \times 1$ convolution tackling the possibly different dimensions between the residual input and output. Finally, a fully connected layer serves as a regressor projecting sequence representations onto the forecasts. We adopt the mean squared error between the forecasts and the ground truths as the main learning objective $\mathcal{L}_F$. The total learning objective function for each regime consists of the forecasting objective and the consistency regularization weighted by a hyperparameter $\lambda$.

$$\mathcal{L}_{total} = \mathcal{L}_F + \lambda \mathcal{L}_G = \frac{1}{\tau'} \sum_{t=t+\tau}^{t=t+\tau+\tau'-1} \| \hat{X}_{:,t} - X_{:,t} \|^2 + \lambda \mathcal{L}_G \tag{2}$$

### 3.3 REPRESENTATION-MATCHING SAMPLE SELECTION FOR CONTINUAL MTS FORECASTING

To tackle the forgetting of variable dependencies and temporal dynamics in sequential training, we store a small subset of MTS samples and the structural knowledge from the previous regimes for memory replay when adapting the model to the current regime. Specifically, we propose an efficient sample selection scheme that maximizes the temporal and dependencies coverage of each regime given a limited memory budget. According to the principle of maximum entropy Guiasu & Shenitzer (1985), we can best represent the underlying knowledge of MTS in each regime with the largest entropy, namely, with the most diverse partitions/modes of relational and temporal patterns. Inspired by this principle and its success in characterizing temporal distribution Du et al. (2021), we perform a distribution characterization by splitting the MTS data to the most diverse modes on the representation space (*i.e.*, the representation of all time-consecutive sequences/samples that encode variable dependencies and temporal dynamics), which is formulated as a constrained optimization problem:

$$\max_{0 < K \le K_0} \max_{n_1, \cdots, n_K} \frac{1}{K} \sum_{1 \le i \ne j \le K} \mathcal{D}(\mathcal{M}_i, \mathcal{M}_j)$$

$$\text{s.t. } \forall i, \Delta_1 < |\mathcal{M}_i| < \Delta_2; \sum_i |\mathcal{M}_i| = n, \tag{3}$$

where $\mathcal{D}(\cdot, \cdot)$ can be any distribution-related distance metric, $n$ is the number of training samples in a single regime, $\Delta1$, $\Delta2$ and $K_0$ are hyperparameters to avoid trivial partitions and over-splitting, $\mathcal{M}$ denotes the subset of representations that corresponds to contiguous samples. Specifically, we choose the Deep Correlation Alignment (CORAL) Sun & Saenko (2016) to measure the temporal distribution similarity, *i.e.*, $D(\cdot, \cdot) = \frac{1}{4q^2} \left\| C_{(\cdot)} - C_{(\cdot)} \right\|_F^2$, where $q$ is the number of dimensions for each hidden state, $C(\cdot)$ denotes the second-order statistics (covariance matrix). The detailed optimization procedure is explained in Appendix A.

After the most diverse modes are obtained, the distribution of a regime can be efficiently preserved by selecting a small number of the most representative samples of each mode. The selection algorithm is shown in Algorithm 1, where we select samples that minimize CORAL to ensure that the selected small number of samples are well aligned/matched to each mode of MTS. By iterating all modes, we update the memory buffer as the union of all selected sample sets.

### 3.4 SEQUENTIAL TRAINING AND TESTING WITH SKI-CL

We briefly introduce the pipeline of continual MTS forecasting, as shown in Figure 1. At the training phase of $i$-th regime, the training objectives $\mathcal{L}_{\text{regime-i}}$ contains the objective for the current training samples, denoted as $\mathcal{L}_{\text{current}}$ and that for the memory of previous $i$-1 regimes, denoted as $\mathcal{L}_{\text{memory}}$, where $\mathcal{L}_{\text{regime-i}} = \mathcal{L}_{\text{current}} + \alpha \mathcal{L}_{\text{memory}}$, with $\alpha$ being the weight of memory loss.

After training, the structural knowledge of the current regime is saved in the structural memory and the training samples are selected to enrich the MTS memory as aforementioned. At the testing phase, SKI-CL is able to maintain the forecasting performance on the queries of testing samples from all regimes up to the current one, and accordingly recover the learned dependency structures informed by the structural knowledge of each regime (as shown in Figure 1(right)). Unlike other graph structure learning methods for continual MTS forecasting, our method

---

**Algorithm 1** Representation-Matching Sample Selection

1: **Input:** Sample representation $H$, hyperparameters $\Delta_1$, $\Delta_2$, $K_0$, memory budget for a single regime $N_{\text{m}}$, the number of training samples in a single regime $n$
2: Split $H$ into modes $\mathcal{M}_1, \ldots, \mathcal{M}_K$ by optimizing (3)
3: Initialize an empty memory buffer $S$ for a regime
4: **for** $k \leftarrow 1$ to $K$ **do**
5:      $n_{\text{sample}} \leftarrow 0$; $n_{\text{select}} \leftarrow N_{\text{m}} \times \frac{|\mathcal{M}_k|}{n}$; $s \leftarrow \{\}$
6:      **while** $n_{\text{sample}} \leq n_{\text{select}}$ **do**
7:          $i_{\text{selected}} \leftarrow \min_i \mathcal{D}\left(H_{s \cup i}, H_{\mathcal{M}_k}\right)$     $\triangleright i \notin s$
8:          $s = s \cup i_{\text{selected}}$; $n_{\text{sample}} += 1$
9:      $S = S \cup s$
10: **Output:** The memory buffer $S$

---

is able to dynamically infer faithful dependency structures for existing and current regimes without accessing the memory buffer, which is more practical in real world applications.

## 4 EXPERIMENTS

### 4.1 EXPERIMENTAL SETUP

**Datasets** To evaluate the performance of SKI-CL on continual MTS forecasting, we conduct experiments on three public benchmark MTS datasets including the traffic (Traffic-CL), solar energy (Solar-CL), and human activity recognition (HAR-CL), as well as one synthetic dataset based on Non-repeating Random Walk  Denton (2005) that is used in Liu et al. (2022b) under continual learning setting. The statistics of these datasets are summarized in Table 1. For Traffic-CL and Solar-CL, the structural knowledge is the spatial proximity of the sensor/station. For HAR-CL, the partial structural knowledge is drawn from the domain-specific motion dynamics. For synthetic data, structural knowledge is the feature similarity of different variables. The edges from the structural knowledge are binary for Traffic-CL and HAR-CL, and continuous for Solar-CL and Synthetic-CL.

**Baselines** We compare SKI-CL with a number of dependency-modeling-based forecasting methods and commonly used continual learning methods to resolve the catastrophic forgetting issue in sequential training. The forecasting methods include statistical model VAR Lütkepohl (2005), ARIMA Box et al. (2015), and deep learning mod-

Table 1: Summary of Datasets

| Dataset | Traffic-CL | Solar-CL | HAR-CL | Synthetic-CL |
|---|---|---|---|---|
| # of nodes | 22 | 50 | 9 | 10 |
| # of all time steps | 106,848 | 52,560 | 600,576 | 24,000 |
| # of regimes | 7 | 5 | 4 | 4 |
| Regime | year | state | activity | adjacency |
| Structure avail. | Completed | Completed | Partial | Completed |

Table 2: Experiment Results for 12 Horizon Prediction. (Lower MAE and RMSE for AP mean better; When AP is comparable, lower MAE and RMSE for AF mean better. )

| Model | Traffic-CL | | | | Solar-CL | | | | HAR-CL ($\times 10^{-2}$) | | | | Synthetic-CL ($\times 10^{-2}$) | | | |
|---|---|---|---|---|---|---|---|---|---|---|---|---|---|---|---|---|
| | AP↓ | | AF | | AP↓ | | AF | | AP↓ | | AF | | AP↓ | | AF | |
| | MAE | RMSE | MAE | RMSE | MAE | RMSE | MAE | RMSE | MAE | RMSE | MAE | RMSE | MAE | RMSE | MAE | RMSE |
| **VAR**$_{seq}$ | 88.19 | 126.01 | 58.38 | 80.58 | 167.30 | 534.42 | 205.27 | 658.80 | 19.59 | 28.38 | 1.93 | 2.19 | 22.34 | 32.70 | 9.18 | 13.54 |
| **ARIMA**$_{seq}$ | 141.75 | 159.89 | 77.61 | 77.40 | 14.97 | 18.92 | 4.75 | 2.92 | 40.68 | 52.87 | 2.35 | 2.38 | 42.24 | 43.51 | 13.39 | 12.98 |
| **TCN**$_{seq}$ | 16.88 | 28.67 | 3.77 | 6.83 | 2.03 | 4.84 | 0.06 | 0.24 | 14.85 | 23.42 | 3.60 | 5.06 | 4.30 | 4.90 | 0.66 | 0.99 |
| **TCN**$_{mir}$ | 15.70 | 26.53 | 1.70 | 3.22 | 1.99 | 4.79 | 0.10 | 0.19 | 13.91 | 22.15 | 2.64 | 2.93 | 3.79 | 4.63 | 0.46 | 0.73 |
| **TCN**$_{herd}$ | 15.55 | 26.21 | 1.49 | 2.81 | 2.01 | 4.82 | 0.13 | 0.23 | 13.87 | 22.08 | 2.05 | 2.88 | 3.72 | 4.61 | 0.35 | 0.67 |
| **TCN**$_{er}$ | 15.51 | 26.23 | 1.46 | 2.80 | 1.98 | 4.73 | -0.05 | 0.02 | 13.66 | 21.78 | 1.82 | 2.59 | 3.29 | 4.30 | 0.34 | 0.61 |
| **TCN**$_{der++}$ | 15.46 | 25.68 | 1.33 | 2.49 | 1.95 | 4.69 | -0.07 | -0.02 | 13.56 | 21.55 | 1.69 | 2.28 | **3.00** | **4.00** | 0.28 | 0.32 |
| **ESG**$_{seq}$ | 18.77 | 30.02 | 6.46 | 10.07 | 2.80 | 5.77 | 1.28 | 1.74 | 17.63 | 26.84 | 7.28 | 9.41 | 8.98 | 14.19 | 1.32 | 1.98 |
| **ESG**$_{mir}$ | 18.24 | 29.83 | 5.02 | 8.25 | 2.03 | 4.83 | 0.25 | 0.49 | 17.25 | 26.63 | 4.01 | 5.21 | 8.95 | 13.91 | 1.21 | 1.81 |
| **ESG**$_{herd}$ | 17.49 | 28.64 | 4.82 | 7.45 | 1.92 | 4.72 | 0.13 | 0.53 | 17.22 | 26.59 | 3.99 | 5.13 | 8.94 | 13.88 | 1.11 | 1.74 |
| **ESG**$_{er}$ | 16.40 | 27.50 | 3.05 | 5.34 | 2.01 | 4.82 | 0.24 | 0.44 | 17.15 | 25.84 | 4.63 | 5.33 | 8.84 | 13.86 | 1.21 | 1.62 |
| **ESG**$_{der++}$ | 17.40 | 29.21 | 4.01 | 6.97 | 1.91 | 4.57 | 0.09 | 0.21 | 16.20 | 24.32 | 5.18 | 6.00 | 8.81 | 13.77 | 1.02 | 1.42 |
| **GTS**$_{seq}$ | 17.26 | 29.11 | 2.33 | 3.48 | 2.19 | 5.20 | 0.27 | 0.59 | 16.44 | 25.41 | 3.68 | 5.10 | 6.51 | 8.89 | 1.88 | 3.39 |
| **GTS**$_{mir}$ | 17.17 | 29.08 | 2.13 | 3.31 | 2.15 | 5.16 | 0.14 | 0.68 | 15.83 | 24.85 | 3.27 | 4.91 | 6.44 | 8.57 | 1.31 | 2.86 |
| **GTS**$_{herd}$ | 17.00 | 29.01 | 2.17 | 2.98 | 2.11 | 5.06 | 0.13 | 0.30 | 15.65 | 24.33 | 1.99 | 2.86 | 6.34 | 8.23 | 1.18 | 1.81 |
| **GTS**$_{er}$ | 15.83 | 26.20 | 1.12 | 2.52 | 2.01 | 4.75 | 0.12 | 0.05 | 15.06 | 23.52 | 2.00 | 2.73 | 5.59 | 6.32 | 0.44 | 0.69 |
| **GTS**$_{der++}$ | 15.84 | 26.05 | 1.15 | 2.33 | 1.94 | 4.57 | -0.25 | -0.19 | 14.80 | 23.01 | 1.52 | 1.88 | 5.43 | 6.67 | 0.23 | 0.30 |
| **MTGNN**$_{seq}$ | 19.88 | 32.94 | 7.83 | 12.68 | 2.12 | 4.75 | 0.38 | 0.44 | 14.86 | 22.58 | 2.59 | 3.61 | 10.26 | 14.92 | 1.16 | 1.81 |
| **MTGNN**$_{mir}$ | 18.01 | 31.84 | 5.03 | 8.97 | 2.00 | 4.73 | 0.21 | 0.40 | 14.59 | 22.52 | 2.24 | 3.53 | 8.92 | 12.91 | 1.07 | 1.33 |
| **MTGNN**$_{herd}$ | 17.93 | 30.70 | 4.90 | 8.40 | 1.89 | 4.68 | 0.13 | 0.35 | 14.09 | 22.50 | 1.13 | 1.62 | 8.11 | 12.88 | 1.03 | 1.27 |
| **MTGNN**$_{er}$ | 15.79 | 26.52 | 2.76 | 4.87 | 1.94 | 4.62 | 0.14 | 0.25 | 13.59 | 21.85 | 1.91 | 2.79 | 8.70 | 13.69 | 0.61 | 1.21 |
| **MTGNN**$_{der++}$ | 15.40 | 25.99 | 2.22 | 4.10 | 1.90 | 4.57 | 0.06 | 0.14 | 13.57 | 21.75 | 1.63 | 2.40 | 8.63 | 13.51 | 0.50 | 0.92 |
| **Autoformer**$_{seq}$ | 23.92 | 40.40 | 2.58 | 4.42 | 6.04 | 11.41 | 0.53 | 1.56 | 19.97 | 28.76 | 2.81 | 3.08 | 5.15 | 6.87 | 0.53 | 0.46 |
| **Autoformer**$_{mir}$ | 23.85 | 40.13 | 2.21 | 4.07 | 5.95 | 11.18 | 0.87 | 1.14 | 18.57 | 27.93 | 1.45 | 2.88 | 5.12 | 6.73 | 0.32 | 0.37 |
| **Autoformer**$_{herd}$ | 23.90 | 40.21 | 2.43 | 4.29 | 5.91 | 11.12 | 0.90 | 1.20 | 18.78 | 28.03 | 1.42 | 2.66 | 5.00 | 6.65 | 0.23 | 0.24 |
| **Autoformer**$_{er}$ | 23.26 | 38.98 | 1.28 | 2.00 | 5.83 | 10.89 | 0.49 | 1.32 | 18.42 | 27.05 | 1.29 | 2.01 | 5.02 | 6.66 | 0.24 | 0.25 |
| **Autoformer**$_{der++}$ | 22.11 | 37.34 | 1.08 | 1.92 | 5.34 | 9.29 | 0.21 | 1.14 | 18.12 | 26.91 | 1.12 | 1.85 | 4.97 | 6.61 | 0.18 | 0.23 |
| **PatchTST**$_{seq}$ | 19.11 | 32.50 | 2.34 | 2.97 | 2.64 | 5.32 | 0.72 | 0.43 | 17.91 | 27.13 | 7.18 | 6.88 | 4.85 | 5.93 | 1.59 | 1.78 |
| **PatchTST**$_{mir}$ | 19.04 | 32.23 | 2.28 | 2.79 | 2.61 | 5.30 | 0.70 | 0.40 | 17.82 | 26.89 | 6.82 | 4.81 | 4.83 | 5.86 | 1.55 | 1.72 |
| **PatchTST**$_{herd}$ | 18.96 | 32.10 | 2.21 | 2.67 | 2.60 | 5.30 | 0.68 | 0.35 | 17.73 | 26.84 | 6.62 | 4.79 | 4.80 | 5.79 | 1.43 | 1.68 |
| **PatchTST**$_{er}$ | 18.77 | 31.50 | 1.98 | 2.01 | 2.57 | 5.27 | 0.47 | 0.30 | 17.57 | 26.40 | 6.02 | 4.69 | 4.72 | 5.26 | 1.03 | 1.54 |
| **PatchTST**$_{der++}$ | 18.53 | 31.34 | 1.75 | 1.98 | 2.53 | 5.17 | 0.43 | 0.28 | 17.12 | 26.13 | 5.79 | 4.32 | 4.64 | 5.13 | 0.83 | 0.88 |
| **DLinear**$_{seq}$ | 19.69 | 32.75 | 2.91 | 2.83 | 3.47 | 6.56 | 1.17 | 1.12 | 17.32 | 26.31 | 2.71 | 3.43 | 4.81 | 5.81 | 1.64 | 1.57 |
| **DLinear**$_{mir}$ | 19.37 | 32.25 | 2.17 | 2.59 | 3.45 | 6.51 | 1.02 | 1.01 | 16.87 | 26.12 | 2.67 | 3.01 | 4.79 | 5.73 | 1.47 | 1.40 |
| **DLinear**$_{herd}$ | 19.53 | 32.40 | 2.25 | 2.68 | 3.41 | 6.50 | 1.03 | 1.00 | 16.83 | 25.81 | 2.57 | 2.91 | 4.77 | 5.70 | 1.59 | 1.46 |
| **DLinear**$_{er}$ | 19.19 | 32.30 | 1.73 | 2.14 | 3.37 | 6.43 | 0.93 | 0.98 | 16.71 | 25.75 | 2.13 | 2.85 | 4.74 | 5.20 | 1.23 | 1.43 |
| **DLinear**$_{der++}$ | 19.02 | 31.97 | 1.75 | 1.93 | 3.25 | 6.37 | 0.83 | 0.79 | 16.58 | 25.47 | 1.92 | 2.77 | 4.21 | 4.88 | 1.12 | 1.13 |
| **TimesNet**$_{seq}$ | 17.77 | 29.91 | 3.13 | 6.93 | 3.92 | 7.18 | 1.46 | 2.51 | 18.38 | 27.61 | 4.33 | 5.15 | 5.18 | 6.13 | 1.72 | 2.03 |
| **TimesNet**$_{mir}$ | 17.53 | 29.61 | 2.44 | 5.32 | 3.77 | 7.15 | 1.22 | 1.57 | 18.27 | 27.59 | 4.01 | 5.08 | 5.12 | 6.05 | 1.69 | 1.97 |
| **TimesNet**$_{herd}$ | 17.38 | 29.53 | 2.56 | 5.83 | 3.83 | 7.10 | 1.03 | 1.44 | 18.01 | 27.53 | 3.46 | 5.03 | 5.10 | 6.03 | 1.68 | 1.95 |
| **TimesNet**$_{er}$ | 17.25 | 29.33 | 1.97 | 4.19 | 3.55 | 7.02 | 0.42 | 0.91 | 17.84 | 27.07 | 3.28 | 4.01 | 4.93 | 5.90 | 1.42 | 1.90 |
| **TimesNet**$_{der++}$ | 17.13 | 29.28 | 1.56 | 4.02 | 3.45 | 6.55 | 0.37 | 0.90 | 17.73 | 26.86 | 3.11 | 3.87 | 4.81 | 5.88 | 1.32 | 1.78 |
| **SKI-CL**$_{seq}$ | 17.30 | 29.38 | 4.38 | 7.80 | 2.02 | 4.73 | 0.30 | 0.50 | 14.73 | 23.31 | 3.91 | 5.07 | 4.70 | 5.85 | 1.97 | 3.46 |
| **SKI-CL**$_{mir}$ | 15.77 | 26.32 | 1.95 | 3.47 | 1.98 | 4.69 | 0.35 | 0.56 | 13.65 | 21.77 | 2.61 | 3.82 | 4.53 | 5.01 | 1.67 | 2.21 |
| **SKI-CL**$_{herd}$ | 15.45 | 25.73 | 1.82 | 3.28 | 2.00 | 4.70 | 0.33 | 0.52 | 13.71 | 21.82 | 2.53 | 3.67 | 4.44 | 4.82 | 1.23 | 2.02 |
| **SKI-CL**$_{er}$ | 15.43 | 25.60 | 1.69 | 2.92 | 1.95 | 4.67 | 0.11 | 0.23 | 13.58 | 21.57 | 1.82 | 2.50 | 3.39 | 4.52 | 0.30 | 0.38 |
| **SKI-CL**$_{der++}$ | 15.39 | 25.57 | 1.63 | 2.87 | 1.91 | 4.60 | 0.10 | 0.21 | 13.50 | 21.47 | 1.74 | 2.42 | 3.33 | 4.43 | 0.28 | 0.30 |
| **SKI-CL** | **15.24** | **25.33** | 1.50 | 2.71 | **1.75** | **4.46** | 0.09 | 0.06 | **13.41** | **21.30** | 1.64 | 2.08 | 3.24 | 4.24 | 0.15 | 0.23 |

els including TCN Bai et al. (2018), LSTNet Lai et al. (2018), STGCN Yu et al. (2018), MTGNN Wu et al. (2020), AGCRN Bai et al. (2020), GTS Shang & Chen (2021) and ESG Ye et al. (2022), StemGNN Cao et al. (2020), Autoformer Wu et al. (2021), PatchTST Nie et al. (2022),Dlinear Zeng et al. (2023),TimesNet Wu et al. (2022) where STGCN and GTS use structural knowledge and ESG learns a dynamic graph in MTS modeling. All forecasting methods are first evaluated on sequential training without any countermeasures (denoted as seq). The continual learning methods employ the memory-replay-based methods including the herding method (denoted as herd) Rebuffi et al. (2016), the randomly replay training samples (denoted as er), a DER++ method Buzzega et al. (2020) that enforces a $L_2$ knowledge distillation loss on the previous logits (denoted as der++) and a MIR method that selects samples with highest forgetting for experience replay (denoted as mir) . For the proposed SKI-CL, we also evaluate our proposed representation-matching sample selection scheme.

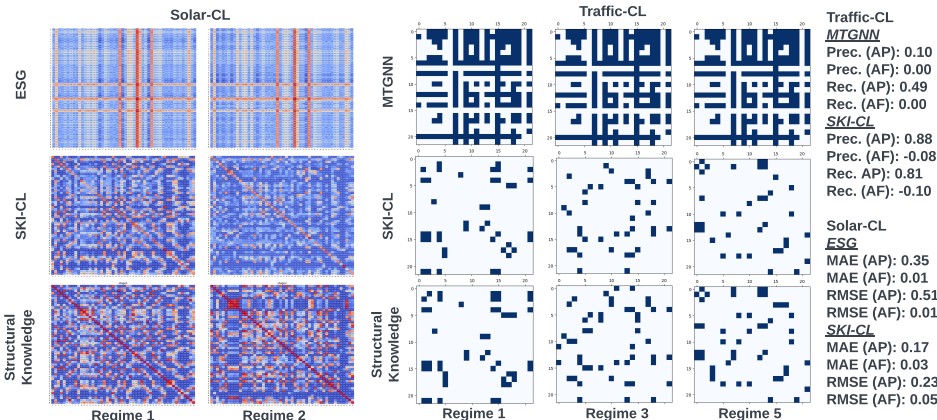

Figure 3: The Structure Visualizations on Traffic-CL and Solar-CL datasets.

**Evaluation Metrics** We adopt two metrics based on Mean Absolute Error (MAE) and Root Mean Squared Error (RMSE) to evaluate the performance on continual MTS forecasting, *i.e.*, the Average Performance (AP) and Average Forgetting (AF) Lopez-Paz & Ranzato (2017); Zhang et al. (2022a). The AP at $i$-th regime is defined as $AP = \sum_{j=1}^{i} P_{i,j}/i$ for $i \geq 1$, where $P_{i,j}$ denotes the performance on regime $j$ after the model has been sequentially trained from stage 1 to $i$. Similarly, the Average Forgetting is defined as $AF = \sum_{j=1}^{i-1} (P_{i,j} - P_{j,j})/(i - 1)$ for $i \geq 2$. Besides the forecasting performance, we also evaluate AP and AF on the learned dependency structures, where the average precision (Prec.) and average recall (Rec.) are used for a binary graph, MAE and RMSE are used for a continuous graph. More details of datasets, baselines, and evaluations are provided in Appendix C.

## 4.2 PERFORMANCE EVALUATION FOR CONTINUAL MTS FORECASTING

In this paper, we focus on a multi-horizon continual forecasting task. Table 2 summarizes the experiment results of selected baselines (full experiment results are shown in Appendix Table 7) and our proposed SKI-CL method with variants for 12 horizon predictions (the corresponding standard deviations and results for different horizons are provided in Table 9 and 8, Appendix F respectively.). Based on the results, we show that statistical methods, such as VAR and ARIMA, cannot perform well under continual learning settings and all other forecasting methods suffer from obvious performance degradation (AF) in sequential training, indicating the existence of catastrophic forgetting phenomenons when regime shifts. In addition, the memory-replay-based methods generally alleviate the forgetting issues with better APs (lower RMSE and MAE) and smaller relative AFs. SKI-CL and its variants (SKI-CL$_{er}$ and SKI-CL$_{der++}$) consistently achieve the best or the second-best APs, showing advantages over other models equipped with memory-replay-based methods (*e.g.*, MTGNN$_{der++}$, TCN$_{er}$, GTS$_{der++}$) since SKI-CL explicit imposes a regularization to enforce the consistency between the learned dynamic graphs and the structure knowledge. These observations demonstrate that learning a dynamic structure is beneficial for nonstationary MTS modeling, and the structural knowledge helps to characterize the general variable behaviors in each regime. Even partially observed structural knowledge can serve as a valid reference to learn the dependency structures. Finally, the proposed SKI-CL achieves better APs or less AFs when the AP metrics are comparable, suggesting the superiority of our representation-matching scheme that maximizes the coverage of dependencies and temporal dynamics.

## 4.3 PRESERVING FAITHFUL DEPENDENCY STRUCTURES

We also evaluate how the baselines and our method preserve the learned structures that are highly correlated to the structural knowledge. Specifically, we compare SKI-CL with GTS on Traffic-CL dataset, and ESG on Solar-CL dataset to investigate the binary edges and continuous edges, respectively. The results of the learned structures and structural knowledge are shown in Figure 3, where the average performance and average forgetting are also annotated. It is clear that SKI-CL is able to alleviate the forgetting of the learned structures on both datasets at the testing phase, while the baselines fail to model the nonstationarity of MTS in different regimes. This observation also

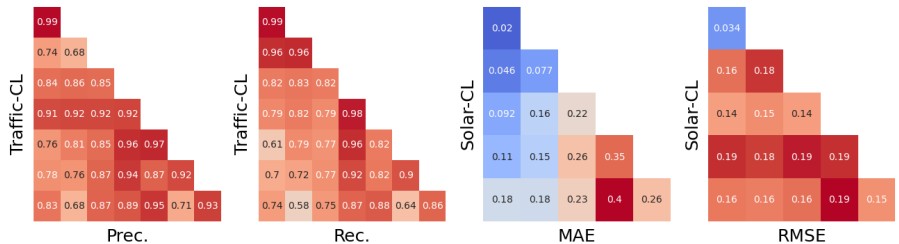

Figure 4: The Structure Similarity Matrices on Traffic-CL and Solar-CL datasets.

Table 3: Analysis of $\lambda$ on Traffic-CL

| $\lambda$ | Forecasting Performance | | | | Structure Similarity | | | |
|---|---|---|---|---|---|---|---|---|
| | AP | | AF | | AP | | AF | |
| | MAE | RMSE | MAE | RMSE | Prec. | Recall | Prec. | Recall |
| 0.01 | 15.42 | 26.08 | 2.30 | 4.13 | 0.52 | 0.46 | -0.52 | -0.57 |
| 0.1 | 15.39 | 25.97 | 2.12 | 4.07 | 0.84 | 0.75 | -0.11 | -0.19 |
| 0.5 | 15.33 | 25.68 | 1.68 | 3.07 | 0.85 | 0.78 | -0.08 | -0.11 |
| 1 | 15.22 | 25.31 | 1.53 | 2.74 | 0.84 | 0.80 | -0.10 | -0.12 |
| 2 | 15.27 | 25.51 | 1.70 | 3.10 | 0.85 | 0.79 | -0.08 | -0.14 |
| 5 | 15.31 | 25.63 | 2.04 | 3.56 | 0.83 | 0.72 | -0.16 | -0.24 |

Table 4: Analysis of Memory Budget on Traffic-CL

| Ratio | Forecasting Performance | | | | Structure Similarity | | | |
|---|---|---|---|---|---|---|---|---|
| | AP | | AF | | AP | | AF | |
| | MAE | RMSE | MAE | RMSE | Prec. | Recall | Prec. | Recall |
| 0.01 | 15.22 | 25.31 | 1.53 | 2.74 | 0.84 | 0.80 | -0.10 | -0.12 |
| 0.05 | 14.49 | 24.35 | 1.16 | 2.03 | 0.86 | 0.79 | -0.10 | -0.13 |
| 0.1 | 14.44 | 24.03 | 0.77 | 1.23 | 0.84 | 0.78 | -0.07 | -0.13 |
| 0.2 | 14.25 | 23.74 | 0.85 | 1.34 | 0.89 | 0.80 | -0.06 | -0.14 |
| 0.5 | 14.18 | 23.06 | 0.55 | 0.79 | 0.90 | 0.81 | -0.05 | -0.09 |

suggests the importance of dynamic structure learning and the incorporation of structural knowledge, to maintain a faithful structure at each regime. We further visualize the similarity matrices between the structures inferred by SKI-CL and structural knowledge, as shown in Figure 4. Each cell corresponds to the aforementioned $P_{i,j}$, *i.e.*, the performance on regime $j$ after the model has been sequentially trained from stage 1 to $i$, where $i$ and $j$ denote the row number and column number, respectively. We observe that the inferred structures at the testing phases of each regime still reveal similarities to the structural knowledge, by comparing the values of each row with the diagonal ones.

We emphasize that we don't intend to use structural knowledge as a ground truth. We have demonstrated that exploiting structural knowledge helps to reduce the performance degradation between consecutive regimes. Besides, it is beneficial to have the model aligned with the structural knowledge for a better interpretation of each regime. We provide a case study on the Synthetic-CL dataset as a demonstration in Figure 9, Appendix G . More visualization results are provided in Appendix D.

### 4.4 HYPERPARAMETER ANALYSIS

We perform experiments on the Traffic-CL dataset to validate the effectiveness and sensitivity of two key hyperparameters in SKI-CL, the weight of structure regularizer $\lambda$ (1 by default) and the memory budget (sampling ratio) at each regime (0.01 by default). As shown in Table 3, within a small range, the model is relatively stable in terms of $\lambda$, resulting in similar average performance and average forgetting on forecasting performance as well as the inferred structure similarity. We also study the model performance when the memory budget varies in Table 4. We can observe that a larger memory budget can achieve better forecasting performance and less forgetting. Meanwhile, the structure similarity is relatively stable regarding the memory budget. The exploration of other hyperparameter settings is provided in Appendix E.

## 5 CONCLUSION

In this paper, we propose a novel Structural Knowledge Informed Continual Learning (SKI-CL) framework to perform continual MTS forecasting, which exploits the structural knowledge to characterize the dynamic variable dependencies within each regime. We first design a dynamic graph learning model for fine-grained dynamic MTS modeling and impose a regularization scheme that ensures the consistency between the learned structure and the variable dependencies revealed by structural knowledge. We further alleviate the catastrophic forgetting by developing a representation-matching memory replay scheme, which maximizes the temporal coverage of MTS data to efficiently preserve each regime's underlying temporal dynamics and dependency structure. Experiments on one synthetic dataset and three real-world benchmark datasets demonstrate the effectiveness and advantages of the proposed SKI-CL on continual MTS forecasting tasks.

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

## A    REPRESENTATION-BASED DISTRIBUTION CHARACTERIZATION

### A.1    OPTIMIZING THE OBJECTIVE FOR COVERAGE MAXIMIZATION

Inspired by the principle of maximum entropy, we perform a distribution characterization by splitting the MTS data into the most diverse modes on the representation space, which incorporates variable dependence and temporal information. The distribution characterization splits the hidden representation by solving a constrained optimization problem whose objective is formulated as:

$$\max_{0 < K \leq K_0} \max_{n_1, \cdots, n_K} \frac{1}{K} \sum_{1 \leq i \neq j \leq K} \mathcal{D}\left(\mathcal{M}_i, \mathcal{M}_j\right)$$
$$\text{s.t. } \forall i, \Delta_1 < |\mathcal{M}_i| < \Delta_2; \sum_i |\mathcal{M}_i| = n, \tag{4}$$

where $\mathcal{D}(\cdot, \cdot)$ can be any distribution-related distance metric, $n$ is the number of training samples in a single regime, $\Delta 1$, $\Delta 2$ and $K_0$ are hyperparameters to avoid trivial partitions and over-splitting, $\mathcal{M}$ denotes the subset of representations that correspond to contiguous samples. The optimization problem can be computationally intractable and the closed-form solution may not exist. We adopt the greedy algorithm proposed by Du et al. (2021). First, we obtain the ordered representation set from the trained model for single regime data. Then, for efficient computation, we evenly split the representation into $N$ parts and randomly search the value of $K$ in $\{2, 3, 4 \cdots N - 1\}$ Denote the start and the end index of the representation by A and B respectively. We first consider $K = 2$ by choosing the first splitting point C from all candidate splitting points via maximizing the distance metric $\mathcal{D}(\mathcal{M}_{AC}, \mathcal{M}_{CB})$. After C is determined, we then consider K = 3 and use the same strategy to select another point D. A similar strategy is applied until the number of representation modes is obtained.

## B    RELATION TO OTHER RESEARCH TOPICS

Our method characterizes dynamic variable dependencies of MTS data within each regime to perform continual MTS forecasting tasks. The dynamic variable dependencies modeling and continual adaptation nature relate our method to two research topics, *i.e.*, the dynamic graph learning and learning with temporal drift of MTS. Dynamic graph learning focuses on adapting to the changes in the explicit graph structure and possibly features over time, and leverage it to facilitate downstream tasks (*e.g.,* node classification Liu et al. (2022a), link prediction Wang et al. (2021), community detection Park et al. (2022)). In contrast, our proposed continual multivariate time series forecasting method needs to discover underlying dependencies structure of variables over different regimes (stages), by capturing the temporal dynamics of MTS data. On the other hand, temporal drift of MTS represents the phenomenon where the data distribution of the target MTS changes over time in unforeseen ways, where the approaches are often more reactive and focused on adapting to changes in data distribution for a specific task Du et al. (2021); Kim et al. (2021); Lee et al. (2022). Based on the notion, it is important to emphasize the continual adaptation nature of continual learning setting regarding the temporal drifts, where the catastrophic forgetting happens due to the shifts over multiple regimes. That being said, continual MTS forecasters need to handle a variety of regimes, not just adapt to changes in data distribution for a single one, which is to some degree more realistic and challenging. Temporal drift represents a specific challenge within this broader spectrum of adaptation.

Moreover, while the aforementioned topics as well as our study deal with the evolving data, the learning objectives can be very different. For the temporal drift methods, the objective is to adapt the forecaster to these new MTS patterns by effectively detecting, responding to, and learning from these changes. Similarly, the focus of dynamic graph learning is to adapt its model to cope with the structural and possible feature changes. However, the main focus this paper is to prevent the model from forgetting previously learned knowledge (coupled temporal dynamics and variable dependencies in our scope) when adapting to new MTS distributions.

# C   DATASETS, BASELINES AND EVALUATIONS FOR CONTINUAL MTS FORECASTING

## C.1   DATASETS

**Traffic-CL** Following the fashion in PEMSD3-Stream Chen et al., we construct the Traffic-CL dataset based on the PEMSD3 benchmark for continual MTS forecasting tasks. The PEMSD3 benchmark data was collected by the Performance Measurement System in California Chen et al. (2001) in real-time by every 30 seconds and further aggregated to 5-min granularity. The PEMSD3-Stream dataset contains traffic data from 2011 to 2017. Specifically, data within a month period from July 10th to August 9th from every year was selected, where the traffic network keeps expanding from year to year. The adjacency matrix for $\tau$-th year is extracted by applying a Gaussian kernel to the spatial all pairwise distances between two traffic sensors, as shown in Equation 5.

$$A_\tau[i,j] = \begin{cases} \exp\left(-\frac{d_{ij}^2}{\sigma_d^2}\right), i \neq j \quad \text{and} \quad d_{ij} \leq \epsilon \\ 0, \text{ otherwise} \end{cases} \tag{5}$$

where $d_{ij}$ denotes the spatial distance between sensor i and j. $\sigma_d$ and $\epsilon$ are the standard deviation and a predefined threshold (controlling the sparsity of the adjacency matrix, set as 1), respectively.

Based on the constructed PEMSD3-Stream dataset, we make the following modifications to further simulate distinct regimes in the setting of continual forecasting. For each year, we rank and select the top 100 traffic sensors with the largest node degrees, based on which we randomly select 22 sensors as a set representing a part of the major traffic. Next, we transform the continuous adjacency weights to binary ones by a threshold, and use it as the structural knowledge. As such, each regime is represented by a different portion of a temporally expanding traffic network from different years, with a binarized structural prior and MTS data defined accordingly. The input horizon for the Traffic-CL dataset is 12.

**Solar-CL** We build our continual MTS forecasting dataset based on the database for NREL's Solar Power Data for Integration Studies[1], which contains 5-minute solar power data for near 6,000 simulated photovoltaic power plants in the United States for the year 2006. Note that the data in Alabama with a 10-minute granularity is also known as the commonly used Solar dataset in many existing MTS studies Wu et al. (2020); Lai et al. (2018); Cao et al. (2020); Liu et al. (2022b).

We construct different regimes by states with different average annual sunlight levels (measured by $kJ/m^2$). Based on the statistics[2] and the aggregated MTS data at 10-minute, we select five states, Massachusetts/MA ($3944 \, kJ/m^2$) - Arizona/AZ ($5755 \, kJ/m^2$) - North Carolina/NC ($4456 \, kJ/m^2$) - Texas/TX ($5137 \, kJ/m^2$) - Washington/WA ($3467 \, kJ/m^2$) as five regimes representing different sunlight patterns in different spatial locations. For each state, 50 photovoltaic power plants are randomly selected, where the spatial information is also used as a valid structural prior, as plants that are geographically close share similar weather and sunlight conditions at a local level. Specially, we first extract the longitude and latitude for each plant and calculate the pairwise geographic distances among all plant pairs. Based on all pairwise distances, we generate the adjacency matrix by applying a Gaussian kernel in Equation 3, where we set $\epsilon = \infty$, indicating a fully connected continuous graph. The input horizon for the Solar-CL dataset is 24.

**HAR-CL** We leverage the class boundaries in MTS classification data to construct different regimes in forecasting tasks. Specifically, we build our continual forecasting dataset based on a commonly used MTS Classification benchmark, the Human Activity Recognition (HAR) dataset Anguita et al. (2013b;a) in the UCI database[3], where the data is collected from a group of 30 volunteers from 19-48 years, wearing a Samsung smartphone on the waist and performing six activities of daily living (Walking, Walking upstairs, Walking downstairs, Standing, Sitting, Lying). Each MTS sequence contains 128 time steps and 9 variables that are recorded based on the accelerometer and gyroscope, including the linear accelerations, the angular velocities, and total accelerations along the X-Y-Z axis. The detailed setup for data collection can be found in Figure 1 of Anguita et al. (2013a).

---

[1] https://www.nrel.gov/grid/solar-power-data.html
[2] https://worldpopulationreview.com/state-rankings/sunniest-states
[3] https://archive.ics.uci.edu/ml/datasets/Human+Activity+Recognition+Using+Smartphones

We notice that different human activities naturally form distinct regimes with unique temporal dynamics, where deep learning methods easily achieve over 94% classification accuracy Wang et al. (2019). Motivated by this fact, we construct the continual HAR forecasting dataset considering the following details. Firstly, we select Walking, Walking upstairs, Walking downstairs, and Lying as four different regimes in our task. Secondly, each regime contains over one thousand sequences that are not always temporally connected due to different volunteers. As such, we iterate all sequences and construct the input-output MTS windows in each 128-step sequence, after which all windows are stacked as the training/validation/testing data in one regime. Secondly, we try to build the structural knowledge for each regime based on the domain knowledge. We've already known that linear accelerations are highly correlated to total accelerations along each axis, regardless of the activities, but the dependencies among other variables are not very clear. Therefore, we examine the Pearson correlations of all training sequences for each regime, where the mean of correlation matrices demonstrate distinct patterns, and validate the strong correlations between linear and total accelerations. However, the standard deviations are only small at the diagonal and the aforementioned entries, suggesting a varying and uncertain structure for other variable pairs. To this end, we check a small (15-th) percentile of each entry in the absolute correlation matrix, and apply a threshold to get a binary mask representing the variable dependencies that we are confident of. Note that we only regularize the learned structures to be consistent with the partially observed structural knowledge at the masked entries. As such, we simulate the structured scenario when we are not aware of the completed structural knowledge of MTS. The input for the HAR-CL dataset has 12 steps.

**Synthetic-CL** Lastly, we generate the synthetic data based on Non-repeating Random Walk Denton (2005), which is used in Liu et al. (2022b) for the evaluation of the learned graph structure in a single regime. Next, we introduce how to generate the MTS data for continual forecasting tasks.

Firstly, we describe how to generate the MTS data step by step. At time step $t$, given a dynamic weighted adjacency matrix $\mathbf{W}^{(t)}$, $\mathbf{X}^{(t)} = \mathcal{N}\left(\mathbf{W}^{(t-1)}\mathbf{X}^{(t-1)}, \sigma\right) \in \mathbb{R}^{N \times 1}$, where $\mathcal{N}$ denotes the Gaussian distribution, $\sigma \in \mathbb{R}$ controls the variance, $N$ denotes the number of variables, and $\mathbf{X}^{(0)}$ is randomly initialized from the set $[-1, -0.5, 0.5, 1]$. Secondly, we describe how the $\mathbf{W}^{(t)}$ is generated and how to construct different regimes based on $\mathbf{W}^{(\cdot)}$. Assuming there are $L$ total time steps, we define $\mathbf{W}^{(t)}$ as one of $S$ constant matrices $\left(\mathbf{G}^{(1)}, \ldots, \mathbf{G}^{(S)}\right)$, where $\mathbf{G}^{(\cdot)}$ is a Laplacian of sparsified random adjacency matrix with sparsity $\delta$, spanning $\lfloor L/S \rfloor$ time steps ($\lfloor \cdot \rfloor$ denotes the floor function). As such, each regime is represented by MTS data with time steps from $(i-1) \times \lfloor L/S \rfloor$ to $i \times \lfloor L/S \rfloor$, with the corresponding weighted adjacency matrix $\mathbf{W}^{(t)} = \mathbf{G}^{(i)}$. Specifically, we set the number of variable $N = 10$, the total time steps $L = 24,000$, the number of regimes $S = 4$, the standard deviation of noise $\sigma = 0.01$, and the matrix sparsity $\delta = 0.1$.

There are two main differences between the synthetic setting of Liu et al. (2022b) and our work despite the setting of continual learning. Firstly, we don't reinitialize the value of each variable when the dynamic weighted adjacency matrix transits to a different one in order to preserve the temporal continuity of MTS data across different regimes. Secondly, the evaluation of graph structure learning is also different due to the forecasting setting. In this particular synthetic dataset, the dynamic weighted adjacency matrix $\mathbf{W}^{(t)}$ describes the data generation process at the single-step level, which can be treated as the ground truth if the non-linear part of $\mathbf{W}^{(t)}$ in the model learns an identity mapping with Gaussian noise. As the graph learned in Liu et al. (2022b) is under the setting of single-step prediction, the $\mathbf{W}^{(t)}$ itself is a reasonable reference for evaluation. In our cases of performing multi-horizon forecasting, the matrix $\mathbf{W}^{(t)}$ raised to a higher power can also demonstrate how the dynamic is propagated in a sequence. In our exploration, we don't assume $\mathbf{W}^{(t)}$ is explicitly given as the structural prior. Instead, we exploit the Pearson correlation of the generated MTS data and formulate a binary structural prior based on strong absolute correlations, where we will examine if the learned graph structure is able to reveal the variable interactions in $\mathbf{W}^{(t)}$. The input horizon for the Synthetic-CL dataset is 12.

## C.2 BASELINES

In this part, we introduce the state-of-the-art baseline methods evaluated in our paper.

- TCN Bai et al. (2018): Temporal convolution networks (TCN) models the temporal causality using causal convolution and do not involve structural dependence modeling.

- LSTNet Lai et al. (2018): LSTNet leverages the Convolution Neural Network (CNN) with a kernel spanning the variable dimension to extract short-term local variable dependencies, and the Recurrent Neural Network (RNN) to discover long-term patterns based on the extracted dependency patterns for MTS forecasting.

- STGCN Yu et al. (2018): STGCN jointly captures the spatial-temporal patterns by stacking spatial graph convolution layers that perform graph convolution using continuous structural knowledge and temporal-gated convolution layers that capture temporal dynamics based on the yielded spatial representations.

- MTGNN Wu et al. (2020): MTGNN learns a parameterized graph with top-k connections for each node, and performs mix-hop propagation for graph convolution and dilated inception for temporal convolution. The parameterized graph is purely optimized based on the forecasting objective. At the testing stage, the inferred graph is static due to the fixed parameters.

- AGCRN Bai et al. (2020): AGCRN models the dependencies graph structure as a product of trainable node embedding and performs graph convolution in the recurrent convolution layer for MTS forecasting. The node embedding and yielded graph are purely optimized based on the forecasting objective. At the testing stage, the inferred graph is static due to the fixed parameters.

- GTS Shang & Chen (2021): GTS infers steady node representations and global node relations from entire training MTS data. The learned dependency structure is used in Diffusion Convolution Recurrent Neural Networks (DCRNN) for MTS forecasting. The parameterized graph is optimized based on the forecasting objective as well as the regularization based on binary structure priors. At the testing stage, the graph is sampled from learned binary edge distributions.

- ESG Ye et al. (2022): ESG learns evolving and scale-specific node relations from features extracted from MTS data. A series of dynamic graphs representing dynamic correlations are utilized in sequential graph convolution and temporal convolution. The dynamic graphs are learned via the optimization of feature extraction layers based on the forecasting objective. At the testing stage, the graphs are dynamics inferred based on each MTS input window.

- StemGNN Cao et al. (2020): The Spectral Temporal Graph Neural Network (StemGNN) is a Graph-based multivariate time-series forecasting model, which jointly learns temporal dependencies and inter-series correlations in the spectral domain, by combining Graph Fourier Transform (GFT) and Discrete Fourier Transform (DFT).

- Autoformer Wu et al. (2021): Autoformer is a Transformer-based model using decomposition architecture with an Auto-Correlation mechanism to capture cross-time dependency for forecasting.

- PatchTST Nie et al. (2022): PatchTST model uses channel-independent and patch techniques to tokenize input time series and perform time series forecasting by utilizing the vanilla Transformer encoders.

- Dlinear Zeng et al. (2023): DLinear adopts trend-seasonal components decomposition techniques for time series data and applies MLP-based architectures for time series forecasting.

- TimesNet Wu et al. (2022): TimesNet model leverages intricate temporal patterns by exploring the multi-periodicity of time series and capture the temporal 2D-variations in 2D space using transformer based backbones.

The differences between the graph learning baselines and our method are illustrated in Table 5. Compared with the state-of-the-art method, our proposed SKI-CL backbone model learns a dynamic graph for nonstationary MTS modeling, which is also capable of incorporating structural knowledge with different forms and availability scenarios so as to characterize the dependency structure and temporal dynamics of each regime.

**Training Details** The dynamic graph inference module consists of 3 stacked 2D convolutional layers. Using $C_{in}$, $C_{out}$ to denote the number of channels coming in and out, the parameters of these convolutional layers are [ $C_{in}$ = 1, $C_{out}$ = 8, kernel size = (1,2), stride = 1, dilation = 2 ], [$C_{in}$ = 8, $C_{out}$ = 16, kernel size = (1,3), stride = 1, dilation = 2 ] and [$C_{in}$ = 16, $C_{out}$ = 32, kernel size = (1,3), stride = 1, dilation = 2 ] respectively. Each batched output is normalized using the Batch Norm2d layer. The hidden dimension for the node feature project is set at 128. For optimization,

Table 5: Taxonomy of Graph Learning Methods for MTS Forecasting

| Method | Learnable | Dynamic | Use Prior | Structure Form | | Structure Availability | |
|---|---|---|---|---|---|---|---|
| | | | | Binary | Continuous | Complete | Partial |
| STGCN | ✗ | - | ✓ | ✗ | ✓ | ✓ | ✗ |
| AGCRN | ✓ | ✗ | ✗ | - | - | - | - |
| MTGNN | ✓ | ✗ | ✗ | - | - | - | - |
| ESG | ✓ | ✓ | ✗ | - | - | - | - |
| StemGNN | ✓ | ✓ | ✗ | - | - | - | - |
| GTS | ✓ | ✗ | ✓ | ✓ | ✗ | ✓ | ✗ |
| **SKI-CL** | ✓ | ✓ | ✓ | ✓ | ✓ | ✓ | ✓ |

we train SKI-CL with 100 epochs for every stage under Adam optimizer with a linear scheduler. For the learning rate schedule, we use a linear scheduler, which drops the linear rate from 0.0001 to the factor of 0.8 for every 20 epochs. The data split is 6/2/2 for training/validation/testing. We use a batch size of 32/64/64 for the Traffic-CL, Solar-CL and HAR-CL datasets and use a batch size of 8 for our synthetic dataset. Considering the sizes of datasets, the default memory for each regime is 1% for Traffic-CL and Synthetic-CL and 0.1% for Solar-CL and HAR-CL, respectively. We also weigh the examples in the current stage and in memory differently. We apply a weighted loss regarding the sizes of memory and training data, as stated in the manuscript, and we also construct a data loader that guarantees the balance between training data and memory data. For the setting of our distribution characterization scheme, the default values are N = 10 and K = 7. We implement our models in Pytorch. All experiments are run on one server with four NVIDIA RTX A6000 GPUs. **We will release our code upon paper acceptance.**

### C.3 EVALUATIONS

**Multi-horizon MTS Forecasting** We use two common evaluation metrics for multi-horizon MTS forecasting, Mean Absolute Error (MAE) and Root Mean Squared Error (RMSE), which are given as:

$$\text{MAE}(Y, \hat{Y}) = \frac{1}{\tau} \sum_{t=1}^{\tau} |y_i - \hat{y}_i| \tag{6}$$

$$\text{RMSE}(Y, \hat{Y}) = \sqrt{\frac{1}{\tau} \sum_{t=1}^{\tau} (y_i - \hat{y}_i)^2} \tag{7}$$

where $\tau$ is the number of time steps, $\hat{y}_t$ is the forecasting results at $t$-th time step and $y_t$ is the corresponding ground truth. Besides, $\bar{y}$ and $\bar{\hat{y}}$ denote the mean values of ground truth and forecasting results, respectively.

**Learning Faithful Dependency Structures** For continuous edge variables, we still use MAE and RMSE to measure the similarity between the learned weighted graphs and continuous structural knowledge in an average sense, where the $\tau$ in Equations 6 and 7 denotes the entry of adjacency matrix. For binary edge variables, we use the average precision (Prec.) and recall (Rec.) to measure the similarity between the learned dependency structures over all testing MTS input windows and the binary structural prior at each regime, which are given as:

$$\text{Prec.} = \frac{\text{TP}}{\text{TP} + \text{FP}} \tag{8}$$

$$\text{Rec.} = \frac{\text{TP}}{\text{TP} + \text{FN}} \tag{9}$$

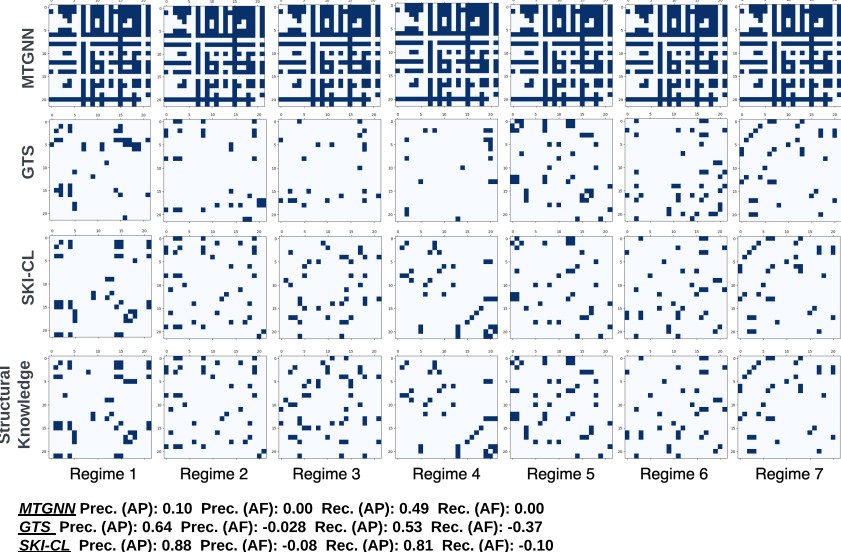

MTGNN Prec. (AP): 0.10  Prec. (AF): 0.00  Rec. (AP): 0.49  Rec. (AF): 0.00
GTS  Prec. (AP): 0.64  Prec. (AF): -0.028  Rec. (AP): 0.53  Rec. (AF): -0.37
SKI-CL  Prec. (AP): 0.88  Prec. (AF): -0.08  Rec. (AP): 0.81  Rec. (AF): -0.10

Figure 5: Visualization of Learned Structures on Traffic-CL Dataset.

where TP denotes the number of identified edges that exist in the structural prior, TN denotes the number of non-identified that do not exist in the structural prior, FP denotes the number of identified edges that do not exist in the structural prior, and FN denotes the number of non-identified edges that exist in the structural prior.

**Continual MTS Forecasting and Dependency Structures Preserving** We adopt two widely used metrics to evaluate the performance on continual MTS forecasting and dependency structures preserving, *i.e.*, the Average Performance (AP) and Average Forgetting (AF)  Lopez-Paz & Ranzato (2017); Zhang et al. (2022a), where the AP and AF at $i$-th regime are defined as:

$$\text{AP} = \sum_{j=1}^{i} \frac{\text{P}_{i,j}}{i}, \forall i \geq 1 \tag{10}$$

$$\text{AF} = \sum_{j=1}^{i-1} \frac{(\text{P}_{i,j} - \text{P}_{j,j})}{i-1}, \forall i \geq 2 \tag{11}$$

where $\text{P}_{i,j}$ denotes the performance on regime $j$ (including the forecasting performance and structure similarity) after the model has been sequentially trained from stage 1 to $i$.

Even if we provide both metrics for performance evaluation, we need to emphasize the superiority of average performance over average forgetting in continual learning. Average performance provides a direct measure of how well a learning system is performing on a task or set of tasks. It reflects the system's ability to retain previously learned knowledge over past regimes while adapting to new information. While average forgetting is a relevant metric in assessing the memory capabilities of a learning system, it does not provide a complete picture of the learning system's retention abilities. The average performance takes into account both the retention of old knowledge and the acquisition of new knowledge, providing a more comprehensive evaluation of the learning system's performance. Therefore, we use average performance as the main evaluation metric and average forgetting as an auxiliary metric to measure knowledge retention and model adaptivity.

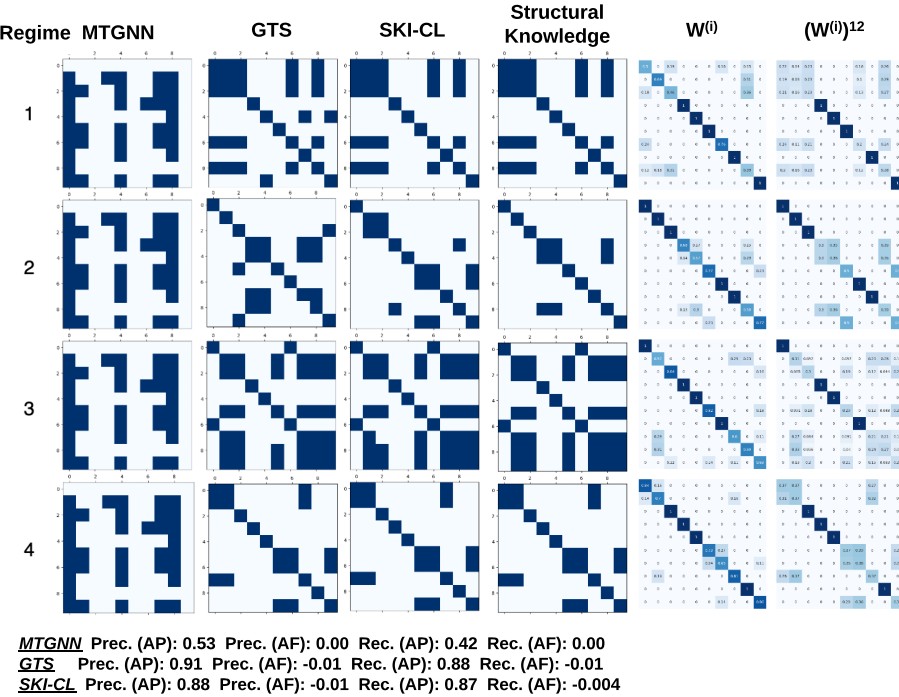

MTGNN  Prec. (AP): 0.53  Prec. (AF): 0.00  Rec. (AP): 0.42  Rec. (AF): 0.00
GTS    Prec. (AP): 0.91  Prec. (AF): -0.01  Rec. (AP): 0.88  Rec. (AF): -0.01
SKI-CL  Prec. (AP): 0.88  Prec. (AF): -0.01  Rec. (AP): 0.87  Rec. (AF): -0.004

Figure 6: Visualization of Learned Structures on Synthetic-CL Dataset.

## D  VISUALIZATION OF LEARNED DEPENDENCY STRUCTURES

In this section, we provide case studies of the learned dependency structures for continual MTS forecasting on all datasets, as shown in Figure 5,6,7, and 8 . For binary edge scenarios, we compare our proposed SKI-CL with MTGNN$_{er}$ and GTS$_{er}$, which only support discrete edge formulation and yield better performance over other baselines. For continuous edge scenarios, we compare our SKI-CL with ESG$_{er}$ that only support continuous edge formulation. For ESG$_{er}$, we visualize the generated graph at the last step representing the temporal dynamics of the whole sequence. All the results are plotted after the model is trained at the last regime, where the average structure similarities over all testing windows are also annotated under the case visualizations.

We first discuss the results on the Traffic-CL dataset as shown in Figure 5. As MTGNN directly learns a parameterized graph, it can only infer one fixed dependency structure reflecting the model at the latest regime. That being said, even if the model with experience replay is able to maintain the forecasting performance over the past regimes, the inferred graph fails to preserve the learned unique variable dependencies for each regime. Similarly, GTS infers a graph based on parameterized edge distributions, which also reveals the dependencies in a static sense. The incorporation of structural knowledge helps GTS characterize each regime in continual learning, which renders more relevant dependency structure for each regime. Nevertheless, learning a static edge distribution over regimes falls short of handling varying relational and temporal dynamics in complex environments. Another inherent drawback of GTS is that *GTS has to access the training data and memory buffer when inferring graphs at the testing stage, which is not realistic for practical model deployment in real-world applications*. Compared to these baselines, our model yields better forecasting performance (as shown in the manuscripts, as well as Table 8, and a faithful dependency structure that is more consistent with structure knowledge for each regime. These observations demonstrate the importance of the joint design of the dynamic graph inference module and the regularizer based on a structural prior.

For results on Synthetic-CL dataset as shown in Figure 6, the aforementioned observations still hold despite the comparative performance of GTS due to the simplicity of this dataset. Moreover, our model and GTS regularized with strong absolute correlations are able to render binary structures that

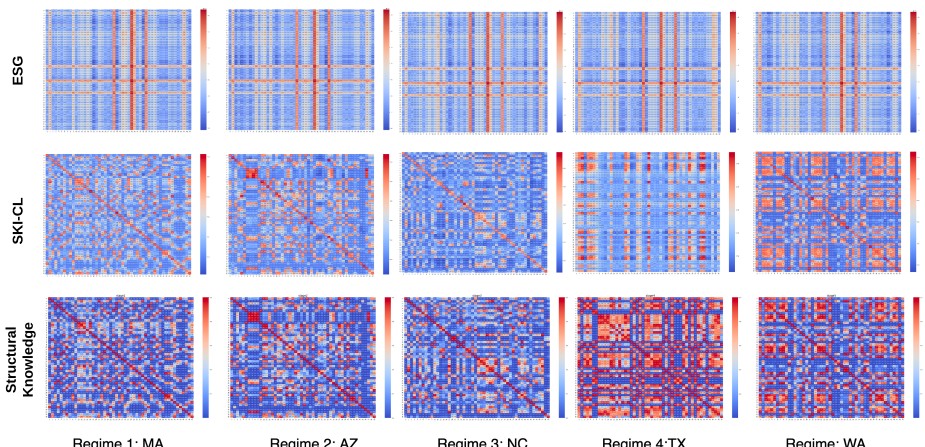

ESG MAE (AP): 0.35  MAE (AF): 0.01  RMSE (AP): 0.51  RMSE (AF): 0.01
SKI-CL MAE (AP): 0.17  MAE (AF): 0.03  RMSE (AP): 0.23  RMSE (AF): 0.05

Figure 7: Visualization of Learned Structures on Solar-CL Dataset.

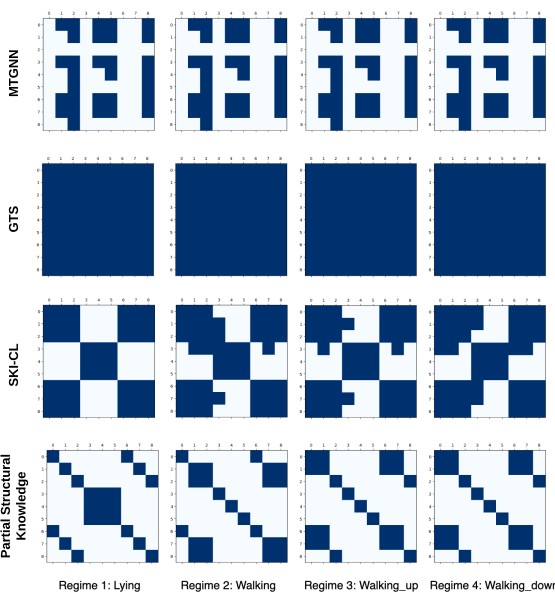

Figure 8: Visualization of Learned Structures on HAR-CL Dataset.

Table 6: Hyperparameter analysis for distribution characterization

| N | K | Forecasting Performance($\times 10^{-2}$) | | | | Structure Inference Performance | | | |
| | | AP | | AF | | AP | | AF | |
| | | MAE | RMSE | MAE | RMSE | Prec. | Recall | Prec. | Recall |
|---|---|---|---|---|---|---|---|---|---|
| 5 | 3 | 3.54 | 4.35 | 0.30 | 0.38 | 0.69 | 0.60 | -0.11 | -0.17 |
| 10 | 3 | 3.34 | 4.27 | 0.20 | 0.27 | 0.65 | 0.69 | -0.04 | -0.11 |
| 10 | 5 | 3.27 | 4.25 | 0.18 | 0.25 | 0.80 | 0.78 | -0.01 | -0.06 |
| 10 | 7 | 3.24 | 4.24 | 0.15 | 0.23 | 0.76 | 0.76 | -0.06 | -0.08 |
| 15 | 5 | 3.45 | 4.18 | 0.27 | 0.33 | 0.75 | 0.71 | -0.06 | -0.12 |
| 15 | 10 | 3.26 | 4.25 | 0.15 | 0.21 | 0.86 | 0.75 | -0.06 | -0.09 |

capture the single-step and multi-step variable interactions of the 'ground truth' dynamic adjacency matrix $W^i$ and its 12th power (the input window size) for each regime.

We next discuss the results on the Solar-CL dataset as shown in Figure 7, where the edge is formulated as a continuous variable. It is clear that our proposed SKI-CL still gains advantages in terms of preserving a faithful continuous dependency structure for each regime. Instead, ESG that learns dynamic graphs still falls short of capturing a relevant structure due to the lack of regime characterizations.

Finally, we investigate our method on the HAR-CL dataset (as shown in Figure 8) when the structural knowledge is partially observed. Here, we do not evaluate the structure similarity due to the incompleteness of the prior as a referencing graph. Instead, we focus on how our model leverages the limited but confident knowledge in dependency structure learning. It can be seen that MTGNN fails to capture the important relationships that reveal in the structural knowledge. Besides, GTS consistently inferred a fully connected graph at the testing stages (even if we have tuned the temperature in the Gumbel-Softmax module), which renders a less meaningful dependency structure. Instead, the proposed SKI-CL exploits partial knowledge and renders faithful structures. For example, SKI-CL is able to capture the correlations of linear accelerations, angular velocities, and total accelerations within three axes, and the irrelevance between accelerations and angular velocities, which is reasonable in a binary sense for lying down behavior. Besides, even if the partial structural knowledge is the same for walking upstairs and walking downstairs, the SKI-CL is able to identify different dependencies patterns for different activities. It demonstrates the effectiveness of the SKI-CL on partially observed structural knowledge, suggesting a certain generalizability of our proposed framework in MTS modeling.

## E  HYPERPARAMETERS ANALYSIS

In this section, we supplement additional analysis of distribution characterization hyperparameters, namely the granularity $N$ (10 by default) and the mode number $K$ (7 by default), using Synthetic-CL dataset. As shown in Table 6, for a fixed $N$, a relatively large $K$ facilitates inferred structure-preserving and mitigated the forgetting in time series forecasting. When $N$ and $K$ are close, average performance and average forgetting behavior are insensitive to the choice of these hyperparameters. However, a small $N$ degrades the structure-preserving ability as the average forgetting on precision and recall increases.

## F  EXPERIMENTAL RESULTS FOR DIFFERENT HORIZON FORECASTING

Table 8 summarize the experiment results of baselines and our proposed SKI-CL method with variants for different horizon forecasting performance based on three rounds of experiments. We intentionally select 3-horizon prediction on Traffic-CL and Solar-CL datasets as the settings in Wu et al. (2020) and Cao et al. (2020). While for HAR-CL and synthetic-CL dataset, 6-horizon prediction performance is reported. Based on the results, the memory-replay-based methods generally alleviate the forgetting issues with better APs and smaller relative AFs. Under different horizon prediction settings, SKI-CL and its variants (SKI-CL$_{er}$ and SKI-CL$_{der++}$) still consistently achieve the best or the second-best APs, demonstrating its advantages over other baseline methods.

Table 7: Experiment Results for 12 Horizon Prediction. (Lower MAE and RMSE for AP mean better; When AP is comparable, lower MAE and RMSE for AF mean better. )

| Model | Traffic-CL | | | | Solar-CL | | | | HAR-CL ($\times 10^{-2}$) | | | | Synthetic-CL ($\times 10^{-2}$) | | | |
|---|---|---|---|---|---|---|---|---|---|---|---|---|---|---|---|---|
| | AP $\downarrow$ | | AF | | AP $\downarrow$ | | AF | | AP $\downarrow$ | | AF | | AP $\downarrow$ | | AF | |
| | MAE | RMSE | MAE | RMSE | MAE | RMSE | MAE | RMSE | MAE | RMSE | MAE | RMSE | MAE | RMSE | MAE | RMSE |
| **VAR**$_{seq}$ | 88.19 | 126.01 | 58.38 | 80.58 | 167.30 | 534.42 | 205.27 | 658.80 | 19.59 | 28.38 | 1.93 | 2.19 | 22.34 | 32.70 | 9.18 | 13.54 |
| **ARIMA**$_{seq}$ | 141.75 | 159.89 | 77.61 | 77.40 | 14.97 | 18.92 | 4.75 | 2.92 | 40.68 | 52.87 | 2.35 | 2.38 | 42.24 | 43.51 | 13.39 | 12.98 |
| **LSTNet**$_{seq}$ | 27.01 | 42.87 | 11.19 | 16.72 | 3.15 | 5.76 | 1.12 | 1.14 | 16.00 | 23.96 | 2.78 | 3.77 | 24.13 | 31.35 | 5.46 | 7.32 |
| **LSTNet**$_{mir}$ | 21.79 | 35.37 | 4.39 | 5.21 | 2.73 | 5.51 | 0.93 | 0.84 | 15.73 | 23.14 | 1.45 | 1.67 | 21.44 | 27.23 | 0.57 | 0.72 |
| **LSTNet**$_{herd}$ | 20.86 | 33.56 | 3.24 | 4.58 | 3.09 | 5.71 | 1.08 | 0.99 | 15.06 | 22.98 | 1.37 | 1.59 | 20.31 | 26.22 | 0.36 | 0.43 |
| **LSTNet**$_{er}$ | 20.67 | 33.38 | 3.21 | 4.83 | 2.49 | 5.18 | 0.25 | 0.32 | 14.79 | 22.44 | 0.79 | 0.86 | 20.12 | 26.17 | 1.01 | 1.08 |
| **LSTNet**$_{der++}$ | 20.05 | 32.22 | 2.66 | 3.96 | 2.44 | 5.13 | 0.16 | 0.20 | 14.96 | 21.83 | 0.70 | 0.83 | 20.14 | 26.08 | 0.33 | 0.76 |
| **STGCN**$_{seq}$ | 30.45 | 49.51 | 13.91 | 20.77 | 2.99 | 5.75 | 0.81 | 0.99 | 17.36 | 26.38 | 1.79 | 2.25 | 8.89 | 13.67 | 4.55 | 7.36 |
| **STGCN**$_{mir}$ | 25.63 | 43.07 | 8.51 | 9.27 | 2.86 | 5.69 | 0.73 | 0.90 | 17.25 | 25.80 | 1.40 | 2.11 | 8.57 | 13.58 | 4.20 | 6.82 |
| **STGCN**$_{herd}$ | 24.89 | 41.42 | 5.04 | 8.51 | 2.83 | 5.62 | 0.67 | 0.74 | 17.30 | 25.84 | 1.49 | 2.16 | 8.50 | 13.21 | 3.80 | 6.46 |
| **STGCN**$_{er}$ | 28.53 | 47.63 | 9.38 | 15.79 | 2.81 | 5.57 | 0.81 | 0.95 | 16.19 | 24.81 | 1.15 | 1.96 | 7.77 | 12.10 | 3.56 | 5.99 |
| **STGCN**$_{der++}$ | 27.06 | 45.77 | 7.78 | 13.88 | 2.71 | 5.60 | 0.56 | 0.53 | 16.05 | 23.53 | 0.94 | 1.16 | 7.64 | 11.37 | 2.77 | 5.06 |
| **AGCRN**$_{seq}$ | 21.84 | 35.93 | 7.87 | 12.06 | 4.02 | 7.32 | -0.41 | -0.39 | 18.01 | 25.91 | 1.10 | 1.62 | 14.58 | 23.68 | 1.37 | 1.75 |
| **AGCRN**$_{mir}$ | 20.03 | 34.52 | 3.13 | 7.01 | 3.81 | 6.92 | -0.33 | -0.35 | 15.51 | 23.57 | 0.83 | 1.03 | 14.40 | 22.56 | 1.33 | 1.65 |
| **AGCRN**$_{herd}$ | 18.63 | 31.32 | 2.39 | 4.19 | 3.14 | 5.78 | -0.23 | -0.13 | 15.65 | 23.92 | 0.21 | 1.93 | 14.42 | 22.58 | 1.31 | 1.67 |
| **AGCRN**$_{er}$ | 18.58 | 31.00 | 2.82 | 5.17 | 2.11 | 4.07 | -1.77 | -2.53 | 15.26 | 23.25 | 0.32 | 0.51 | 13.67 | 20.98 | 0.56 | 1.11 |
| **AGCRN**$_{der++}$ | 18.29 | 30.41 | 2.58 | 4.56 | 4.36 | 7.63 | -0.65 | -0.86 | 15.23 | 23.23 | 0.28 | 0.49 | 13.14 | 20.62 | 1.25 | 1.54 |
| **StemGNN**$_{seq}$ | 18.53 | 31.23 | 4.48 | 7.44 | 2.79 | 5.53 | 0.42 | 0.52 | 16.19 | 24.81 | 2.21 | 3.01 | 13.83 | 20.01 | 1.91 | 1.88 |
| **StemGNN**$_{mir}$ | 17.73 | 29.75 | 2.21 | 5.17 | 2.75 | 5.51 | 0.15 | 0.39 | 16.11 | 23.93 | 1.53 | 1.77 | 12.75 | 19.23 | 1.13 | 1.45 |
| **StemGNN**$_{herd}$ | 17.55 | 29.63 | 1.73 | 3.29 | 2.78 | 5.50 | 1.05 | 1.16 | 15.95 | 23.77 | 1.04 | 1.53 | 12.69 | 18.46 | 1.10 | 1.30 |
| **StemGNN**$_{er}$ | 17.01 | 28.68 | 2.07 | 3.69 | 2.73 | 5.52 | 0.04 | 0.16 | 15.95 | 23.32 | 1.21 | 1.25 | 12.13 | 18.18 | 0.65 | 0.59 |
| **StemGNN**$_{der++}$ | 17.26 | 29.21 | 1.63 | 3.29 | 2.20 | 4.88 | -0.04 | 0.02 | 15.78 | 23.12 | 1.01 | 0.92 | 12.19 | 17.98 | 0.26 | 0.61 |
| **TCN**$_{seq}$ | 16.88 | 28.67 | 3.77 | 6.83 | 2.03 | 4.84 | 0.06 | 0.24 | 14.85 | 23.42 | 3.60 | 5.06 | 4.30 | 4.90 | 0.66 | 0.99 |
| **TCN**$_{mir}$ | 15.70 | 26.53 | 1.70 | 3.22 | 1.99 | 4.79 | 0.10 | 0.19 | 13.91 | 22.15 | 2.64 | 2.93 | 3.79 | 4.63 | 0.46 | 0.73 |
| **TCN**$_{herd}$ | 15.55 | 26.21 | 1.49 | 2.81 | 2.01 | 4.82 | 0.13 | 0.23 | 13.87 | 22.08 | 2.05 | 2.88 | 3.72 | 4.61 | 0.35 | 0.67 |
| **TCN**$_{er}$ | 15.51 | 26.23 | 1.46 | 2.80 | 1.98 | 4.73 | -0.05 | 0.02 | 13.66 | 21.78 | 1.82 | 2.59 | 3.29 | 4.30 | 0.34 | 0.61 |
| **TCN**$_{der++}$ | 15.46 | 25.68 | 1.33 | 2.49 | 1.95 | 4.69 | -0.07 | -0.02 | 13.56 | 21.55 | 1.69 | 2.28 | **3.00** | **4.00** | 0.28 | 0.32 |
| **ESG**$_{seq}$ | 18.77 | 30.02 | 6.46 | 10.07 | 2.80 | 5.77 | 1.28 | 1.74 | 17.63 | 26.84 | 7.28 | 9.41 | 8.98 | 14.19 | 1.32 | 1.98 |
| **ESG**$_{mir}$ | 18.24 | 29.83 | 5.02 | 8.25 | 2.03 | 4.83 | 0.25 | 0.49 | 17.25 | 26.63 | 4.01 | 5.21 | 8.95 | 13.91 | 1.21 | 1.81 |
| **ESG**$_{herd}$ | 17.49 | 28.64 | 4.82 | 7.45 | 1.92 | 4.72 | 0.13 | 0.53 | 17.22 | 26.59 | 3.99 | 5.13 | 8.94 | 13.88 | 1.11 | 1.74 |
| **ESG**$_{er}$ | 16.40 | 27.50 | 3.05 | 5.34 | 2.01 | 4.82 | 0.24 | 0.44 | 17.15 | 25.84 | 4.63 | 5.33 | 8.84 | 13.86 | 1.21 | 1.62 |
| **ESG**$_{der++}$ | 17.40 | 29.21 | 4.01 | 6.97 | 1.91 | 4.57 | 0.09 | 0.21 | 16.20 | 24.32 | 5.18 | 6.00 | 8.81 | 13.77 | 1.02 | 1.42 |
| **GTS**$_{seq}$ | 17.26 | 29.11 | 2.33 | 3.48 | 2.19 | 5.20 | 0.27 | 0.59 | 16.44 | 25.41 | 3.68 | 5.10 | 6.51 | 8.89 | 1.88 | 3.39 |
| **GTS**$_{mir}$ | 17.17 | 29.08 | 2.13 | 3.31 | 2.15 | 5.16 | 0.14 | 0.68 | 15.83 | 24.85 | 3.27 | 4.91 | 6.44 | 8.57 | 1.31 | 2.86 |
| **GTS**$_{herd}$ | 17.00 | 29.01 | 2.17 | 2.98 | 2.11 | 5.06 | 0.13 | 0.30 | 15.65 | 24.33 | 1.99 | 2.86 | 6.34 | 8.23 | 1.18 | 1.81 |
| **GTS**$_{er}$ | 15.83 | 26.20 | 1.12 | 2.52 | 2.01 | 4.75 | 0.12 | 0.05 | 15.06 | 23.52 | 2.00 | 2.73 | 5.59 | 6.32 | 0.44 | 0.69 |
| **GTS**$_{der++}$ | 15.84 | 26.05 | 1.15 | 2.33 | 1.94 | 4.57 | -0.25 | -0.19 | 14.80 | 23.01 | 1.52 | 1.88 | 5.43 | 6.67 | 0.23 | 0.30 |
| **MTGNN**$_{seq}$ | 19.88 | 32.94 | 7.83 | 12.68 | 2.12 | 4.75 | 0.38 | 0.44 | 14.86 | 22.58 | 2.59 | 3.61 | 10.26 | 14.92 | 1.16 | 1.81 |
| **MTGNN**$_{mir}$ | 18.01 | 31.84 | 5.03 | 8.97 | 2.00 | 4.73 | 0.21 | 0.40 | 14.59 | 22.52 | 2.24 | 3.53 | 8.92 | 12.91 | 1.07 | 1.33 |
| **MTGNN**$_{herd}$ | 17.93 | 30.70 | 4.90 | 8.40 | 1.89 | 4.68 | 0.13 | 0.35 | 14.09 | 22.50 | 1.13 | 1.62 | 8.11 | 12.88 | 1.03 | 1.27 |
| **MTGNN**$_{er}$ | 15.79 | 26.52 | 2.76 | 4.87 | 1.94 | 4.62 | 0.14 | 0.25 | 13.59 | 21.85 | 1.91 | 2.79 | 8.63 | 13.69 | 0.61 | 1.21 |
| **MTGNN**$_{der++}$ | 15.40 | 25.99 | 2.22 | 4.10 | 1.90 | 4.57 | 0.06 | 0.14 | 13.57 | 21.75 | 1.63 | 2.40 | 8.63 | 13.51 | 0.50 | 0.92 |
| **Autoformer**$_{seq}$ | 23.92 | 40.40 | 2.58 | 4.42 | 6.04 | 11.41 | 0.53 | 1.56 | 19.97 | 28.76 | 2.81 | 3.08 | 5.15 | 6.87 | 0.53 | 0.46 |
| **Autoformer**$_{mir}$ | 23.85 | 40.13 | 2.21 | 4.07 | 5.95 | 11.18 | 0.87 | 1.14 | 18.57 | 27.93 | 1.45 | 2.88 | 5.12 | 6.73 | 0.32 | 0.37 |
| **Autoformer**$_{herd}$ | 23.90 | 40.21 | 2.43 | 4.29 | 5.91 | 11.12 | 0.90 | 1.20 | 18.78 | 28.03 | 1.42 | 2.66 | 5.00 | 6.65 | 0.23 | 0.24 |
| **Autoformer**$_{er}$ | 23.26 | 38.98 | 1.28 | 2.00 | 5.83 | 10.89 | 0.49 | 1.32 | 18.42 | 27.05 | 1.29 | 2.01 | 5.02 | 6.66 | 0.24 | 0.25 |
| **Autoformer**$_{der++}$ | 22.11 | 37.34 | 1.08 | 1.92 | 5.34 | 9.29 | 0.21 | 1.14 | 18.12 | 26.91 | 1.12 | 1.85 | 4.97 | 6.61 | 0.18 | 0.23 |
| **PatchTST**$_{seq}$ | 19.11 | 32.50 | 2.34 | 2.97 | 2.64 | 5.32 | 0.72 | 0.43 | 17.91 | 27.13 | 7.18 | 6.88 | 4.85 | 5.93 | 1.59 | 1.78 |
| **PatchTST**$_{mir}$ | 19.04 | 32.23 | 2.28 | 2.79 | 2.61 | 5.30 | 0.70 | 0.40 | 17.82 | 26.89 | 6.82 | 4.81 | 4.83 | 5.86 | 1.55 | 1.72 |
| **PatchTST**$_{herd}$ | 18.96 | 32.10 | 2.21 | 2.67 | 2.60 | 5.30 | 0.68 | 0.35 | 17.73 | 26.84 | 6.62 | 4.79 | 4.80 | 5.79 | 1.43 | 1.68 |
| **PatchTST**$_{er}$ | 18.77 | 31.50 | 1.98 | 2.01 | 2.57 | 5.27 | 0.47 | 0.30 | 17.57 | 26.40 | 6.02 | 4.69 | 4.72 | 5.26 | 1.03 | 1.54 |
| **PatchTST**$_{der++}$ | 18.53 | 31.34 | 1.75 | 1.98 | 2.53 | 5.17 | 0.43 | 0.28 | 17.12 | 26.13 | 5.79 | 4.32 | 4.64 | 5.13 | 0.83 | 0.88 |
| **DLinear**$_{seq}$ | 19.69 | 32.75 | 2.91 | 2.83 | 3.47 | 6.56 | 1.17 | 1.12 | 17.32 | 26.31 | 2.71 | 3.43 | 4.81 | 5.81 | 1.64 | 1.57 |
| **DLinear**$_{mir}$ | 19.37 | 32.25 | 2.17 | 2.59 | 3.45 | 6.51 | 1.02 | 1.01 | 16.87 | 26.12 | 2.67 | 3.01 | 4.79 | 5.73 | 1.47 | 1.40 |
| **DLinear**$_{herd}$ | 19.53 | 32.40 | 2.25 | 2.68 | 3.41 | 6.50 | 1.03 | 1.00 | 16.83 | 25.81 | 2.57 | 2.91 | 4.77 | 5.70 | 1.59 | 1.46 |
| **DLinear**$_{er}$ | 19.19 | 32.30 | 1.73 | 2.14 | 3.37 | 6.43 | 0.93 | 0.98 | 16.71 | 25.75 | 2.13 | 2.85 | 4.74 | 5.20 | 1.23 | 1.43 |
| **DLinear**$_{der++}$ | 19.02 | 31.97 | 1.75 | 1.93 | 3.25 | 6.37 | 0.83 | 0.79 | 16.58 | 25.47 | 1.92 | 2.77 | 4.21 | 4.88 | 1.12 | 1.13 |
| **TimesNet**$_{seq}$ | 17.77 | 29.91 | 3.13 | 6.93 | 3.92 | 7.18 | 1.46 | 2.51 | 18.38 | 27.61 | 4.33 | 5.15 | 5.18 | 6.13 | 1.72 | 2.03 |
| **TimesNet**$_{mir}$ | 17.53 | 29.61 | 2.44 | 5.32 | 3.77 | 7.15 | 1.22 | 1.57 | 18.27 | 27.59 | 4.01 | 5.08 | 5.12 | 6.05 | 1.69 | 1.97 |
| **TimesNet**$_{herd}$ | 17.38 | 29.53 | 2.56 | 5.83 | 3.83 | 7.10 | 1.03 | 1.44 | 18.01 | 27.53 | 3.46 | 5.03 | 5.10 | 6.03 | 1.68 | 1.95 |
| **TimesNet**$_{er}$ | 17.25 | 29.33 | 1.97 | 4.19 | 3.55 | 7.02 | 0.42 | 0.91 | 17.84 | 27.07 | 3.28 | 4.01 | 4.93 | 5.90 | 1.42 | 1.90 |
| **TimesNet**$_{der++}$ | 17.13 | 29.28 | 1.56 | 4.02 | 3.45 | 6.55 | 0.37 | 0.90 | 17.73 | 26.86 | 3.11 | 3.87 | 4.81 | 5.88 | 1.32 | 1.78 |
| **SKI-CL**$_{seq}$ | 17.30 | 29.38 | 4.38 | 7.80 | 2.02 | 4.73 | 0.30 | 0.50 | 14.73 | 23.31 | 3.91 | 5.07 | 4.70 | 5.85 | 1.97 | 3.46 |
| **SKI-CL**$_{mir}$ | 15.77 | 26.32 | 1.95 | 3.47 | 1.98 | 4.69 | 0.35 | 0.56 | 13.65 | 21.77 | 2.61 | 3.82 | 4.53 | 5.01 | 1.67 | 2.21 |
| **SKI-CL**$_{herd}$ | 15.45 | 25.73 | 1.82 | 3.28 | 2.00 | 4.70 | 0.33 | 0.52 | 13.71 | 21.82 | 2.53 | 3.67 | 4.44 | 4.82 | 1.23 | 2.02 |
| **SKI-CL**$_{er}$ | 15.43 | 25.60 | 1.69 | 2.92 | 1.95 | 4.67 | 0.11 | 0.23 | 13.58 | 21.57 | 1.82 | 2.50 | 3.39 | 4.52 | 0.30 | 0.38 |
| **SKI-CL**$_{der++}$ | 15.39 | 25.57 | 1.63 | 2.87 | 1.91 | 4.60 | 0.10 | 0.21 | 13.50 | 21.47 | 1.74 | 2.42 | 3.33 | 4.43 | 0.28 | 0.30 |
| **SKI-CL** | **15.24** | **25.33** | 1.50 | 2.71 | **1.75** | **4.46** | 0.09 | 0.06 | **13.41** | **21.30** | 1.64 | 2.08 | 3.24 | 4.24 | 0.15 | 0.23 |

We also provide the standard deviation of AP and AF for the above continual forecasting performance, as shown in Table 9. It can be seen that the standard deviation of SKI-CL is generally comparative or lower to other variants except the Synthetic-CL dataset.

Table 8: Experiment Results for Different Horizon Prediction (3-step Horizon Prediction for Traffic-CL and Solar-CL and 6-step Horizon Prediction for HAR-CL and Synthetic-CL) (Lower MAE and RMSE for AP mean better; When AP is comparable, lower MAE and RMSE for AF mean better. )

| Model | Traffic-CL (3) | | | | Solar-CL (3) | | | | HAR-CL (6) ($\times 10^{-2}$) | | | | Synthetic-CL (6) ($\times 10^{-2}$) | | | |
|---|---|---|---|---|---|---|---|---|---|---|---|---|---|---|---|---|
| | AP↓ | | AF | | AP↓ | | AF | | AP↓ | | AF | | AP↓ | | AF | |
| | MAE | RMSE | MAE | RMSE | MAE | RMSE | MAE | RMSE | MAE | RMSE | MAE | RMSE | MAE | RMSE | MAE | RMSE |
| **VAR**$_{seq}$ | 73.36 | 104.97 | 70.26 | 97.19 | 59.30 | 196.75 | 72.88 | 242.41 | 18.04 | 27.37 | 1.01 | 1.28 | 18.61 | 27.33 | 7.70 | 11.40 |
| **ARIMA**$_{seq}$ | 141.53 | 159.81 | 77.65 | 78.19 | 14.93 | 19.12 | 5.84 | 3.71 | 41.02 | 53.11 | 2.42 | 2.11 | 42.25 | 43.53 | 13.41 | 13.01 |
| **TCN**$_{seq}$ | 13.89 | 22.12 | 2.57 | 4.77 | 0.93 | 2.71 | 0.12 | 0.13 | 12.71 | 20.46 | 2.98 | 4.35 | 3.81 | 3.37 | 0.26 | 0.33 |
| **TCN**$_{herd}$ | 13.62 | 21.97 | 1.63 | 1.89 | 0.90 | 2.64 | 0.04 | 0.03 | 11.85 | 19.34 | 1.70 | 2.53 | 2.77 | 3.31 | 0.23 | 0.31 |
| **TCN**$_{er}$ | 13.21 | 21.85 | 1.58 | 1.63 | 0.92 | 2.64 | 0.03 | 0.02 | 11.68 | 18.98 | 1.54 | 2.29 | 2.67 | 3.18 | 0.18 | 0.22 |
| **TCN**$_{der++}$ | 13.50 | 21.90 | 1.48 | 1.57 | 0.87 | 2.63 | 0.03 | 0.05 | 11.63 | 18.97 | 1.46 | 2.11 | 2.73 | 3.29 | 0.15 | 0.17 |
| **Autoformer**$_{seq}$ | 18.19 | 30.22 | 29.37 | 4.23 | 3.19 | 6.47 | 0.82 | 1.69 | 16.79 | 24.78 | 1.83 | 1.98 | 3.83 | 5.38 | 0.35 | 0.24 |
| **Autoformer**$_{herd}$ | 17.72 | 2.67 | 1.89 | 3.02 | 2.89 | 5.34 | 0.67 | 1.23 | 16.01 | 24.12 | 1.58 | 1.78 | 3.71 | 5.26 | 0.31 | 0.20 |
| **Autoformer**$_{er}$ | 16.07 | 26.82 | 1.36 | 2.54 | 2.54 | 5.02 | 0.13 | 0.23 | 15.66 | 23.88 | 1.57 | 1.58 | 3.65 | 5.26 | 0.24 | 0.17 |
| **Autoformer**$_{der++}$ | 16.17 | 27.01 | 1.74 | 2.93 | 2.47 | 4.79 | 0.12 | 0.33 | 16.14 | 24.28 | 1.59 | 1.56 | 3.52 | 5.01 | 0.17 | 0.13 |
| **LSTNet**$_{seq}$ | 24.89 | 39.49 | 10.50 | 15.63 | 2.45 | 4.77 | 1.09 | 1.45 | 14.13 | 21.90 | 4.05 | 5.49 | 25.31 | 33.71 | 7.44 | 10.61 |
| **LSTNet**$_{herd}$ | 19.11 | 30.68 | 3.22 | 4.73 | 2.26 | 4.43 | 0.81 | 0.89 | 13.12 | 20.66 | 2.64 | 3.66 | 19.75 | 26.28 | 0.60 | 0.90 |
| **LSTNet**$_{er}$ | 18.29 | 29.42 | 2.55 | 3.95 | 1.87 | 4.10 | 0.37 | 0.55 | 12.65 | 20.06 | 2.08 | 2.94 | 19.31 | 24.76 | 0.31 | 0.77 |
| **LSTNet**$_{der++}$ | 17.94 | 28.96 | 2.03 | 3.18 | 1.78 | 3.98 | 0.18 | 0.28 | 12.55 | 19.94 | 1.85 | 2.86 | 18.92 | 24.26 | 0.77 | 0.57 |
| **STGCN**$_{seq}$ | 26.80 | 42.22 | 11.90 | 14.48 | 3.14 | 6.30 | 1.86 | 3.16 | 16.52 | 25.04 | 5.21 | 6.52 | 8.21 | 12.90 | 2.27 | 4.63 |
| **STGCN**$_{herd}$ | 26.53 | 43.56 | 9.36 | 14.79 | 2.57 | 5.99 | 1.15 | 2.65 | 15.41 | 23.97 | 4.12 | 5.69 | 5.53 | 8.65 | 1.42 | 3.37 |
| **STGCN**$_{er}$ | 25.54 | 42.43 | 8.29 | 11.71 | 2.41 | 4.88 | 1.10 | 1.52 | 15.11 | 23.54 | 3.72 | 4.25 | 5.12 | 8.49 | 1.15 | 1.76 |
| **STGCN**$_{der++}$ | 25.53 | 42.43 | 8.12 | 10.03 | 2.26 | 4.41 | 0.66 | 0.63 | 15.51 | 23.64 | 3.52 | 4.13 | 6.95 | 11.12 | 1.86 | 2.89 |
| **AGCRN**$_{seq}$ | 16.13 | 26.21 | 3.65 | 5.29 | 1.44 | 3.21 | 0.31 | 0.60 | 13.82 | 21.88 | 2.97 | 4.06 | 13.00 | 21.16 | 0.99 | 0.47 |
| **AGCRN**$_{herd}$ | 15.17 | 25.32 | 1.59 | 2.71 | 1.30 | 3.14 | 0.11 | 0.33 | 12.32 | 20.63 | 2.03 | 3.96 | 12.18 | 19.27 | 0.12 | 0.16 |
| **AGCRN**$_{er}$ | 14.88 | 24.72 | 1.47 | 2.43 | 1.27 | 3.04 | 0.10 | 0.04 | 12.28 | 20.54 | 1.98 | 3.03 | 10.84 | 16.82 | 0.10 | 0.17 |
| **AGCRN**$_{der++}$ | 15.02 | 25.15 | 1.41 | 2.66 | 1.17 | 3.01 | 0.03 | 0.03 | 12.01 | 20.18 | 1.76 | 2.95 | 12.40 | 19.56 | 0.25 | 0.23 |
| **StemGNN**$_{seq}$ | 14.10 | 23.51 | 1.92 | 1.63 | 1.30 | 3.10 | 0.24 | 0.28 | 17.31 | 25.43 | 6.31 | 7.23 | 11.71 | 17.77 | 1.54 | 1.01 |
| **StemGNN**$_{herd}$ | 13.78 | 23.01 | 0.98 | 1.03 | 1.17 | 2.92 | 0.22 | 0.25 | 16.51 | 25.03 | 5.21 | 4.23 | 9.17 | 13.97 | 1.03 | 0.65 |
| **StemGNN**$_{er}$ | 13.49 | 22.59 | 0.36 | 0.42 | 1.00 | 2.73 | 0.04 | 0.05 | 16.43 | 24.83 | 5.19 | 4.13 | 8.38 | 12.95 | 0.96 | 0.97 |
| **StemGNN**$_{der++}$ | 13.37 | 22.42 | 0.31 | 0.37 | 1.03 | 2.87 | 0.04 | 0.07 | 16.25 | 24.51 | 5.10 | 4.01 | 9.02 | 13.65 | 0.89 | 0.85 |
| **ESG**$_{seq}$ | 14.62 | 23.83 | 3.72 | 5.39 | 1.75 | 2.93 | 0.92 | 0.83 | 16.57 | 25.07 | 5.34 | 6.21 | 8.68 | 13.95 | 1.06 | 1.57 |
| **ESG**$_{herd}$ | 14.53 | 23.79 | 3.26 | 4.71 | 1.78 | 2.78 | 0.14 | 0.27 | 14.31 | 23.17 | 4.48 | 5.39 | 7.61 | 12.25 | 0.71 | 1.15 |
| **ESG**$_{er}$ | 13.07 | 22.11 | 1.62 | 2.93 | 1.15 | 2.71 | 0.32 | 0.25 | 14.21 | 23.01 | 4.37 | 5.21 | 7.21 | 11.65 | 0.61 | 0.89 |
| **ESG**$_{der++}$ | 13.92 | 23.17 | 2.52 | 3.98 | 0.98 | 2.67 | 0.16 | 0.21 | 14.10 | 22.87 | 4.13 | 4.83 | 8.08 | 12.95 | 0.35 | 0.52 |
| **GTS**$_{seq}$ | 15.20 | 27.09 | 3.23 | 7.32 | 1.20 | 3.19 | 0.19 | 0.48 | 14.44 | 23.41 | 3.51 | 4.63 | 4.90 | 6.71 | 0.56 | 0.61 |
| **GTS**$_{herd}$ | 14.33 | 23.55 | 2.82 | 3.78 | 1.03 | 2.69 | 0.05 | 0.10 | 13.78 | 22.89 | 3.08 | 4.05 | 4.71 | 6.66 | 0.13 | 0.20 |
| **GTS**$_{er}$ | 14.54 | 24.65 | 1.96 | 3.58 | 0.94 | 2.67 | 0.08 | 0.05 | 13.38 | 22.45 | 2.95 | 3.97 | 4.41 | 6.40 | 0.13 | 0.19 |
| **GTS**$_{der++}$ | 14.07 | 24.12 | 1.25 | 3.39 | 0.96 | 2.67 | 0.05 | 0.15 | 13.34 | 22.41 | 2.93 | 3.89 | 4.39 | 6.39 | 0.12 | 0.18 |
| **MTGNN**$_{seq}$ | 14.42 | 23.94 | 3.34 | 5.13 | 1.14 | 2.76 | 0.32 | 0.30 | 12.87 | 20.66 | 3.51 | 4.80 | 7.63 | 12.44 | 0.59 | 0.67 |
| **MTGNN**$_{herd}$ | 13.55 | 22.55 | 2.13 | 3.35 | 0.96 | 2.69 | 0.13 | 0.23 | 11.98 | 19.78 | 2.42 | 3.61 | 7.43 | 11.99 | 0.45 | 0.61 |
| **MTGNN**$_{er}$ | 13.95 | 22.29 | 2.03 | 3.02 | 0.95 | 2.74 | 0.05 | 0.16 | 11.30 | 18.67 | 1.51 | 2.15 | 6.38 | 10.13 | 0.35 | 0.53 |
| **MTGNN**$_{der++}$ | 12.99 | 21.86 | 1.24 | 2.24 | 0.91 | 2.72 | 0.03 | 0.10 | 11.27 | 18.64 | 1.22 | 1.81 | 6.31 | 10.10 | 0.18 | 0.41 |
| **SKI-CL**$_{seq}$ | 13.81 | 23.16 | 2.52 | 4.25 | 0.90 | 2.63 | 0.09 | 0.14 | 12.41 | 19.37 | 2.54 | 3.78 | 3.72 | 5.26 | 2.02 | 3.10 |
| **SKI-CL**$_{herd}$ | 12.91 | 21.80 | 1.43 | 2.55 | 0.87 | 2.65 | 0.07 | 0.12 | 11.78 | 19.04 | 1.89 | 2.95 | 3.11 | 4.18 | 0.23 | 0.45 |
| **SKI-CL**$_{er}$ | 12.66 | 21.21 | 1.26 | 2.27 | 0.88 | 2.64 | 0.05 | 0.06 | 11.67 | 18.95 | 1.54 | 2.35 | 2.85 | 3.60 | 0.17 | 0.16 |
| **SKI-CL**$_{der++}$ | 12.61 | 21.20 | 1.10 | 1.72 | 0.85 | 2.56 | 0.04 | 0.05 | 11.55 | 18.84 | 1.40 | 2.06 | 2.33 | 3.05 | 0.10 | 0.13 |
| **SKI-CL** | **12.49** | **21.01** | 0.93 | 1.60 | **0.82** | **2.54** | 0.02 | 0.02 | **11.01** | **18.25** | 1.21 | 1.78 | **2.01** | **2.85** | 0.05 | 0.03 |

Table 9: Standard Deviations for 12 Horizon Prediction.

| Model | Traffic-CL AP MAE | RMSE | AF MAE | RMSE | Solar-CL AP MAE | RMSE | AF MAE | RMSE | HAR-CL ($\times 10^{-2}$) AP MAE | RMSE | AF MAE | RMSE | Synthetic-CL ($\times 10^{-2}$) AP MAE | RMSE | AF MAE | RMSE |
|---|---|---|---|---|---|---|---|---|---|---|---|---|---|---|---|---|
| LSTNet$_{seq}$ | 0.42 | 0.65 | 0.55 | 0.74 | 0.08 | 0.03 | 0.09 | 0.03 | 0.41 | 0.51 | 0.45 | 0.61 | 0.48 | 0.64 | 0.81 | 0.17 |
| LSTNet$_{mir}$ | 0.21 | 0.27 | 0.42 | 0.64 | 0.05 | 0.05 | 0.08 | 0.04 | 0.32 | 0.43 | 0.39 | 0.58 | 0.47 | 0.41 | 0.42 | 0.11 |
| LSTNet$_{herd}$ | 0.17 | 0.21 | 0.40 | 0.60 | 0.04 | 0.03 | 0.07 | 0.03 | 0.28 | 0.42 | 0.44 | 0.57 | 0.43 | 0.45 | 0.39 | 0.12 |
| LSTNet$_{rnd}$ | 0.19 | 0.25 | 0.39 | 0.65 | 0.02 | 0.07 | 0.05 | 0.05 | 0.34 | 0.48 | 0.67 | 0.52 | 0.40 | 0.40 | 0.41 | 0.11 |
| LSTNet$_{der++}$ | 0.07 | 0.13 | 0.15 | 0.21 | 0.03 | 0.05 | 0.04 | 0.04 | 0.28 | 0.37 | 0.29 | 0.45 | 0.30 | 0.35 | 0.49 | 0.61 |
| STGCN$_{seq}$ | 1.09 | 1.91 | 1.30 | 2.25 | 0.07 | 0.09 | 0.11 | 0.12 | 0.35 | 0.47 | 0.37 | 0.31 | 1.02 | 1.62 | 0.60 | 0.90 |
| STGCN$_{mir}$ | 1.71 | 1.49 | 1.90 | 2.71 | 0.08 | 0.22 | 0.03 | 0.14 | 0.25 | 0.43 | 0.31 | 0.23 | 0.27 | 0.60 | 0.39 | 0.57 |
| STGCN$_{herd}$ | 1.54 | 2.10 | 1.72 | 2.50 | 0.11 | 0.30 | 0.07 | 0.31 | 0.28 | 0.27 | 0.20 | 0.32 | 0.38 | 0.59 | 0.36 | 0.63 |
| STGCN$_{rnd}$ | 1.65 | 1.41 | 1.87 | 2.94 | 0.07 | 0.16 | 0.05 | 0.19 | 0.27 | 0.41 | 0.25 | 0.27 | 0.25 | 0.63 | 0.42 | 0.59 |
| STGCN$_{der++}$ | 1.58 | 2.18 | 1.77 | 2.54 | 0.10 | 0.27 | 0.15 | 0.35 | 0.23 | 0.27 | 0.19 | 0.25 | 0.38 | 0.59 | 0.42 | 0.60 |
| AGCRN$_{seq}$ | 0.44 | 0.58 | 0.45 | 0.88 | 0.33 | 0.26 | 0.24 | 0.35 | 0.61 | 0.51 | 0.41 | 0.37 | 0.26 | 0.79 | 0.13 | 0.35 |
| AGCRN$_{mir}$ | 0.37 | 0.41 | 0.37 | 0.70 | 0.27 | 0.32 | 0.33 | 0.20 | 0.53 | 0.41 | 0.35 | 0.44 | 0.35 | 0.83 | 0.31 | 0.51 |
| AGCRN$_{herd}$ | 0.49 | 0.99 | 0.38 | 0.92 | 0.36 | 0.26 | 0.43 | 0.17 | 0.48 | 0.52 | 0.55 | 0.59 | 0.28 | 0.95 | 0.36 | 0.47 |
| AGCRN$_{rnd}$ | 0.31 | 0.36 | 0.34 | 0.64 | 0.15 | 0.35 | 0.31 | 0.13 | 0.58 | 0.47 | 0.39 | 0.40 | 0.30 | 0.89 | 0.25 | 0.50 |
| AGCRN$_{der++}$ | 0.56 | 1.15 | 0.43 | 0.85 | 0.29 | 0.27 | 0.37 | 0.18 | 0.45 | 0.62 | 0.49 | 0.56 | 0.30 | 0.95 | 0.32 | 0.21 |
| StemGNN$_{seq}$ | 0.22 | 0.18 | 0.77 | 1.24 | 0.16 | 0.13 | 0.18 | 0.19 | 0.25 | 0.47 | 0.23 | 0.31 | 0.95 | 0.44 | 0.38 | 0.58 |
| StemGNN$_{mir}$ | 0.20 | 0.23 | 0.32 | 0.43 | 0.19 | 0.20 | 0.14 | 0.26 | 0.29 | 0.35 | 0.31 | 0.22 | 0.24 | 0.27 | 0.42 | 0.28 |
| StemGNN$_{herd}$ | 0.25 | 0.43 | 0.72 | 0.85 | 0.15 | 0.19 | 0.15 | 0.13 | 0.28 | 0.33 | 0.26 | 0.24 | 0.45 | 0.70 | 0.40 | 0.43 |
| StemGNN$_{rnd}$ | 0.16 | 0.16 | 0.21 | 0.22 | 0.17 | 0.18 | 0.14 | 0.22 | 0.23 | 0.39 | 0.27 | 0.21 | 0.26 | 0.26 | 0.49 | 0.32 |
| StemGNN$_{der++}$ | 0.29 | 0.40 | 0.65 | 0.89 | 0.12 | 0.11 | 0.19 | 0.07 | 0.28 | 0.41 | 0.29 | 0.31 | 0.54 | 0.71 | 0.55 | 0.47 |
| TCN$_{seq}$ | 0.11 | 0.15 | 0.13 | 0.17 | 0.05 | 0.07 | 0.08 | 0.13 | 0.01 | 0.03 | 0.05 | 0.08 | 0.05 | 0.05 | 0.25 | 0.44 |
| TCN$_{mir}$ | 0.18 | 0.10 | 0.12 | 0.21 | 0.15 | 0.05 | 0.12 | 0.09 | 0.09 | 0.11 | 0.09 | 0.07 | 0.15 | 0.35 | 0.34 | 0.55 |
| TCN$_{herd}$ | 0.09 | 0.08 | 0.09 | 0.10 | 0.06 | 0.03 | 0.05 | 0.13 | 0.08 | 0.13 | 0.07 | 0.12 | 0.17 | 0.27 | 0.16 | 0.25 |
| TCN$_{rnd}$ | 0.10 | 0.11 | 0.11 | 0.13 | 0.04 | 0.04 | 0.10 | 0.10 | 0.04 | 0.16 | 0.02 | 0.02 | 0.21 | 0.37 | 0.29 | 0.51 |
| TCN$_{der++}$ | 0.10 | 0.10 | 0.08 | 0.09 | 0.01 | 0.08 | 0.06 | 0.10 | 0.04 | 0.09 | 0.03 | 0.07 | 0.25 | 0.37 | 0.14 | 0.23 |
| ESG$_{seq}$ | 0.48 | 0.70 | 0.65 | 0.87 | 0.62 | 0.67 | 0.76 | 0.83 | 0.31 | 0.30 | 0.35 | 0.31 | 0.10 | 0.65 | 0.35 | 0.33 |
| ESG$_{mir}$ | 0.43 | 0.37 | 0.19 | 0.45 | 0.18 | 0.18 | 0.13 | 0.27 | 0.82 | 0.57 | 0.59 | 0.54 | 0.88 | 1.12 | 0.97 | 1.08 |
| ESG$_{herd}$ | 0.45 | 0.52 | 0.41 | 0.48 | 0.16 | 0.30 | 0.20 | 0.36 | 0.73 | 1.32 | 1.01 | 1.33 | 0.51 | 0.56 | 0.55 | 0.41 |
| ESG$_{rnd}$ | 0.43 | 0.49 | 0.44 | 0.35 | 0.19 | 0.29 | 0.25 | 0.42 | 1.05 | 1.88 | 1.43 | 2.60 | 0.45 | 0.63 | 0.53 | 0.43 |
| ESG$_{der++}$ | 0.37 | 0.47 | 0.39 | 0.54 | 0.06 | 0.02 | 0.06 | 0.08 | 0.89 | 0.67 | 0.59 | 0.45 | 0.85 | 1.21 | 0.95 | 1.15 |
| GTS$_{seq}$ | 0.11 | 0.27 | 0.10 | 0.21 | 0.14 | 0.43 | 0.19 | 0.55 | 0.29 | 0.34 | 0.38 | 0.51 | 0.57 | 0.80 | 0.56 | 0.63 |
| GTS$_{mir}$ | 0.38 | 0.29 | 0.07 | 0.20 | 0.22 | 0.20 | 0.12 | 0.11 | 0.32 | 0.50 | 0.02 | 0.21 | 0.67 | 0.94 | 0.34 | 0.83 |
| GTS$_{herd}$ | 0.20 | 0.52 | 0.28 | 0.70 | 0.10 | 0.23 | 0.07 | 0.10 | 0.55 | 0.75 | 0.40 | 0.70 | 0.54 | 0.57 | 0.21 | 0.40 |
| GTS$_{rnd}$ | 0.18 | 0.29 | 0.20 | 0.39 | 0.03 | 0.14 | 0.06 | 0.08 | 0.58 | 0.81 | 0.51 | 0.65 | 0.43 | 0.45 | 0.28 | 0.42 |
| GTS$_{der++}$ | 0.23 | 0.28 | 0.16 | 0.19 | 0.04 | 0.15 | 0.07 | 0.21 | 0.30 | 0.45 | 0.11 | 0.20 | 0.61 | 0.95 | 0.48 | 0.85 |
| MTGNN$_{seq}$ | 0.31 | 0.35 | 0.15 | 0.30 | 0.03 | 0.06 | 0.05 | 0.07 | 0.10 | 0.15 | 0.10 | 0.24 | 0.41 | 0.64 | 0.24 | 0.10 |
| MTGNN$_{mir}$ | 0.35 | 0.12 | 0.17 | 0.15 | 0.05 | 0.12 | 0.10 | 0.18 | 0.11 | 0.23 | 0.31 | 0.31 | 0.53 | 0.52 | 0.25 | 0.19 |
| MTGNN$_{herd}$ | 0.11 | 0.29 | 0.33 | 0.31 | 0.13 | 0.15 | 0.11 | 0.22 | 0.09 | 0.39 | 0.26 | 0.36 | 0.60 | 0.58 | 0.17 | 0.27 |
| MTGNN$_{rnd}$ | 0.10 | 0.12 | 0.12 | 0.17 | 0.03 | 0.05 | 0.11 | 0.17 | 0.13 | 0.19 | 0.14 | 0.27 | 0.34 | 0.57 | 0.23 | 0.13 |
| MTGNN$_{der++}$ | 0.15 | 0.25 | 0.35 | 0.54 | 0.06 | 0.02 | 0.08 | 0.07 | 0.17 | 0.16 | 0.18 | 0.23 | 0.48 | 0.56 | 0.21 | 0.17 |
| Autoformer$_{seq}$ | 0.58 | 1.11 | 1.51 | 2.88 | 0.11 | 0.27 | 0.20 | 0.23 | 0.12 | 0.31 | 0.27 | 0.24 | 0.65 | 0.41 | 0.32 | 0.51 |
| Autoformer$_{mir}$ | 0.69 | 1.23 | 0.75 | 1.33 | 0.10 | 0.24 | 0.21 | 0.37 | 0.22 | 0.39 | 0.18 | 0.25 | 0.40 | 0.31 | 0.19 | 0.30 |
| Autoformer$_{herd}$ | 0.77 | 1.43 | 0.67 | 1.20 | 0.36 | 0.36 | 0.23 | 0.41 | 0.12 | 0.13 | 0.14 | 0.12 | 0.90 | 0.61 | 0.71 | 0.70 |
| Autoformer$_{rnd}$ | 0.86 | 1.42 | 0.74 | 1.18 | 0.18 | 0.31 | 0.24 | 0.31 | 0.17 | 0.27 | 0.35 | 0.31 | 0.46 | 0.31 | 0.25 | 0.43 |
| Autoformer$_{der++}$ | 0.84 | 1.54 | 0.63 | 1.34 | 0.26 | 0.43 | 0.22 | 0.40 | 0.18 | 0.30 | 0.21 | 0.23 | 0.72 | 0.71 | 0.54 | 0.47 |
| PatchTST$_{seq}$ | 0.27 | 0.21 | 0.25 | 0.46 | 0.24 | 0.17 | 0.12 | 0.21 | 0.24 | 0.10 | 0.26 | 0.15 | 0.16 | 0.54 | 0.40 | 0.54 |
| PatchTST$_{mir}$ | 0.13 | 0.11 | 0.17 | 0.21 | 0.29 | 0.12 | 0.21 | 0.23 | 0.21 | 0.11 | 0.09 | 0.10 | 0.32 | 0.42 | 0.49 | 0.57 |
| PatchTST$_{herd}$ | 0.17 | 0.13 | 0.19 | 0.17 | 0.12 | 0.08 | 0.12 | 0.14 | 0.24 | 0.10 | 0.27 | 0.13 | 0.10 | 0.54 | 0.40 | 0.49 |
| PatchTST$_{er}$ | 0.146 | 0.25 | 0.19 | 0.18 | 0.16 | 0.09 | 0.09 | 0.17 | 0.07 | 0.09 | 0.08 | 0.09 | 0.14 | 0.11 | 0.26 | 0.35 |
| PatchTST$_{der++}$ | 0.20 | 0.15 | 0.04 | 0.16 | 0.14 | 0.10 | 0.15 | 0.31 | 0.15 | 0.27 | 0.17 | 0.19 | 0.26 | 0.23 | 0.21 | 0.48 |
| Dlinear$_{seq}$ | 0.21 | 0.19 | 0.23 | 0.36 | 0.18 | 0.21 | 0.17 | 0.18 | 0.16 | 0.11 | 0.26 | 0.26 | 0.14 | 0.25 | 0.39 | 0.53 |
| Dlinear$_{mir}$ | 0.25 | 0.24 | 0.23 | 0.22 | 0.19 | 0.13 | 0.15 | 0.12 | 0.15 | 0.29 | 0.11 | 0.14 | 0.32 | 0.53 | 0.36 | 0.49 |
| Dlinear$_{herd}$ | 0.15 | 0.17 | 0.17 | 0.25 | 0.14 | 0.22 | 0.13 | 0.26 | 0.12 | 0.37 | 0.18 | 0.27 | 0.25 | 0.14 | 0.21 | 0.43 |
| Dlinear$_{er}$ | 0.23 | 0.17 | 0.18 | 0.12 | 0.11 | 0.14 | 0.16 | 0.23 | 0.14 | 0.31 | 0.27 | 0.23 | 0.22 | 0.46 | 0.56 | 0.62 |
| Dlinear$_{der++}$ | 0.16 | 0.08 | 0.08 | 0.14 | 0.13 | 0.14 | 0.08 | 0.28 | 0.01 | 0.16 | 0.13 | 0.12 | 0.29 | 0.49 | 0.23 | 0.42 |
| TimesNet$_{seq}$ | 0.17 | 0.14 | 0.23 | 0.22 | 0.09 | 0.13 | 0.15 | 0.08 | 0.16 | 0.21 | 0.11 | 0.14 | 0.32 | 0.53 | 0.36 | 0.49 |
| TimesNet$_{mir}$ | 0.12 | 0.16 | 0.18 | 0.20 | 0.10 | 0.08 | 0.14 | 0.12 | 0.14 | 0.12 | 0.27 | 0.21 | 0.14 | 0.25 | 0.31 | 0.35 |
| TimesNet$_{herd}$ | 0.14 | 0.19 | 0.17 | 0.25 | 0.14 | 0.20 | 0.09 | 0.26 | 0.15 | 0.37 | 0.09 | 0.32 | 0.23 | 0.14 | 0.21 | 0.43 |
| TimesNet$_{er}$ | 0.08 | 0.10 | 0.12 | 0.20 | 0.18 | 0.13 | 0.12 | 0.20 | 0.09 | 0.15 | 0.07 | 0.24 | 0.11 | 0.21 | 0.18 | 0.36 |
| TimesNet$_{der++}$ | 0.07 | 0.13 | 0.11 | 0.15 | 0.09 | 0.11 | 0.08 | 0.26 | 0.07 | 0.13 | 0.06 | 0.15 | 0.18 | 0.31 | 0.21 | 0.33 |
| SKI-CL$_{seq}$ | 0.30 | 0.40 | 0.39 | 0.54 | 0.03 | 0.03 | 0.04 | 0.04 | 0.03 | 0.02 | 0.04 | 0.17 | 0.17 | 0.26 | 0.17 | 0.23 |
| SKI-CL$_{mir}$ | 0.14 | 0.31 | 0.22 | 0.20 | 0.01 | 0.03 | 0.08 | 0.02 | 0.01 | 0.22 | 0.22 | 0.29 | 0.36 | 0.31 | 0.11 | 0.31 |
| SKI-CL$_{herd}$ | 0.15 | 0.21 | 0.14 | 0.20 | 0.05 | 0.07 | 0.07 | 0.09 | 0.05 | 0.12 | 0.15 | 0.19 | 0.18 | 0.25 | 0.25 | 0.36 |
| SKI-CL$_{rnd}$ | 0.04 | 0.10 | 0.04 | 0.16 | 0.06 | 0.06 | 0.05 | 0.10 | 0.02 | 0.09 | 0.23 | 0.21 | 0.16 | 0.22 | 0.35 | 0.49 |
| SKI-CL$_{der++}$ | 0.04 | 0.07 | 0.12 | 0.17 | 0.04 | 0.05 | 0.08 | 0.11 | 0.04 | 0.12 | 0.15 | 0.13 | 0.25 | 0.29 | 0.15 | 0.11 |
| SKI-CL | 0.04 | 0.06 | 0.03 | 0.03 | 0.02 | 0.05 | 0.03 | 0.08 | 0.04 | 0.09 | 0.08 | 0.09 | 0.27 | 0.27 | 0.13 | 0.09 |

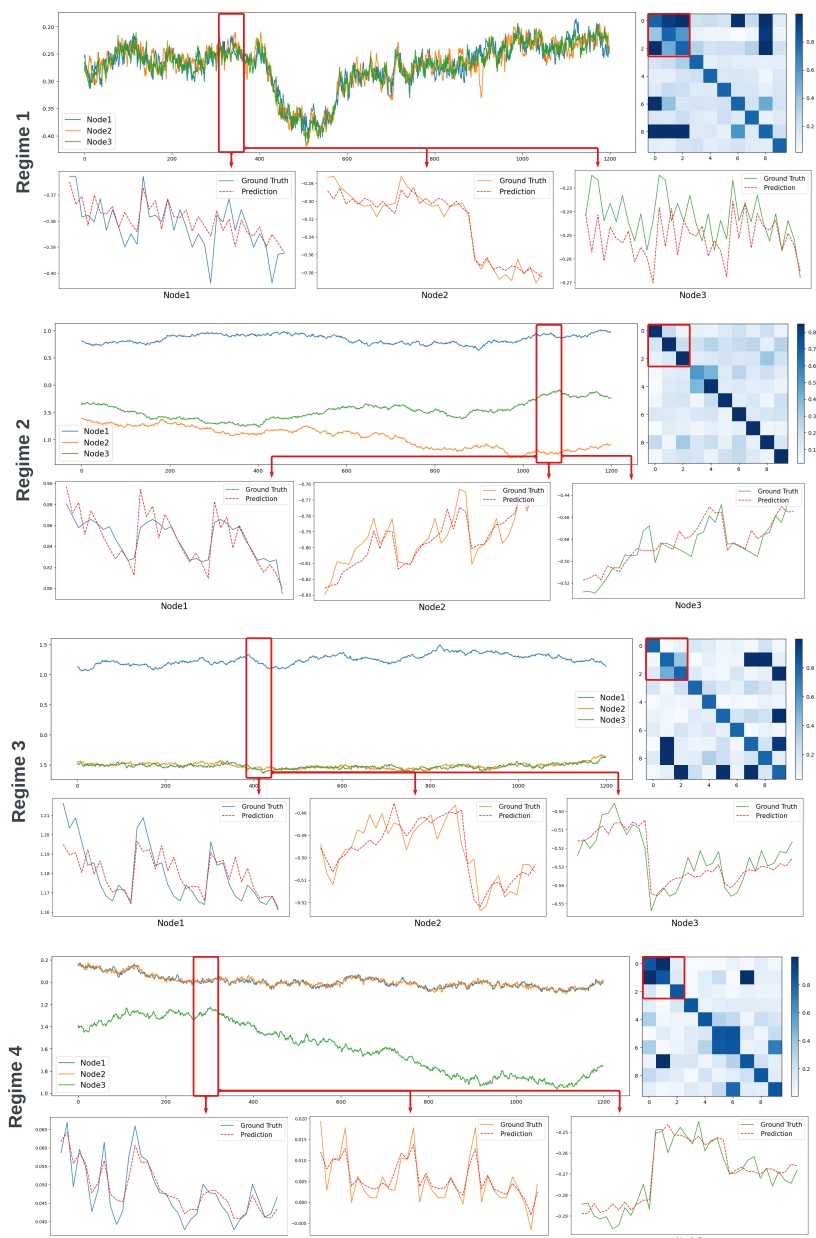

Figure 9: A case study of SKI-CL on Synthetic-CL dataset. In each regime, red rectangles indicate the correspondence between the ground truth time series (top-left) and the inferred variable dependencies (top-right), red arrows indicate the comparisons between these ground truth values and corresponding predictions (bottoms).

# G  A CASE STUDY WITH INFERRED DEPENDENCY STRUCTURES AND PREDICTION VISUALIZATIONS

We provide a case study on the Synthetic-CL dataset to illustrate the efficacy of SKI-CL, as shown in Figure 9. Our analysis is based on the final SKI-CL model that has been sequentially trained over all regimes. We select three variables (nodes) and visualize the testing data, where the temporal dynamics obviously differ across four regimes. It is clear that SKI-CL can render a faithful dependency structure that well aligns the similarity of variables in each regime (*e.g.*, three nodes are highly similar to each other in regime 1; only node 1 and node 2 are similar in regime 4). Moreover, SKI-CL gives relatively accurate forecasts that capture each variable's temporal dynamics of ground truths.

