# OpenReview forum: "Structural Knowledge Informed Continual Multivariate Time Series Forecasting"
_ICLR.cc/2024/Conference — Submitted to ICLR 2024_

### Official Review · Reviewer_1Nyy · 2023-10-24

**Soundness:** 2 fair
**Presentation:** 2 fair
**Contribution:** 2 fair
**Rating:** 3
**Confidence:** 4

**Summary:**

The paper proposes a novel framework called Structural Knowledge Informed Continual Multivariate Time Series Forecasting (SKI-CL) that leverages structural knowledge to improve MTS forecasting under the continual learning setting. The proposed framework consists of a deep forecasting model that incorporates a graph learner to capture the variable dependencies and a regularization scheme to ensure consistency between learned variable dependencies and structural knowledge. The authors address the challenge of modeling variable dependencies across different regimes while maintaining forecasting performance. The paper presents experimental results on several real-world datasets, demonstrating the effectiveness of the proposed framework in improving forecasting performance and maintaining consistency with structural knowledge.

**Strengths:**

1. The paper proposes a novel framework, SKI-CL, that leverages structural knowledge to improve MTS forecasting under the continual learning setting.
2. The proposed framework consists of a deep forecasting model that incorporates a graph learner to capture the variable dependencies and a regularization scheme to ensure consistency between learned variable dependencies and structural knowledge.
3. The paper presents experimental results on several real-world datasets, demonstrating the effectiveness of the proposed framework in improving forecasting performance and maintaining consistency with structural knowledge.

**Weaknesses:**

1. This paper is written in a way that takes time to understand the term "regime", so it is not easy to follow.
2. This paper omits related work on continual learning in time series and graph domains.
3. The hyperparameter analysis used in the sensitivity analysis is unclear, which reduces confidence in the experimental results.
4. There is no comparison of model complexity or training time with other baselines.
5. This paper can be read as an incremental work that brings the existing graph continual learning problem to the MTS domain. While the authors define the catastrophic forgetting problem in different regime settings in this paper, they overshadow all other similar problems in the time series domain. For example, this paper almost entirely ignores concept/temporal drift, dynamic graph learning, and continual learning on traditional time series. In this paper, it is necessary to clarify the similarities and differences with these fields and show the relative position in these research lines.

**Questions:**

1. The definition of regime written by the authors in the introduction is unclear, so readers may have difficulty understanding it. The motivation to solve problems caused by different regimes is good, but if readers do not understand regimes, the motivation of this paper may be meaningless. Readers may want to look at Figure 1 for simple examples of problems caused by different regimes instead of SKI-CL's framework.
2. The introduction on page 2 of this paper explains that the catastrophic forgetting problem in the MTS task is "the model performance will deteriorate over existing regimes as their associated structural knowledge cannot be maintained." Compared to the definitions of catastrophic forgetting in other existing domains, this is a setting in which no tasks or classes are added/incremented. Is this description appropriate compared to definitions in other domains?
3. This paper needs to add related research to papers that solve concept drift and temporal drift, where the distribution changes with time in a time series. In this paper, there is a need to add related research to the paper that addresses concept drift and temporal drift, where distributions change over time in time-series data. The authors also need to compare the models and experiments in these papers.
4. This paper needs to describe the position in the research of continuous learning, incremental learning, concept/temporal drift, and dynamic graph learning in each domain. Can the authors explain what the similarities and differences are with these research topics?
5. As written in the last paragraph of Section 2.2, the authors are aware that FSNet [1] uses MTS data, but do not consider it in the baseline due to the online learning setting. However, the experimental baseline of FSNet also performs including **DER++** and **ER**. This appears to be possible in the experimental settings of this paper, and comparative experiments are needed. If you can compare FSNet, compare the performance difference compared to the model proposed by the author.
6. Additionally, the paper makes no mention of the MIR [2] method. This should be considered similarly to FSNet above.
7. The authors need to mention that papers [3-5] dealing with temporal drift are either experimental baselines or related work.
8. Can you explain how the dependency structure learning proposed by the author differs from the graph structure parameterization in the GTS paper [6]? If there is a similarity, it is necessary to cite the GTS paper.
9. Section 4.4 of the paper explains as follows: “We perform experiments on the Traffic-CL dataset to validate the effectiveness and sensitivity of two key hyperparameters in SKI-CL, the weight of structure regularizer $\lambda$ (1 by default) and the memory budget (sampling ratio) at each regime (0.01 by default).” However, the results for $\lambda$ of 1 are not shown in Table 3. And the results in Table 3 and Table 4 do not show any settings that match the results in Table 2, so the reliability of the experimental results is reduced.
10. From the experimental results in Table 2, the AF of SKI-CL of the proposed model does not result in the lowest error compared to all baselines. For example, $\text{Autoformer}_{\text{der++}}$ has the lowest MAE for AF in Traffic-CL. There seems to be a lack of explanation as to why other baselines have lower errors.
11. There is a grammatical error in the sentence "given the a collection" above Eq.(1) on page 5.


> [1]: Learning Fast and Slow for Online Time Series Forecasting, ICLR 2023
> [2]: Online continual learning with maximal interfered retrieval, NeurIPS 2019
> [3]: AdaRNN: Adaptive learning and forecasting of time series, CIKM 2021
> [4]: Reversible Instance Normalization for Accurate Time-Series Forecasting against Distribution Shift, ICLR 2022
> [5]: Time Series Forecasting with Hypernetworks Generating Parameters in Advance, arXiv preprint arXiv:2211.12034, 2022
> [6]: Discrete Graph Structure Learning for Forecasting Multiple Time Series, ICLR, 2021

---

> ### Author Response · Authors · 2023-11-15
> **Response to Reviewer 1Nyy (1)**
>
> We sincerely thank the reviewer for recognizing our contribution and the constructive comments. The answers to the concerns and questions are listed below:
>
> **Weakness 1 and Question 1: Concerns regarding the definition of regime**
>
> We sincerely appreciate the reviewer for recognizing our contributions and raising concerns in terms of the definition of regimes. In the abstract and introduction, we referred “regime” to “stage” and indicated the underlying temporal dynamics and dependency structures of multivariate time series data for each regime is different (**abstract, the end of the first paragraph and the beginning of the second paragraph in introduction, page 1**). We also discussed this term multiple times under the context of continual learning across the introduction. As shown in Figure 1, we defined different regimes with raw multivariate time series and the corresponding structural knowledge, which are marked and annotated using different colors. Besides, in the beginning of the methodology, we gave the mathematical formulation of our continual forecasting task with a clear notion of regime. Again, we appreciate the reviewer for raising this concern and would emphasize the definition of regime at the beginning of our final manuscript and add another real-world example for demonstration.
>
> **Weakness 2: Paper omits related work on continual learning in time series and graph domains**
>
> We explicitly discussed the methods of continual learning in time series in our related work, **section 2.2**, where we also explicitly mentioned methods of continual learning on graphs.
>
> **Weakness 3 and Question 9:  Concerns of ablation study results with main experiment results**
>
> Main experiment results and ablation study for memory budgets are average experiment results that run by using different random seeds. The results have small differences due to randomness but similar trends are observed. We will release our code and models in the final version.
>
> **Weakness 4: Lack of comparison of model complexity or training time**
>
> We compare the number of parameters for each backbone models as following table:
> | Model | Parameter Number | Rank |
> | :--- | :---: | :---: |
> | LSTNet | 53253| 12 |
> | STGCN| 96606 | 11 |
> | MTGNN| 139990 | 10 |
> | AGCRN| 252130 | 8 |
> | GTS | 14647763 | 2 |
> | ESG | 5999516 | 4 |
> | TCN | 170886 | 9 |
> | StemGNN | 1060802 | 6 |
> | Autoformer | 10612758 | 3 |
> | PatchTST | 3226124 | 5 |
> | Dlinear | 49168 | 13 |
> | TimesNet | 36849590 | 1 |
> | SKI-CL | 614731 | 7 |
>
> The number of our models' parameters is significantly smaller than the SOTA baseline models, including Autoformer, PatchTST, TimesNet and can achieve the best performance for continual learning tasks.
>
> **Question 2 Concerns of the description of catastrophic forgetting with other domain**
>
> We appreciate the reviewer for raising this question. Yes, our description of catastrophic forgetting is appropriate. We would like to emphasize the **full context** of catastrophic forgetting that the reviewer refers to, in the first paragraph of page 2, is about modeling variable dependencies of MTS data for forecasting tasks (which is our research focus, and this problem is still underexplored as we discussed in the literature review, **the first paragraph of section 2.2 in page 3**).
> We refer to the definition of catastrophic forgetting in one of the most cited paper in continual learning [1],  *“Continual learning poses particular challenges for artificial neural networks due to the tendency for knowledge of the previously learned task(s) (e.g., task A) to be abruptly lost as information relevant to the current task (e.g., task B) is incorporated. This phenomenon, termed catastrophic forgetting, occurs specifically when the network is trained sequentially on multiple tasks because the weights in the network that are important for task A are changed to meet the objectives of task B.”*. Our description fits this definition, as the so-called “task” means regime in our case, and the “knowledge” within catastrophic forgetting involves the structural knowledge of MTS data learned by dependency modeling methods.
> The reviewer mentions that this is a setting in which no tasks or classes are added/incremented, which is under the context of classification tasks. In essence, our continual forecasting is a regime incremental setting in regression tasks, which is analogous to class-incremental tasks in classification tasks.
> Therefore, the description demonstrates the catastrophic forgetting problem under our multivariate time series forecasting task and is appropriate compared with continual learning in other domains.
>
> [1] Kirkpatrick, James, Razvan Pascanu, Neil Rabinowitz, Joel Veness, Guillaume Desjardins, Andrei A. Rusu, Kieran Milan et al. "Overcoming catastrophic forgetting in neural networks." Proceedings of the national academy of sciences 114, no. 13 (2017): 3521-3526.

---

> ### Author Response · Authors · 2023-11-15
> **Response to Reviewer 1Nyy (2)**
>
> **Weakness 5 and Question 4: The discussions of concept/temporal drift, dynamic graph learning and continual learning on traditional time series. Research positions**
>
> We appreciate the reviewer for raising this concern. Firstly, we would like to clarify that the scope of our paper is different from that of graph continual learning. Graph continual learning aims to maintain the model performance of all previous tasks on a sequence of explicit topological structures that is potentially evolving. However, in our work, we resort to modeling the dependency structure of MTS data that is coupled with the temporal dynamics, and leverage it for downstream forecasting tasks. This means that the dependency structure is not fixed and implicit within each regime.
> In addition to that, several terms are mentioned in this comment: concept/temporal drift, dynamic graph learning, and continual learning on traditional time series. For continual learning on traditional time series, we have discussed the continual learning methods for MTS data in the **section 2.2**, and we are not sure which reference the reviewer refers to here. It would be helpful to facilitate the discussion if specific literature can be pointed out.
> **Dynamic graph learning:** Next, we discuss the main differences between dynamic graph learning and our method. While both dynamic graph learning and our method deal with the evolving data, dynamic graph learning focuses on adapting to the changes in the explicit graph structure and possibly features over time. Like graph continual learning, dynamic graph learning uses the explicitly given graph structure for message passing and performs downstream tasks like node classification, link prediction, or community detection. While our proposed continual multivariate time series forecasting method needs to discover underlying dependencies structure of variables over different regimes, by capturing the temporal dynamics of MTS data. This also leads to the other key difference between dynamic graph learning and our method: the former always adapts its model to cope with the change (perform well on the new data), while the primary goal of the latter is the model's capability to continually learn without forgetting previous knowledge (i.e., addressing catastrophic forgetting). The focus of methodology can be very different due to the different learning objectives. We have similar discussion regarding the differences between online learning and our setting, in the literature review, **the second paragraph of section 2.2 in page 3**.
> **Concept drift/temporal drift:** Then we would like to discuss the relationship between our research scope and concept drift/temporal drift, and explain why these methods cannot serve as baselines for comparison(as the reviewer mentioned in the other question). Concept/temporal drift of MTS represents the phenomenon where the data distribution of the target MTS changes over time in unforeseen ways. Based on this notion, it is important to emphasize the continual adaptation nature of continual learning setting regarding the concept/temporal drifts, where the catastrophic forgetting is introduced due to the shifts over multiple regimes. That being said, continual MTS forecasters need to handle a variety of regimes, not just adapt to changes in data distribution for a single one, which is to some degree more realistic and challenging. Concept/temporal drift represents a specific challenge within this broader goal.
> Moreover, the learning objectives between concept/temporal drift methods and continual learning methods are very different. For the concept/temporal drift methods, the objective is to adapt the forecaster to these new MTS patterns by effectively detecting, responding to, and learning from these changes. In contrast, as we emphasized before, the main focus of continual learning methods is to prevent the model from forgetting previously learned knowledge (coupled temporal dynamics and variable dependencies in our scope) when adapting to new MTS distribution. Due to the aforementioned differences in adaptation nature and learning objectives, it is very difficult to make fair experimental comparisons under the same setting.
> Finally, regarding the reviewer’s concern, we would like to restate the position/scope of our research compared to the fields mentioned above. We start from the continual learning perspective, explore and address the challenges of MTS forecasting problem under this setting, where we resort to variable dependency modeling and characterization aided by structural knowledge. As we discussed in **section 2.2**, continual learning on MTS data and especially continual forecasting is still an underexplored problem, and we proposed a solution connecting both research areas. Even if the mentioned research fields and our work all tackle the evolving data, the problem formulation, challenges and learning objectives are significantly different as we discussed above.

---

> ### Author Response · Authors · 2023-11-15
> **Response to Reviewer 1Nyy (3)**
>
> **Question 3 and Question 7: The discussions of concept/temporal drift methods are needed.**
>
> As we discussed in our response to weakness 5 and question 4, while both concept drift methods and continual learning strategies deal with changing data environments, concept/temporal drift approaches are often more reactive and focused on adapting to changes in data distribution for a specific task. In contrast, continual learning involves a broader spectrum of adaptation, focusing on learning from new data and preventing forgetting over existing data. The continual adaptation nature of continual learning and very different objectives between both fields make it very difficult to include the concept/temporal drift methods as baselines of continual learning and make fair comparisons under the same setting.
> We agree with the reviewer to some extent that the concept/temporal drift and continual learning in MTS both tackle the evolving environment, and some ideas from one field can be shared to facilitate the advancement of each field. **Regarding the reviewer’s concern, we have discussed the aforementioned connections in our updated manuscript.** We would like to thank the reviewer for these references of temporal drifts and will try to involve more recent methods.
>
> **Question 5 and Question 6: Include the FSNet as a backbone baseline, and MIR as a continual learning baseline.**
>
> We sincerely appreciate the reviewer’s advice. However, FSNet is designed for online forecasting, which focuses on fast adaptation to newly incoming data and uses accumulated error on the training data as evaluation protocols (there is no notion of testing data to evaluate model performance at each regime). The objective of continual learning is to maintain the learned knowledge and model performance over all seen data when adapting to new data (as we discussed in **the second paragraph of section 2,2, page 3**). As such, the problem formulation, learning objective and evaluation protocols are very different to the continual learning setting, making it not very meaningful for comparison. DER++ and ER are commonly used experience-replayed methods for continual learning, so it is natural for us to include them as our baselines.
> The MIR method is also originally designed for online continual learning setting, which uses an in-batch virtual updated model to select samples that cause the highest forgetting to perform experience replay. To enable the fair comparison, we use the aforementioned MIR criteria to select a number of samples from the fixed budget memory per regime.  We report the other SOTA model results below and our proposed method consistently achieves the best performance.
>
> Traffic-CL:
> | Model | |  |  |  |
> | :---: | :---: | :---: | :---: | :---: |
> |  | AP_MAE | AP_RMSE | AF_MAE | AF_RMSE |
> | PatchTST_seq |19.11 | 32.50  | 2.34  | 2.97 |
> | PatchTST_mir | 19.04 | 32.23  | 2.28 | 2.79 |
> | PatchTST_herd| 18.96 | 32.10  | 2.21  | 2.67 |
> | PatchTST_er|   18.77 |  31.50 | 1.98 | 2.01 |
> | PatchTST_der++| 18.53 |  31.34  | 1.75  | 1.98 |
> | DLinear_seq| 19.69 | 32.75  |2.91  |2.83  |
> | DLinear_mir| 19.37 | 32.25  |2.17  |2.59 |
> | DLinear_herd| 19.53|  32.40 | 2.25 | 2.68 |
> | DLinear_er| 19.19 |  32.30  |1.73  |2.14 |
> | DLinear_der++| 19.02|  31.97  | 1.75  | 1.93 |
> | TimesNet_seq| 17.77 | 29.91 | 3.13 | 6.93 |
> | TimesNet_mir| 17.53 | 29.61 | 2.44 | 5.32 |
> | TimesNet_herd| 17.38 | 29.53 | 2.56 | 5.83 |
> | TimesNet_er| 17.25 | 29.33 | 1.97 | 4.19 |
> | TimesNet_der++| 17.13 | 29.28 | 1.56 | 4.02 |
> | SKI-CL| **15.24** | **25.33** | 1.50 | 2.71|
>
>
>
> Solar-CL:
> | Model | |  |  |  |
> | :---: | :---: | :---: | :---: | :---: |
> |  | AP_MAE | AP_RMSE | AF_MAE | AF_RMSE |
> | PatchTST_seq   | 2.64 | 5.32 | 0.72 | 0.43 |
> | PatchTST_mir   | 2.61 | 5.30 | 0.70 | 0.40 |
> | PatchTST_herd  | 2.60 | 5.30 | 0.68 | 0.35 |
> | PatchTST_er   | 2.57 | 5.27 | 0.47 | 0.30 |
> | PatchTST_der++ | 2.53 | 5.17 | 0.43 | 0.28 |
> | DLinear_seq    | 3.47 | 6.56 | 1.17 | 1.12 |
> | DLinear_mir    | 3.45 | 6.51 | 1.02 | 1.01 |
> | DLinear_herd   | 3.41 | 6.50 | 1.03 | 1.00 |
> | DLinear_er     | 3.37 | 6.43 | 0.93 | 0.98 |
> | DLinear_der++  | 3.25 | 6.37 | 0.83 | 0.79 |
> | TimesNet_seq   | 3.92 | 7.18 | 1.46 | 2.51 |
> | TimesNet_mir   | 3.77 | 7.15 | 1.22 | 1.57 |
> | TimesNet_herd  | 3.83 | 7.10 | 1.03 | 1.44 |
> | TimesNet_er    | 3.55 | 7.02 | 0.42 | 0.91 |
> | TimesNet_der++ | 3.45 | 6.55 | 0.37 | 0.90 |
> | SKI-CL                   | **1.75** | **4.46** | 0.09 | 0.06 |

---

> > ### Author Response · Authors · 2023-11-15
> > **Response to Reviewer 1Nyy (4)**
> >
> > HAR-CL:
> >
> > | Model | |  |  |  |
> > | :---: | :---: | :---: | :---: | :---: |
> > |  | AP_MAE | AP_RMSE | AF_MAE | AF_RMSE |
> > | PatchTST_seq   | 17.91 | 27.13 | 7.18 | 6.88 |
> > | PatchTST_mir  | 17.82 | 26.89 | 6.82 | 4.81 |
> > | PatchTST_herd>  | 17.73 | 26.84 | 6.62 | 4.79 |
> > | PatchTST_er   | 17.57 | 26.40 | 6.02 | 4.69 |
> > | PatchTST_der++ | 17.12 | 26.13 | 5.79 | 4.32 |
> > | DLinear_seq    | 17.32 | 26.31 | 2.71 | 3.43 |
> > | DLinear_mir    | 16.87 | 26.12 | 2.67 | 3.01 |
> > | DLinear_herd   | 16.83 | 25.81 | 2.57 | 2.91 |
> > | DLinear_er    | 16.71 | 25.75 | 2.13 | 2.85 |
> > | DLinear_der++  | 16.58 | 25.47 | 1.92 | 2.77 |
> > | TimesNet_seq   | 18.38 | 27.61 | 4.33 | 5.15 |
> > | TimesNet_mir   | 18.27 | 27.59 | 4.01 | 5.08 |
> > | TimesNet_herd  | 18.01 | 27.53 | 3.46 | 5.03 |
> > | TimesNet_er | 17.84 | 27.07 | 3.28 | 4.01 |
> > | TimesNet_der++ | 17.73 | 26.86 | 3.11 | 3.87 |
> > | SKI-CL                   | **13.41** | **21.30** | 1.64 | 2.08 |
> >
> >
> > Synthetic-CL:
> >
> > | Model | |  |  |  |
> > | :---: | :---: | :---: | :---: | :---: |
> > |  | AP_MAE | AP_RMSE | AF_MAE | AF_RMSE |
> > | PatchTST_seq   | 4.85 | 5.93 | 1.59 | 1.78 |
> > | PatchTST_mir   | 4.83 | 5.86 | 1.55 | 1.72 |
> > | PatchTST_herd  | 4.80 | 5.79 | 1.43 | 1.68 |
> > | PatchTST_er    | 4.72 | 5.26 | 1.03 | 1.54 |
> > | PatchTST_der++ | 4.64 | 5.13 | 0.83 | 0.88 |
> > | DLinear_seq    | 4.81 | 5.81 | 1.64 | 1.57 |
> > | DLinear_mir    | 4.79 | 5.73 | 1.47 | 1.40 |
> > | DLinear_herd   | 4.77 | 5.70 | 1.59 | 1.46 |
> > | DLinear_er     | 4.74 | 5.20 | 1.23 | 1.43 |
> > | DLinear_der++  | 4.21 | 4.88 | 1.12 | 1.13 |
> > | TimesNet_seq   | 5.18 | 6.13 | 1.72 | 2.03 |
> > | TimesNet_mir   | 5.12 | 6.05 | 1.69 | 1.97 |
> > | TimesNet_herd  | 5.10 | 6.03 | 1.68 | 1.95 |
> > | TimesNet_er.   | 4.93 | 5.90 | 1.42 | 1.90 |
> > | TimesNet_der++ | 4.81 | 5.88 | 1.32 | 1.78 |
> > | SKI-CL                   | **3.24** | **4.24** | 0.15 | 0.23 |

---

> ### Author Response · Authors · 2023-11-15
> **Response to Reviewer 1Nyy (5)**
>
> **Question 8: Explain the difference of dependency structure learning between the proposed method and GTS**
>
> We have cited the GTS paper in **introduction(page 2)**, **section 2.1 in literature review(page 3)** and the beginning of our dynamic graph inference module in **section 3.2 (page 4)**. Our dependency structure learning method has multiple important differences from GTS. Firstly, GTS learns a fixed discrete edge distribution based on Gumbel reparameterization trick from the whole training dataset, and samples static dependency structures for each time series window. However, our graph inference module learns a dynamic dependency structure based on the context of each time series window, which models the structure in a more fine-grained manner and copes with the dependencies variations within each regime. Furthermore, this leads to an important difference when doing continual learning, as we emphasized in **section 3.4 (page 6), Figure 1(page 2)** and the second paragraph in **Appendix C (page 19)**, our method is able to dynamically infer faithful dependency structures for existing and current regimes without accessing the memory buffer, while *GTS has to access the training data and memory buffer when inferring graphs at the testing stage, which is not realistic for practical model deployment in real-world applications*. Secondly, our method considers both discrete and continuous edge variable formulations with a flexible learning component and objective, while GTS focuses on the discrete edge scenario with a specialized Gumbel design.  Nevertheless, we agree with the reviewer that GTS and our method share the design choice of modeling an edge for all variable pairs instead of directly modeling the whole graph’s adjacency matrix.
>
> **Question 10: The concern regarding average forgetting in the performance evaluation**
>
> As we discussed in **Appendix B.3**, we emphasized the superiority of average performance over average forgetting in continual learning (**page 18**):
> *“Even if we provide both metrics for performance evaluation, we need to **emphasize the superiority of average performance over average forgetting** in continual learning. Average performance provides a direct measure of how well a learning system is performing on a task or set of tasks. It reflects the system’s ability to retain previously learned knowledge over past regimes while adapting to new information. While average forgetting is a relevant metric in assessing the memory capabilities of a learning system, it does not provide a complete picture of the learning system’s retention abilities. The average performance takes into account both the retention of old knowledge and the acquisition of new knowledge, providing a more comprehensive evaluation of the learning system’s performance. Therefore, we **use average performance as the main evaluation metric** and **average forgetting as an auxiliary metric** to measure knowledge retention and model adaptivity.”*
> Therefore, we are supposed to firstly look at the average performance. If the average performance is very close, then we pay attention to the average forgetting to further compare the model superiority. In the aforementioned example, autoformer-der++ does not yield satisfactory average performance over past regimes, thus it is less meaningful to directly evaluate the model solely via average forgetting.
>
> **Question 11: A typo in the sentence "given the a collection" above Eq.(1) on page 5**
>
> We sincerely appreciate the reviewer for pointing out the typo here, we corrected it in our updated manuscript.

---

> > ### Author Response · Authors · 2023-11-18
> > **We are happy to address any remaining/further concerns or questions.**
> >
> > We would like to thank the reviewer for acknowledging our contribution and providing us with constructive comments. We have made every effort to address the concerns and queries raised. As the time for discussion is limited, we would appreciate it if the reviewer could let us know if the concerns are resolved, and if there are any additional concerns or questions. We are happy to address them.

---

> ### Comment · Reviewer_1Nyy · 2023-11-21
> **Additional questions**
>
> The authors have addressed my concerns well. However, there are still some issues that remain. While I understand that our time is limited, I will consider this in my decision-making process.
>
> ------
>
> ## Questions on Traffic-CL dataset
> Firstly, I appreciate the authors' clarification regarding the definition of regimes, which has greatly aided my understanding. I have a question about the Traffic-CL dataset. I have a question about how the Traffic-CL dataset was used, as shown below:
>
> > Based on the constructed PEMSD3-Stream dataset, we make the following modifications to further simulate distinct regimes in the setting of continual forecasting. For each year, we rank and select the top 100 traffic sensors with the largest node degrees, based on which we randomly select 22 sensors as a set representing a part of the major traffic.
>
> Furthermore, the authors have sampled using 22 sensors to create the different regimes for each year. My concern arises from the choice of selecting sensor nodes based on the **largest node degree**. As you know, the graph structure for PeMSD3 is typically created based on the distance between sensors. In the PeMSD3-Stream dataset [1], *can we assume that selecting nodes based on node degree will generate distinct regimes?*
>
> This doubt leads me to suspect that the efficacy of the method proposed by the authors for Traffic-CL may be due to non-distinctive regimes.
> Since the nodes with the largest node degree are sampled, the dataset is already concentrated around the central nodes or hubs of the network in all regimes. For instance, in a road network, nodes with the largest degrees are likely to be located at points with heavy traffic (e.g., intersections). This could mean that areas with relatively less traffic might be overlooked in your Traffic-CL dataset. This suggests that the traffic volume at each node in each regime is likely to be already high relative to the traffic volume of others not selected. Alternatively, because nodes with high connectivity are selected, the traffic volume of nodes is likely to be similar across regimes.
>
> I believe this is a different issue from what is visualized in Figure 3. Could you provide information on how the traffic volume distribution varies across each regime in Traffic-CL to clarify this issue?
>
> In other words, it is necessary to reconsider whether the Traffic-CL dataset in which changes in structural dependency do not affect temporal dependency is suitable for continuous learning using structural knowledge for MTS.
>
>
> ## Appendix B
> My concerns have been somewhat addressed in Appendix B. While the explanation in Appendix B highlights the key differences between temporal/concept drift and continual learning, it might slightly oversimplify the relationship between these two areas. As you know, the temporal and concept drift methods can be an essential part of continual learning strategies.
>
> Additionally, the distinction between dynamic graph learning and the method proposed by the authors is still not readily apparent to the reader. I understand the authors' motivation and the differences between your method and others, but I suggest that elaborating on these differences more clearly in the introduction of the paper would be beneficial. Researchers in the dynamic graph learning and concept drift communities are likely to be particularly interested in the positioning of this paper.
>
>
>
> ## Minor comment
>
> Regarding citations in the paper, the use of \cite{} makes it difficult to distinguish between \cite{} and \citet{}. It would be more reader-friendly if the authors could change \cite{} to \citep{}.
>
> > [1] Xu Chen, Junshan Wang, and Kunqing Xie. "TrafficStream: A Streaming Traffic Flow Forecasting Framework Based on Graph Neural Networks and Continual Learning,” IJCAI, 2021. https://www.ijcai.org/proceedings/2021/0498.pdf

---

> > ### Author Response · Authors · 2023-11-21
> > **Response to Reviewer 1Nyy's Additional Questions (1)**
> >
> > **(Additional Question 1)  Question on Traffic-CL Dataset**
> >
> > We are pleased that we well addressed your previous concerns and questions. Next, we would like to address your new question and remaining concerns as follows, from the perspectives of experimental results and dataset statistics.
> > Firstly, we would like to have the reviewer pay attention to the performance evaluation (e.g., table 2 on page 7) of all methods with the very basic sequential training baseline without any aids from continual learning methods (with seq as the subscript). The sequential training baseline yields the worst forecasting performance and largest performance drops on *all* deep learning backbones, which *consistently demonstrates the existence of catastrophic forgetting issues* introduced by distinct regimes. Moreover, all continual learning methods improve the performance of sequential training baselines, which is also consistent for all backbones. These observations are very strong evidence showing the distribution disparity across regimes and demonstrate the efficacy of all continual learning methods including ours. If the regimes are non-distinct, the sequential training baseline should yield satisfactory results, which is apparently not the case in our experiments.
> >
> > Secondly, we would like to mention the statistics of the dataset for demonstrating our modification. The original PeMSD3-Stream dataset [1] is highly sparse given the number of nodes and edges (Please refer to Table 1 in this paper [1] the data statistics, based on which we calculated the edge density defined as the number of edges divided by the number of all possible edge, and the results from year 2011 to 2017 are, 0.736% / 0.754% / 0.75% / 0.75% / 0.746% / 0.744% / 0.736%).  Moreover, we have visualized the original traffic networks for each year. It turns out the nodes form many dense clusters which are mutually isolated in the sparse network. That being said, with many local dense structures in the network, it is not the case that the nodes in the current regimes are often selected from exactly the same clusters that have been sampled in the previous regimes.
> > Moreover, we would like to remind the reviewer of the temporal evolving nature of this traffic network. As this is a temporal expanding network, the dynamics of the same node also temporarily evolves as a consequence, which creates distribution disparity across years (again, this can be validated from our aforementioned results).
> >
> >
> > [1] Xu Chen, Junshan Wang, and Kunqing Xie. "TrafficStream: A Streaming Traffic Flow Forecasting Framework Based on Graph Neural Networks and Continual Learning,” IJCAI, 2021.
> >
> >
> > **(Additional Question 2) Concerns on Appendix B**
> >
> > Firstly, we recall our previous responses (weakness 5, questions 3,4,7) and emphasize that, due to the more challenging continual adaptation nature of continual learning and very different objectives between the two fields, it is not straightforward to use temporal and concept drift methods for continual learning. That being said, as our scope is continual forecasting, our focus is always to mitigate the catastrophic forgetting of the *past knowledge* under *continual* adaptation to the new regimes of MTS data. It needs more detailed explorations and rigorous considerations so that concept and temporal drift methods can be beneficial for continual forecasting, and it is all about design choices in the methodology development and research novelty. We would like to thank the reviewer for this idea and this meaningful discussion. We will consider and explore the underlying possibility in our future works.
> > Secondly, we also recap our response to weakness 5 and the updated manuscript for the distinctions that we have described. The first key difference is that, dynamic graph learning uses the *explicitly given graph structure for message passing* and performs downstream tasks, while our method needs to discover the underlying dependency structure of variables over different regimes (topological discovery), by capturing the temporal dynamics of MTS data. We perform *message passing on the discovered topology*, where the topological discovery is also involved in the objective optimization. The second key difference, as we discussed before, is the objective aspect. Dynamic graph learning always adapts its model to cope with the given topological and possibly feature change (perform well on the new data), while the primary goal of the continual MTS forecasting is to continually learn the model without forgetting previous knowledge (i.e., addressing catastrophic forgetting). We believe that these key differences make this concept less relevant to our scope. Nevertheless, we sincerely appreciate the kind advice on bringing forward these concepts to the introduction and may reconsider this advice for the manuscript organization, given the limited space.

---

> > > ### Author Response · Authors · 2023-11-21
> > > **Response to Reviewer 1Nyy's Additional Questions (2)**
> > >
> > > **(Additional Question 3) Minor comment on citing references**
> > >
> > > We appreciate the reviewer for this comment and would take the advice in our final manuscript.
> > >
> > >
> > >
> > > **We appreciate the reviewer's recognition and constructive feedback. Please let us know if there are any remaining concerns or questions and we are more than happy to address them.**

---

> ### Comment · Reviewer_1Nyy · 2023-11-23
> **Maintaing original score**
>
> I maintain my score due to concern regarding the regimes in the Traffic-CL dataset.
>
> As the authors say, in graphs where local clusters are formed, selecting the top 100 nodes with the highest node degree likely results in a high probability of choosing duplicate nodes. The selected nodes still represent structural dependency within the original networks, which calls into question the claim of different regimes.
>
> The visualizations in the paper for Traffic-CL only show the 22 nodes for each regime. From these visualizations, I cannot discern how different sensor nodes are positioned across each regime. Additionally, referring to the statistics in the Traffic-Stream paper, the rate of expansion in the graphs is not very fast. In fact, the expansion from 2014 to 2015 involves the addition of only about 12 nodes, suggesting that these added nodes are still likely to fall within local clusters.
>
> My continued skepticism about the “distinct regimes” claimed by the authors remains one of the most disappointing aspects of the paper. I find the methods used to form distinct regimes in Solar-CL and HAR-CL to be more convincing. In Solar-CL, different 5 states are used as distinct 5 regimes, whereas the distinct regimes in Traffic-CL seem less well-defined.

---

> ### Author Response · Authors · 2023-11-23
>
> We would really appreciate if the reviewer could check our earlier responses and the experimental results in the paper carefully, rather than misinterpreting it.
>
> **“As the authors say, in graphs where local clusters are formed, selecting the top 100 nodes with the highest node degree likely results in a high probability of choosing duplicate nodes.”**  This comment is completely opposite to our responses and experimental results, as we said **“it is not the case”**.
>
> Moreover, we need to remind the reviewer that the performance evaluation part is very strong evidence showing regime differences. Specifically, on Traffic-CL (as shown in Tables 2, 8, and 9), we have observed that *_seq* based methods perform significantly worse than other continual learning methods. This observation is consistent with the results (_seq) on other datasets, i.e., Solar-CL and HAR-CL (for which the reviewer has admitted the validity). This is strong evidence to demonstrate the validity of our setting and the potential distribution disparity.

---

### Official Review · Reviewer_RPp8 · 2023-10-31

**Soundness:** 3 good
**Presentation:** 3 good
**Contribution:** 4 excellent
**Rating:** 8
**Confidence:** 4

**Summary:**

This paper develops a novel structural knowledge informed continual learning framework to perform multivariate time series forecasting under continual learning setting. The key idea is to exploit structural knowledge to characterize the variable dependencies within each different regime and leverage a representation matching sample selection technique to construct memory buffer for replay. The experiment results on 4 public datasets showed the effectiveness of the proposed SKI-CL.

**Strengths:**

+ This paper is well written and organized. The structural knowledge informed continual learning framework is well-motivated and a comprehensive overview of related research is provided.
+ This paper introduces a novel SKI-CL framework for MTS forecasting and dependency structure inference under continual learning. This is an interesting, new, and practical MTS forecasting scenario to explore.
+ The proposed dynamic graph structure learning module to capture temporal dependencies and infer dependency structures are elegantly articulated and technically sound.
+ The idea to incorporate structural knowledge via adaptive regularization over the parameterized graph can enable the proposed model to infer the structural knowledge from all learned scenarios (regimes).
+ An innovative representation-matching memory replay scheme is proposed to maximize temporal data coverage and preserve dynamics and structures.
+ The experiment results are comprehensive and solid. Ablation studies of different components and some case studies are also provided.

**Weaknesses:**

- This paper studies MTS forecasting under continual learning setting, however, some datasets used for evaluation is standard benchmark. How do you define different regimes in this case?
- I notice that Autoformer which focuses on long term forecasting has been compared here. I wonder what’s the performance of PatchTST here, although it is also originally designed for long term forecasting.

**Questions:**

Please find the weaknesses.

---

> ### Author Response · Authors · 2023-11-15
> **Response to Reviewer RPp8 (1)**
>
> We sincerely thank the reviewer for recognizing our contribution and the constructive comments. The answers to the concerns and questions are listed below:
>
> **Weakness 1: How do you define different regimes for some benchmark data?**
> Thanks for raising this question! Two datasets are related to the standard benchmarks (PEMS and Solar), but extended by fetching more data under identical settings from the same databases. To be specific, the Traffic-CL dataset is based on an existing PEMSD3-Stream dataset [1] which uses the traffic data (Performance Measurement System in California, District 3) from the year 2011 to 2017. The standard PEMS benchmarks for general multivariate time series forecasting is a subset from this database. Based on the PEMSD3-Stream dataset, we further simulate distinct regimes represented/defined by a different portion of a temporally expanding traffic network (sensors with different geo-locations) from different years.
> For Solar-CL, we use the database of NREL’s Solar Power Data for Integration Studies, which contains solar power data in the United States for the year 2006. The standard benchmark uses the Alabama subset of this database for general multivariate time series forecasting. We construct different regimes by states (spatial locations) with different average annual sunlight levels based on the statistics.
> The detailed description of how we generate datasets with different regimes is provided in **appendix B.1 (pages 14-16)**.
> [1] Chen, Xu, Junshan Wang, and Kunqing Xie. "TrafficStream: A Streaming Traffic Flow Forecasting Framework Based on Graph Neural Networks and Continual Learning." Proceedings of the Thirtieth International Joint Conference on Artificial Intelligence (IJCAI-21)
>
> **Weakness 2: Comparison with PatchTST baseline**
>
> We agree with the reviewer that some long-term forecasting methods are beyond the scope of our research focus. The performance of PatchTST is shown as follows. (We also provide the evaluation results for Dlinear and TimesNet). Our proposed method consistently achieves the best performance.
>
> Traffic-CL:
> | Model | |  |  |  |
> | :---: | :---: | :---: | :---: | :---: |
> |  | AP_MAE | AP_RMSE | AF_MAE | AF_RMSE |
> | PatchTST_seq |19.11 | 32.50  | 2.34  | 2.97 |
> | PatchTST_mir | 19.04 | 32.23  | 2.28 | 2.79 |
> | PatchTST_herd| 18.96 | 32.10  | 2.21  | 2.67 |
> | PatchTST_er|   18.77 |  31.50 | 1.98 | 2.01 |
> | PatchTST_der++| 18.53 |  31.34  | 1.75  | 1.98 |
> | DLinear_seq| 19.69 | 32.75  |2.91  |2.83  |
> | DLinear_mir| 19.37 | 32.25  |2.17  |2.59 |
> | DLinear_herd| 19.53|  32.40 | 2.25 | 2.68 |
> | DLinear_er| 19.19 |  32.30  |1.73  |2.14 |
> | DLinear_der++| 19.02|  31.97  | 1.75  | 1.93 |
> | TimesNet_seq| 17.77 | 29.91 | 3.13 | 6.93 |
> | TimesNet_mir| 17.53 | 29.61 | 2.44 | 5.32 |
> | TimesNet_herd| 17.38 | 29.53 | 2.56 | 5.83 |
> | TimesNet_er| 17.25 | 29.33 | 1.97 | 4.19 |
> | TimesNet_der++| 17.13 | 29.28 | 1.56 | 4.02 |
> | SKI-CL| **15.24** | **25.33** | 1.50 | 2.71|
>
> Solar-CL:
> | Model | |  |  |  |
> | :---: | :---: | :---: | :---: | :---: |
> |  | AP_MAE | AP_RMSE | AF_MAE | AF_RMSE |
> | PatchTST_seq   | 2.64 | 5.32 | 0.72 | 0.43 |
> | PatchTST_mir   | 2.61 | 5.30 | 0.70 | 0.40 |
> | PatchTST_herd  | 2.60 | 5.30 | 0.68 | 0.35 |
> | PatchTST_er   | 2.57 | 5.27 | 0.47 | 0.30 |
> | PatchTST_der++ | 2.53 | 5.17 | 0.43 | 0.28 |
> | DLinear_seq    | 3.47 | 6.56 | 1.17 | 1.12 |
> | DLinear_mir    | 3.45 | 6.51 | 1.02 | 1.01 |
> | DLinear_herd   | 3.41 | 6.50 | 1.03 | 1.00 |
> | DLinear_er     | 3.37 | 6.43 | 0.93 | 0.98 |
> | DLinear_der++  | 3.25 | 6.37 | 0.83 | 0.79 |
> | TimesNet_seq   | 3.92 | 7.18 | 1.46 | 2.51 |
> | TimesNet_mir   | 3.77 | 7.15 | 1.22 | 1.57 |
> | TimesNet_herd  | 3.83 | 7.10 | 1.03 | 1.44 |
> | TimesNet_er    | 3.55 | 7.02 | 0.42 | 0.91 |
> | TimesNet_der++ | 3.45 | 6.55 | 0.37 | 0.90 |
> | SKI-CL                   | **1.75** | **4.46** | 0.09 | 0.06 |
>
>
> HAR-CL:
>
> | Model | |  |  |  |
> | :---: | :---: | :---: | :---: | :---: |
> |  | AP_MAE | AP_RMSE | AF_MAE | AF_RMSE |
> | PatchTST_seq   | 17.91 | 27.13 | 7.18 | 6.88 |
> | PatchTST_mir  | 17.82 | 26.89 | 6.82 | 4.81 |
> | PatchTST_herd>  | 17.73 | 26.84 | 6.62 | 4.79 |
> | PatchTST_er   | 17.57 | 26.40 | 6.02 | 4.69 |
> | PatchTST_der++ | 17.12 | 26.13 | 5.79 | 4.32 |
> | DLinear_seq    | 17.32 | 26.31 | 2.71 | 3.43 |
> | DLinear_mir    | 16.87 | 26.12 | 2.67 | 3.01 |
> | DLinear_herd   | 16.83 | 25.81 | 2.57 | 2.91 |
> | DLinear_er    | 16.71 | 25.75 | 2.13 | 2.85 |
> | DLinear_der++  | 16.58 | 25.47 | 1.92 | 2.77 |
> | TimesNet_seq   | 18.38 | 27.61 | 4.33 | 5.15 |
> | TimesNet_mir   | 18.27 | 27.59 | 4.01 | 5.08 |
> | TimesNet_herd  | 18.01 | 27.53 | 3.46 | 5.03 |
> | TimesNet_er | 17.84 | 27.07 | 3.28 | 4.01 |
> | TimesNet_der++ | 17.73 | 26.86 | 3.11 | 3.87 |
> | SKI-CL                   | **13.41** | **21.30** | 1.64 | 2.08 |
>
> Please see the next cell for experiment results for Synthetic-CL dataset

---

> ### Author Response · Authors · 2023-11-15
> **Response to Reviewer RPp8 (2)**
>
> Synthetic-CL:
>
> | Model | |  |  |  |
> | :---: | :---: | :---: | :---: | :---: |
> |  | AP_MAE | AP_RMSE | AF_MAE | AF_RMSE |
> | PatchTST_seq   | 4.85 | 5.93 | 1.59 | 1.78 |
> | PatchTST_mir   | 4.83 | 5.86 | 1.55 | 1.72 |
> | PatchTST_herd  | 4.80 | 5.79 | 1.43 | 1.68 |
> | PatchTST_er    | 4.72 | 5.26 | 1.03 | 1.54 |
> | PatchTST_der++ | 4.64 | 5.13 | 0.83 | 0.88 |
> | DLinear_seq    | 4.81 | 5.81 | 1.64 | 1.57 |
> | DLinear_mir    | 4.79 | 5.73 | 1.47 | 1.40 |
> | DLinear_herd   | 4.77 | 5.70 | 1.59 | 1.46 |
> | DLinear_er     | 4.74 | 5.20 | 1.23 | 1.43 |
> | DLinear_der++  | 4.21 | 4.88 | 1.12 | 1.13 |
> | TimesNet_seq   | 5.18 | 6.13 | 1.72 | 2.03 |
> | TimesNet_mir   | 5.12 | 6.05 | 1.69 | 1.97 |
> | TimesNet_herd  | 5.10 | 6.03 | 1.68 | 1.95 |
> | TimesNet_er.   | 4.93 | 5.90 | 1.42 | 1.90 |
> | TimesNet_der++ | 4.81 | 5.88 | 1.32 | 1.78 |
> | SKI-CL                   | **3.24** | **4.24** | 0.15 | 0.23 |

---

> > ### Author Response · Authors · 2023-11-18
> > **We are happy to address any further concerns or questions.**
> >
> > We sincerely thank the reviewer for acknowledging our contribution and providing us with constructive comments. We would appreciate it if the reviewer could let us know if the concerns are resolved, and if there are any additional concerns or questions. We are happy to address them.

---

> > > ### Comment · Reviewer_RPp8 · 2023-11-19
> > > **Maintain original score**
> > >
> > > All of my concerns have been addressed, and I maintain my original score.

---

### Official Review · Reviewer_YCYx · 2023-11-08

**Soundness:** 3 good
**Presentation:** 3 good
**Contribution:** 3 good
**Rating:** 5
**Confidence:** 3

**Summary:**

This paper propose a novel Structural Knowledge Informed Continual Learning (SKI-CL) framework to perform MTS forecasting under the
continual learning setting, which leverages the structural knowledge to characterize the dynamic variable dependencies within each regime.



In my opinion, the proposed dynamic graph learning module is not very novel, and many papers have used this structure, such as adaptive GCN, for forecasting. And the main contribution is applying your model in a continuous learning setting, which is not enough for ICLR.

**Strengths:**

1.  The paper is well written and easy to understand

2. The paper proposes incorporating knowledge for Dependencies Characterization.

**Weaknesses:**

1. My main concern is: why not compare it with a series of Sota time series models? And why not compare it with other continuing learning models?

2. The dynamic structure learning is not novel.

**Questions:**

What is structural knowledge? prior knowledge? or learned knowledge?

---

> ### Author Response · Authors · 2023-11-15
> **Response to Reviewer YCYx (1)**
>
> We sincerely thank the reviewer for recognizing our contribution and the constructive comments. The answers to the concerns and questions are listed below:
>
> **Weakness1 : Concerns about comparisons with other SOTA baselines**
>
> Thank you for the comments, this paper mainly focuses on continual learning setting of multivariate short-term forecasting and graph inference. We have compared our method with the multiple state-of-the-art graph learning backbones (that model variable dependencies for forecasting). We included results of autoformer, which currently is the third place of short-term forecasting tasks according to TSLib[1]. Regarding the reviewer’s concern, here we present more state-of-the-art model results, including PatchTST[2], Dlinear[3] and TimesNet[4] as below. As for the continual learning method, we have already included commonly used baselines for comparison. Now we also adapt and use MIR[5] for memory-replay continual learning baselines. Our proposed method consistently achieves best performance and outperform all baselines.
>
> Traffic-CL:
> | Model | |  |  |  |
> | :---: | :---: | :---: | :---: | :---: |
> |  | AP_MAE | AP_RMSE | AF_MAE | AF_RMSE |
> | PatchTST_seq |19.11 | 32.50  | 2.34  | 2.97 |
> | PatchTST_mir | 19.04 | 32.23  | 2.28 | 2.79 |
> | PatchTST_herd| 18.96 | 32.10  | 2.21  | 2.67 |
> | PatchTST_er|   18.77 |  31.50 | 1.98 | 2.01 |
> | PatchTST_der++| 18.53 |  31.34  | 1.75  | 1.98 |
> | DLinear_seq| 19.69 | 32.75  |2.91  |2.83  |
> | DLinear_mir| 19.37 | 32.25  |2.17  |2.59 |
> | DLinear_herd| 19.53|  32.40 | 2.25 | 2.68 |
> | DLinear_er| 19.19 |  32.30  |1.73  |2.14 |
> | DLinear_der++| 19.02|  31.97  | 1.75  | 1.93 |
> | TimesNet_seq| 17.77 | 29.91 | 3.13 | 6.93 |
> | TimesNet_mir| 17.53 | 29.61 | 2.44 | 5.32 |
> | TimesNet_herd| 17.38 | 29.53 | 2.56 | 5.83 |
> | TimesNet_er| 17.25 | 29.33 | 1.97 | 4.19 |
> | TimesNet_der++| 17.13 | 29.28 | 1.56 | 4.02 |
> | SKI-CL| **15.24** | **25.33** | 1.50 | 2.71|
>
>
> Solar-CL:
> | Model | |  |  |  |
> | :---: | :---: | :---: | :---: | :---: |
> |  | AP_MAE | AP_RMSE | AF_MAE | AF_RMSE |
> | PatchTST_seq   | 2.64 | 5.32 | 0.72 | 0.43 |
> | PatchTST_mir   | 2.61 | 5.30 | 0.70 | 0.40 |
> | PatchTST_herd  | 2.60 | 5.30 | 0.68 | 0.35 |
> | PatchTST_er   | 2.57 | 5.27 | 0.47 | 0.30 |
> | PatchTST_der++ | 2.53 | 5.17 | 0.43 | 0.28 |
> | DLinear_seq    | 3.47 | 6.56 | 1.17 | 1.12 |
> | DLinear_mir    | 3.45 | 6.51 | 1.02 | 1.01 |
> | DLinear_herd   | 3.41 | 6.50 | 1.03 | 1.00 |
> | DLinear_er     | 3.37 | 6.43 | 0.93 | 0.98 |
> | DLinear_der++  | 3.25 | 6.37 | 0.83 | 0.79 |
> | TimesNet_seq   | 3.92 | 7.18 | 1.46 | 2.51 |
> | TimesNet_mir   | 3.77 | 7.15 | 1.22 | 1.57 |
> | TimesNet_herd  | 3.83 | 7.10 | 1.03 | 1.44 |
> | TimesNet_er    | 3.55 | 7.02 | 0.42 | 0.91 |
> | TimesNet_der++ | 3.45 | 6.55 | 0.37 | 0.90 |
> | SKI-CL                   | **1.75** | **4.46** | 0.09 | 0.06 |
>
> HAR-CL:
>
> | Model | |  |  |  |
> | :---: | :---: | :---: | :---: | :---: |
> |  | AP_MAE | AP_RMSE | AF_MAE | AF_RMSE |
> | PatchTST_seq   | 17.91 | 27.13 | 7.18 | 6.88 |
> | PatchTST_mir  | 17.82 | 26.89 | 6.82 | 4.81 |
> | PatchTST_herd>  | 17.73 | 26.84 | 6.62 | 4.79 |
> | PatchTST_er   | 17.57 | 26.40 | 6.02 | 4.69 |
> | PatchTST_der++ | 17.12 | 26.13 | 5.79 | 4.32 |
> | DLinear_seq    | 17.32 | 26.31 | 2.71 | 3.43 |
> | DLinear_mir    | 16.87 | 26.12 | 2.67 | 3.01 |
> | DLinear_herd   | 16.83 | 25.81 | 2.57 | 2.91 |
> | DLinear_er    | 16.71 | 25.75 | 2.13 | 2.85 |
> | DLinear_der++  | 16.58 | 25.47 | 1.92 | 2.77 |
> | TimesNet_seq   | 18.38 | 27.61 | 4.33 | 5.15 |
> | TimesNet_mir   | 18.27 | 27.59 | 4.01 | 5.08 |
> | TimesNet_herd  | 18.01 | 27.53 | 3.46 | 5.03 |
> | TimesNet_er | 17.84 | 27.07 | 3.28 | 4.01 |
> | TimesNet_der++ | 17.73 | 26.86 | 3.11 | 3.87 |
> | SKI-CL                   | **13.41** | **21.30** | 1.64 | 2.08 |
>
> Synthetic-CL:
>
> | Model | |  |  |  |
> | :---: | :---: | :---: | :---: | :---: |
> |  | AP_MAE | AP_RMSE | AF_MAE | AF_RMSE |
> | PatchTST_seq   | 4.85 | 5.93 | 1.59 | 1.78 |
> | PatchTST_mir   | 4.83 | 5.86 | 1.55 | 1.72 |
> | PatchTST_herd  | 4.80 | 5.79 | 1.43 | 1.68 |
> | PatchTST_er    | 4.72 | 5.26 | 1.03 | 1.54 |
> | PatchTST_der++ | 4.64 | 5.13 | 0.83 | 0.88 |
> | DLinear_seq    | 4.81 | 5.81 | 1.64 | 1.57 |
> | DLinear_mir    | 4.79 | 5.73 | 1.47 | 1.40 |
> | DLinear_herd   | 4.77 | 5.70 | 1.59 | 1.46 |
> | DLinear_er     | 4.74 | 5.20 | 1.23 | 1.43 |
> | DLinear_der++  | 4.21 | 4.88 | 1.12 | 1.13 |
> | TimesNet_seq   | 5.18 | 6.13 | 1.72 | 2.03 |
> | TimesNet_mir   | 5.12 | 6.05 | 1.69 | 1.97 |
> | TimesNet_herd  | 5.10 | 6.03 | 1.68 | 1.95 |
> | TimesNet_er.   | 4.93 | 5.90 | 1.42 | 1.90 |
> | TimesNet_der++ | 4.81 | 5.88 | 1.32 | 1.78 |
> | SKI-CL                   | **3.24** | **4.24** | 0.15 | 0.23 |

---

> > ### Author Response · Authors · 2023-11-18
> > **We are happy to address any remaining/further concerns or questions.**
> >
> > We sincerely thank the reviewer for acknowledging our contribution and providing constructive feedback. We have made our best effort to address the raised concerns and questions. Considering the limited time for discussion, we would appreciate it if the reviewer could let us know if all the concerns have been satisfactorily resolved. We are also happy to address any remaining or further concerns or questions.

---

> > > ### Author Response · Authors · 2023-11-21
> > > **A friendly reminder to Reviewer YCYx that the discussion ends in one day**
> > >
> > > Greetings! We would like to express our gratitude for your insightful comments during the review process. We have taken great care to address the concerns comprehensively and provide responses. Given that the discussion is ending in only one day, we would really appreciate if the reviewer could let us know if the concerns are resolved, and if there is any remaining/additional concern/question. We are more than happy to address any remaining/further concern/question. Thank you for your attention to this, and we look forward to receiving your valuable insights.

---

> > > > ### Author Response · Authors · 2023-11-23
> > > > **A friendly reminder to Reviewer YCYx that the discussion period is ending soon**
> > > >
> > > > As the review period is ending soon, we sincerely appreciate your recognition of our efforts and the invaluable feedback provided. we have taken great care to address all concerns and questions raised. Given the critically short time remaining, it is important for us to know if our responses have resolved all your concerns. Please let us know any unresolved issues or additional inquiries you may have.

---

> ### Author Response · Authors · 2023-11-15
> **Response to Reviewer YCYx (2)**
>
> **Weakness2: The dynamic structure learning is not novel.**
>
> The existing literature either learns the structure via training parameters or representations. Our novelty lies in the role of dynamic structure learning for regime characterization, rather than an individual structure learning component. It is important that we impose the consistency regularization in the dynamic structure learning process, which aligns the learned structure with the universal and task-irrelevant structural knowledge to characterize a specific regime. The joint structure modeling based on both components leads to an capability that the existing backbones cannot achieve, i.e., **during the inference stage, our model can automatically infer a consistent structure solely based on the time series data, without knowing the regime and accessing the memory buffer**, as shown in **Figure 1: SKI-CL: Testing (page 2) and validated in Figure 4 (page 9)**. Furthermore, we are the first to explore and test different structure learning scenarios regarding the formulation of edge variables (discrete or continuous) and availability of structural knowledge (completed or partial) for multivariate time series forecasting, with visualizations in **Figures 5-8 in appendix (page 19-21)**.
>
>
> **Question 1: Questions about structure knowledge**
>
> Thanks for raising this question. As we discussed in the introduction (**the last three lines in page 1 and the first six lines in page 2**) and literature review(**section 2.1 lines 10-17**), the structural knowledge can be prior knowledge (e.g., physical constraints like traffic network and power grids, domain knowledge of application scenarios like Mel-frequency cepstral coefficients) or the so-called ‘learned’ knowledge if the structure is directly inferred from data by traditional statistical methods(e.g., correlation, k-NN), or learned and inferred individually (e.g., transfer entropy/granger causality).
>
>
> [1]THUML. "Time-Series-Library." GitHub repository, URL: https://github.com/thuml/Time-Series-Library. Accessed on November 12, 2023
>
> [2]Yuqi Nie, Nam H. Nguyen, Phanwadee Sinthong, Jayant Kalagnanam "A time series is worth 64 words: Long-term forecasting with transformers." in Proceedings of the International Conference on Learning Representations (2023)
>
> [3]Zeng, Ailing, Muxi Chen, Lei Zhang, and Qiang Xu. "Are transformers effective for time series forecasting?." In Proceedings of the AAAI conference on artificial intelligence(2023)
>
> [4]Haixu Wu and Tengge Hu and Yong Liu and Hang Zhou and Jianmin Wang and Mingsheng Long. ”TimesNet: Temporal 2D-Variation Modeling for General Time Series Analysis.” in Proceedings of the International Conference on Learning Representations (2023)
>
> [5]Rahaf, Aljundi, and Caccia Lucas. "Online Continual Learning with Maximally Interfered Retrieval." Advances in neural information processing systems 32 (2019).

---

### Author Response · Authors · 2023-11-22
**Summary of main contributions, concerns and our responses (Part1)**

Dear Area Chairs and Reviewers,

Thanks for the constructive suggestions from all reviewers. We summarize the main contributions, concerns from the reviewers, and our corresponding replies as follows.

**Main Contributions**

All three reviewers acknowledged the incorporation of structural knowledge that characterizes regime dependencies and facilitates continual MTS forecasting. Reviewer $RPp8$ acknowledges our problem setting is an interesting, new, and practical MTS forecasting scenario, and our proposed SKI-CL framework is well motivated based on a comprehensive overview of related work. Two reviewers ($RPp8$, $1Nyy$) highlighted the novelty of the framework design that models dynamic variable dependencies with a consistency regularization, reviewer $RPp8$ also thought it is elegantly articulated and technically sound. Reviewer $RPp8$ saw an innovation of our memory design that maximizes temporal data coverage to preserve structural and temporal dependencies. Two reviewers ($RPp8$, $1Nyy$) acknowledged that our experiments are comprehensive and solid/can demonstrate the effectiveness of our methodology and our improvements regarding forecasting performance. Two reviewers($YCYx$, $RPp8$) agreed that our paper is well-written and organized.

**Main Concerns and Responses**

**(Reviewer $YCYx$)**

**Why not compare it with other SOTA time series models and other continual learning models?**

We stated that we have compared our method with the multiple SOTA graph learning based backbones, and commonly used continual learning baselines. For backbones, We further added three more recent SOTA methods (PatchTST, DLinear, TimesNet). For the continual learning methods, we adapted an online continual learning method (MIR) and used it as the baseline. The experiments on all datasets still demonstrate our method’s superiority. The full experiment results can be found in the appendix of our updated manuscript (table 7, page 25).

**The dynamic structure learning is not novel.**

We referred to our manuscript and justified that our novelty lies in the role of dynamic structure learning for regime characterization (which leads to a capability that the existing backbones cannot achieve), rather than an individual structure learning component. In our work, we have also explored and tested different structure learning scenarios (edge discreteness and the availability of the structural knowledge) based on this formulation.

**What is structural knowledge? prior knowledge? or learned knowledge?**

We referred to our original explanations in the introduction and related work sections and clarified that structural knowledge can be prior knowledge, knowledge directly inferred by traditional statistical methods, and knowledge learned and inferred individually via sophisticated methods.

**(Reviewer $RPp8$)**

**Some standard benchmarks for MTS forecasting are used in the continual forecasting setting, how different regimes are defined?**

We clarified that two of our datasets (Traffic-CL, Solar-CL) and the standard benchmarks (PEMS and Solar) belong to the same databases, but our datasets can be treated as an extension as they also contain the data fetched from more years/geo-locations(which define regimes). We also referred to the detailed dataset description in the appendix.

**Since Autoformer on long term forecasting has been compared, the reviewer wondered about the performance of PatchTST.**

We presented the performance of PatchTST, where we also provided the evaluation results for DLinear and TimesNet. The experiments show that our proposed method consistently achieves the best performance for continual MTS forecasting.  The full experiment results can be found in the appendix of our updated manuscript (table 7, page 25).

**(Reviewer $1Nyy$)**

**It takes time to understand the definition of regime**

We referred to the abstract and introduction where we described the regime behavior and discussed this term multiple times under the context of continual learning. We also gave the mathematical formulation at the beginning of the methodology. We would emphasize the definition of regime at the beginning of our final manuscript with another real-world example for demonstration.

**Paper omits related work on continual learning in time series and graph domains**

In our manuscript, we explicitly discussed the methods of continual learning in time series in our related work, where we also explicitly mentioned methods of continual learning on graphs.

**Concerns of ablation study results with main experiment results**

We clarified that the main experiment results and ablation study for memory budgets are average experiment results that run by using different random seeds. The results have small differences due to randomness but similar trends are observed. We will release our code and models in the final version.

---

> ### Author Response · Authors · 2023-11-22
> **Summary of main contributions, concerns and our responses (Part2)**
>
> **Lack of comparison of model complexity or training time**
>
> We compared the number of parameters for each backbone model in the table.
>
> **This paper can be read as an incremental work that brings the existing graph continual learning problem to the MTS domain. The discussions of concept/temporal drift, dynamic graph learning and continual learning on traditional time series are needed for the research positions of this paper.**
>
> Firstly, we argued that we have discussed continual learning on time series in our manuscript. We then referred to our manuscript and clarified that the scope of our paper is different from that of the aforementioned research topics, even if those methods all tackle evolving data. We made detailed discussions regarding the differences and restated our research position in our response. We also added the discussion in the appendix of our updated manuscript.
>
> **Compared to the definitions of catastrophic forgetting in other existing domains, this is a setting in which no tasks or classes are added/incremented. Is this description appropriate compared to definitions in other domains?**
>
> We emphasized the full context of catastrophic forgetting that the reviewer refers to, which is about modeling variable dependencies of MTS data for forecasting tasks. This is our research focus, and this problem is still underexplored as we discussed in the literature review. Then we referred to the definition of catastrophic forgetting in one of the most cited papers in continual learning, and justified the appropriateness of our description. Regarding the comment “no tasks or classes are added/incremented”, which is under the context of classification tasks, we clarified that our continual forecasting is essentially a regime incremental setting in regression tasks, which is analogous to class-incremental tasks in classification tasks.
>
> **The discussions and experiments of concept/temporal drift methods are needed.**
>
> We restated the major differences between our method and these methods (detailed discussion can be found in the previous response), and indicated that it is hard to include these methods as baselines for fair comparisons due to these differences.
>
> **Include the FSNet as a backbone baseline, and MIR as a continual learning baseline.**
>
> We indicated that FSNet is an online time series forecasting method whose problem formulation, learning objective and evaluation protocols are very different to the continual learning setting, making it not very meaningful for comparison. We then adapted the MIR method (originally designed for online continual learning) to our setting and used it as the baseline. We also reported the experiment results with more backbone models, which demonstrated that our proposed method consistently achieves the best performance. The full experiment results can be found in the appendix of our updated manuscript (table 7, page 25).
>
>
> **Explain the difference of dependency structure learning between the proposed method and GTS**
>
> We clarified that we have discussed the GTS paper in our manuscript. We referred to our manuscript and emphasized the key differences between our method and GTS, and an important advantage of our method over GTS (our method automatically infers faithful dependency structures for the corresponding regimes without accessing the memory buffer). We agreed with the reviewer that both methods share the design choice of modeling edge variables.
>
> ** The concern regarding average forgetting in performance evaluation **
>
> We referred to the appendix and emphasized the superiority of average performance over average forgetting in continual learning.
>
> **A typo in the sentence "given the a collection" above Eq.(1) on page 5**
>
> We appreciated the reviewer for pointing out the typo and corrected it in our updated manuscript.
>
>
> **New question regarding the non-distinct regime in Traffic-CL dataset**
>
> We are delighted that we have addressed the previous concerns and we demonstrated the regime distinction by referring to and discussing the worst forecasting performance of the sequential training baseline in the experiment results. We also discussed the distinctions from the dataset statistics, and further emphasized the evolving nature of this dataset that creates distribution disparity.
>
>
> **Remaining concern regarding the discussion of related topics in the updated manuscript**
>
> We recalled our previous responses to emphasize the validity of what we have discussed, and thanked the reviewer for the ideas, advice and meaningful discussions.
>
> **Comment on the way of citing papers**
>
> We sincerely thanked the reviewer's suggestions and would take the advice in our final manuscript.

---

### Meta-Review · Area_Chair_8dyG · 2023-12-10

**Metareview:**

The reviewers raised multiple concerns, including the experimental setting, baselines and writing. Despite likely large improvements from author feedback, a consensus on paper acceptance wasn't reached due to a brief discussion period and substantial revisions undertaken during this period.

**Justification For Why Not Higher Score:**

Despite likely large improvements from author feedback, a consensus on paper acceptance wasn't reached due to a brief discussion period and substantial revisions undertaken during this period.

**Justification For Why Not Lower Score:**

n/a

---

### Decision · Program_Chairs · 2024-01-16

Reject